# Deep learning-driven adaptive optics for single-molecule localization microscopy

Peiyi Zhang [1], Donghan Ma [1,2], Xi Cheng[3,4], Andy P. Tsai[5], Yu Tang[3,4], Hao-Cheng Gao[1], Li Fang[1], Cheng Bi[1], Gary E. Landreth [5,6] ✉, Alexander A. Chubykin [3,4] ✉ & Fang Huang [1,4,7] ✉

The inhomogeneous refractive indices of biological tissues blur and distort single-molecule emission patterns generating image artifacts and decreasing the achievable resolution of single-molecule localization microscopy (SMLM). Conventional sensorless adaptive optics methods rely on iterative mirror changes and image-quality metrics. However, these metrics result in inconsistent metric responses and thus fundamentally limit their efficacy for aberration correction in tissues. To bypass iterative trial-then-evaluate processes, we developed deep learning-driven adaptive optics for SMLM to allow direct inference of wavefront distortion and near real-time compensation. Our trained deep neural network monitors the individual emission patterns from single-molecule experiments, infers their shared wavefront distortion, feeds the estimates through a dynamic filter and drives a deformable mirror to compensate sample-induced aberrations. We demonstrated that our method simultaneously estimates and compensates 28 wavefront deformation shapes and improves the resolution and fidelity of three-dimensional SMLM through >130-μm-thick brain tissue specimens.

Fluorescence microscopy is an indispensable tool in visualizing cellular and tissue machinery with molecular specificity; however, in its conventional form, the resolution is limited to 250–700 nm laterally and axially due to the diffraction of light[1]. Molecular features smaller than this limit cannot be resolved. Super-resolution microscopies such as stimulated emission depletion microscopy[2], structured illumination microscopy[3] and SMLM[4–6] have overcome this barrier, allowing biological observations[7–10] well beyond this fundamental limit of light. In particular, SMLM detects individual molecules using photo-switchable or convertible fluorescent dyes or proteins, pinpoints the centers of probes from their emission patterns and reconstructs the molecular centers into a super-resolution image. The unique advantage of SMLM

lies in measuring individual molecules without ensemble averaging and, therefore, its potential in molecular counting and ultra-high resolution in both live and fixed specimens[11,12]. Localization precision as low as 1–10 nm can be achieved in fixed and living cells[13–16].

SMLM in tissues, however, is challenging. One major reason is the distortion and blurring of single-molecule emission patterns (that is, point spread functions (PSFs)) caused by the inhomogeneous refractive indices within the tissue. Such alterations often reduce the information content[17] carried by each detected photon, worsening the theoretically achievable localization precision and thus causing resolution loss, which is irreversible by post-processing[18]. Reversing these sample-induced aberrations requires optical path modifications in a

[1]Weldon School of Biomedical Engineering, Purdue University, West Lafayette, IN, USA. [2]Davidson School of Chemical Engineering, Purdue University, West Lafayette, IN, USA. [3]Department of Biological Sciences, Purdue University, West Lafayette, IN, USA. [4]Purdue Institute for Integrative Neuroscience, Purdue University, West Lafayette, IN, USA. [5]Stark Neurosciences Research Institute, Indiana University School of Medicine, Indianapolis, IN, USA. [6]Department of Anatomy, Cell Biology and Physiology, Indiana University School of Medicine, Indianapolis, IN, USA. [7]Purdue Institute of Inflammation, Immunology and Infectious Disease, Purdue University, West Lafayette, IN, USA. ✉e-mail: glandret@iu.edu; chubykin@purdue.edu; fanghuang@purdue.edu

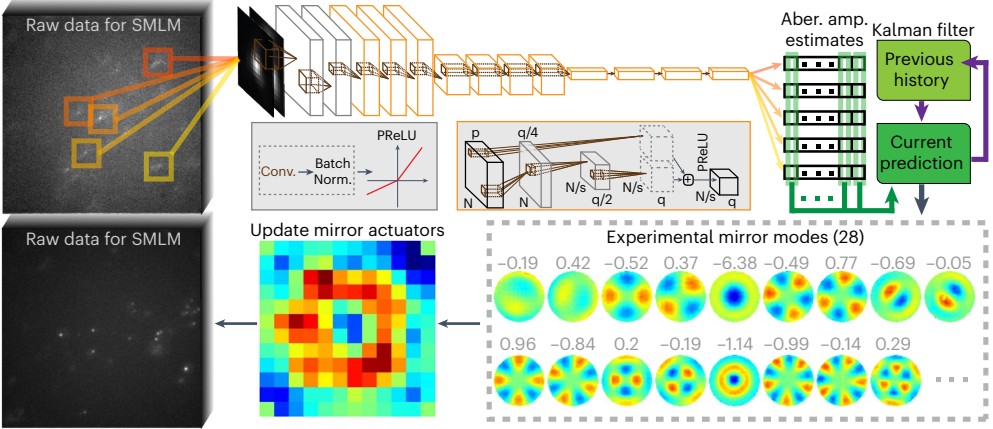

**Fig. 1 | Deep learning-driven adaptive optics for single-molecule localization microscopy.** Upon the acquisition of camera frames, detected single-molecule emission patterns from stochastic lateral and axial positions are isolated and sent to a trained DNN. The network outputs a vector of mirror deformation-mode amplitudes, for each detection of a single molecule. The estimations before and after each compensation are then combined through a Kalman filter to drive the next deformable mirror update. 'p' and 'q' represent numbers of feature maps input and output to a residue block (the orange box). 'N' represents the image width/height. 's' is stride size in a convolutional layer. The detailed sizes in each layer of the network architecture can be found in Supplementary Table 1.

microscopy system, commonly with a deformable mirror or a spatial light modulator, responsive toward each specimen and field of view to adaptively restore the PSFs of single emitters and thus the achievable resolution. This process is known as adaptive optics (AO)[19–23].

To guide a deformable mirror to compensate sample-induced aberrations, the distorted wavefront needs to be measured[21,22]. For point-scanning methods, such as confocal and multiphoton microscopy, the detection focus serves as a 'guide star' providing a stable wavefront measurable with a sensor[22–26]. For wide-field fluorescence modalities, such as structured illumination microscopy, a guide star could be generated by multiphoton excitation[27,28], or by embedding a fluorescent bead in the specimen. In contrast, wavefronts from single-molecule emissions, despite their abundance in SMLM experiments, cannot be directly measured as signals from individual molecules blinking stochastically with limited photons, and thus they do not provide the bright and stable signals required for guide stars[18]. In addition, introducing external guide stars, such as fluorescent beads in SMLM, may drastically increase the fluorescence background, which reduces the detectability of single-molecule emission patterns and generates structured background patterns resulting in localization artifacts[18]. Besides, if measured directly, wavefronts are composed of not only the aberrated wavefront induced by the specimen, but also the wavefront variations from both in-focus and out-of-focus single emitters at different lateral and axial positions, making it difficult to measure wavefront distortion specific for the SMLM imaging volume.

For this reason, current sensorless AO-SMLM developments[29–33] focus on iteratively introducing mirror changes and then evaluating the changes with image-quality metrics. While intensity or sharpness metrics may work robustly for confocal[34], two-photon[35] structured illumination microscopy[36–38] and stimulated emission depletion microscopy[39], it is difficult to design an image-quality metric that summarizes aberration-related information from a single-molecule blinking frame, while ignoring irrelevant variations, such as intensity, background and molecule positions. In addition to these iterative methods requiring many cycles, including image acquisition and mirror changes, to reach the optimal correction, the optimal metric design varies with structures[36]. Previous methods for metric-based AO in SMLM provide robust corrections for tissue-induced aberrations only when the target tissue structures are planar or very thin (Extended Data Fig. 1). This is because emission patterns from single molecules at different axial positions result in inconsistent and, in some cases, even opposite metric responses and thus fundamentally

limit the efficacy of these approaches for aberration correction in tissues (Supplementary Note 1).

Bypassing the previous iterative trial-then-evaluate processes, we developed deep learning-driven adaptive optics (DL-AO) for SMLM to allow direct inference of wavefront distortion and near real-time compensation. Our trained deep neural network (DNN) monitors the individual emission patterns from single-molecule experiments, infers their shared wavefront distortion, feeds the estimates through a dynamic filter (Kalman) and drives a deformable mirror to compensate sample-induced aberrations. The method, referred to as DL-AO for single-molecule imaging, simultaneously estimates and compensates 28 types of wavefront deformation shapes, restores single-molecule emission patterns approaching the system optimum and improves the precision and fidelity of three-dimensional (3D) SMLM through thick brain tissue over 130 μm, with as few as 3–20 mirror changes.

## Results

### Design of deep learning-driven adaptive optics

Single-molecule emission patterns generated by individual fluorescence molecules carry information not only about their molecular center positions, but also about the shared wavefront distortion[40]. The random lateral and axial positions of the blinking fluorescent molecules and their limited photons emitted in SMLM experiments make these emission patterns unsuitable for direct wavefront measurement[18]. A single-molecule deep neural network (smNet)[41] was demonstrated in its capacity to infer wavefront distortions from individual PSFs in simulation and its responsiveness in experimental datasets. Moving from the inference task to active control of a deformable mirror driven by deep learning is, however, nontrivial. Here, we describe our developments in experimental wavefront-based training, stacked estimation networks and stabilized feedback controls through a Kalman filter (Fig. 1) built to allow a robust control and adaptive element correcting 28 aberration modes in near real-time during SMLM imaging, in the presence of complex wavefront distortions, including the distortion induced by refractive index mismatch. Simultaneously compensating a large number of aberration types also enables the capacity of DL-AO in autonomous control of the deformable mirror in response to random and dynamic aberration changes.

Upon detection of SMLM frames, single-molecule-containing subregions are segmented and sent to the network (Supplementary Note 2). Each input subregion goes through a sequence of template matching processes, which are organized as convolutional layers[42,43]

and residual blocks[44] with PReLU activations[45] and batch normalizations[46] in between, then 'fully connects' through $1 \times 1$ convolutional layers to an output vector of 28 values—amplitude estimates for wavefront shapes in terms of the native mirror deformation modes[47] (hereafter referred to as mirror modes). Representing wavefront with coefficients of orthogonal basis helps cut down on the number of outputs and network parameters to be optimized in training. Forming this orthogonal basis directly from native mirror deformations further ensured the coefficients' accuracy in representing mirror responses. With this consideration, the conversion from mirror modes to Zernike polynomials[48]—commonly used as the analytical basis to describe aberrations—is dropped to minimize mismatches between mirror responses and Zernike-based wavefront shapes (Supplementary Note 3). The residual differences between theoretical expectations and experimental mirror deformations (Fig. SS3) are incorporated into training data generation.

To build an accurate link between experimentally detected emission patterns and the mirror control with neural networks, it is imperative to train the network with data that match those obtained experimentally. However, experimental training data of single molecules are challenging to obtain, because the ground-truth wavefronts are usually unknown and the extensive variations of the intensity, background and the lateral and axial locations of single emitters, are impractical to cover experimentally. To this end, we simulate wavefront distortions by linearly combining the mirror deformations obtained experimentally in the SMLM system (Supplementary Note 4). We then use the coefficients of these experimental patterns to form the output of the network. The static residue of system aberration after optimizing the microscope system is also incorporated as the baseline of the wavefront shapes. This allows us to efficiently generate millions of training PSFs based on experimentally measured wavefronts with highly accurate training ground truth (Supplementary Note 4, Supplementary Fig. 1 and Extended Data Figs. 2 and 3; 3D-normalized cross-correlation (NCC) value of >0.95, comparing measured PSFs with those generated from network estimation).

Compensating wavefront distortions inferred from PSFs of blinking molecules, we found that the network proposed mirror change fluctuates with non-vanishing uncertainty before/after each mirror update. This uncertainty increases with the network training range, resulting in a trade-off between the compensation range and stability (Fig. SS3). To this end, we drive the deformable mirror by dynamically switching three networks trained with different aberration scales where the transitions between networks are based on the inference uncertainty (Supplementary Note 2). To stabilize network transitions, we used a Kalman filter[49] (Supplementary Notes 2 and 5) to reduce the estimation uncertainty by recursively combining wavefront measurements before and after each correction. Due to the uncontrollable availability of single-molecule emission patterns with a high signal-to-background ratio and the evolving PSFs after each correction (Extended Data Figs. 4–6, Supplementary Fig. 2 and

Fig. SS3), this process weighs heavily on high-precision measurements against the uncertain ones to ensure stable feedback from the network.

## Deep learning-driven adaptive optics characterization

First, we characterized the response accuracy of DL-AO network using controlled wavefront distortions generated by the deformable mirror. These wavefront distortions resulted in aberrated emission patterns, which were then collected and sent to DL-AO network (Methods). By comparing the induced deformation amplitudes with those estimated by DL-AO, we observed that DL-AO network responded toward individual mirror deformations mostly in a one-to-one manner. This behavior was consistently observed with both beads samples and blinking single molecules from immunofluorescence-labeled cell specimens (Extended Data Figs. 2 and 4 and Fig. SS4). At the same time, we also observed that DL-AO sensed changes in other mirror modes besides the one actually being changed, an expected behavior considering that mirror modes are coupled experimentally (Supplementary Note 3). Due to such coupling, mapping between the wavefront shape and mirror mode amplitudes is no longer unique; therefore, we further quantified the network response accuracy through wavefront shape errors and PSF similarities. We observed that independent measurements from DL-AO and phase retrieval[18,50] using PSFs of fluorescence beads resulted in nearly identical wavefront shapes with a small difference of $0.13 \pm 0.02$ rad (mean ± s.d., $N = 28$) quantified in root-mean-square wavefront error[48] ($W_{rms}$; Methods and Extended Data Fig. 2). Further, comparing the wavefronts estimated by DL-AO network using single-molecule blinking data (100 PSFs) to those retrieved by phase retrieval from beads, we observed high similarities of $0.83 \pm 0.06$ (mean ± s.d., $N = 28$, NCC) and a small wavefront difference of $0.15 \pm 0.03$ rad (mean ± s.d, $N = 28$) in $W_{rms}$ (Extended Data Fig. 4). We observed similar one-to-one responses to mirror changes in both biplane and astigmatism-based setups (Supplementary Note 7), and in an initial investigation on controlling 50 mirror modes simultaneously with DL-AO (Supplementary Note 8). Besides, for the majority of our introduced distortions below 3 radians in $W_{rms}$, a single mirror update can already reduce the wavefront error by 50% (Fig. 2e,f and Supplementary Fig. 3). Caused by the nonlinear mirror deformation response to control input[51], and the decreased network response amplitudes with the decreasing signal-to-noise level or the increasing network training range (Extended Data Fig. 4 and Fig. SS4), we observed that it usually requires 3–20 mirror updates for full compensation.

DL-AO aims to restore PSFs to the level unmodified by the specimen. To characterize the capacity of DL-AO for PSF restoration, we introduced random wavefront distortions using the deformable mirror and compensated these distortions with DL-AO during SMLM experiments with immunofluorescence-labeled Tom20 in COS-7 cells. Visualizing the raw blinking data during the correction, we found the PSFs became less distorted even after a single compensation, and the mirror shape became stable after ~4 mirror updates (Fig. 2a). Because PSFs from blinking

**Fig. 2 | Characterization of deep learning-driven adaptive optics. a**, Measured feedback flow of DL-AO. **b**, An example of PSFs, pupil phases and mirror mode coefficients before and after compensating artificially induced aberrations with DL-AO. For more examples, see Supplementary Videos 1, 10 and 11. **c**, Comparison between DL-AO and metric-based AO on compensating sample-induced distortion at bottom coverslip surface. Results shown are representative of six trials. **d**, Comparison between DL-AO and metric-based AO on compensating sample-induced distortion at 134 μm from bottom coverslip surface in water-based medium ($n = 1.35$; Methods). Results shown are representative of nine trials. For more examples, see Supplementary Videos 2 and 12. **e**, 15 repeated tests (mean ± s.d.) of DL-AO for compensating aberrations of different levels (in $W_{rms}$) in simulation (128 × 128 pixels, 119 nm pixel size, 13 PSFs on average sampled from Poisson distribution, with axial positions ranging from −1 to 1, generated from uniform distribution, 2,500 photon counts on average generated from exponential distribution, 10 background photon counts in each frame.) **f**, 15 repeated tests (mean ± s.d.) of DL-AO for compensating aberrations in different

levels (in $W_{rms}$) based on blinking frames from immunofluorescence-labeled Tom20 specimen. **g**, 3D NCC between PSFs measured under instrument optimum and those measured after DL-AO or metric-based AO. IMM denotes index mismatched specimens at 134 μm. The x-axis labels with 'i–j' format denote jth repeated tests for compensation at area i. **h**, DL-AO compensates for random and sudden wavefront changes during continuous SMLM acquisition. Images in the top row are the distorted wavefronts introduced during continuous imaging. A dot with a blue circle corresponds to a mirror update that introduces a random wavefront distortion (targeted level of 0.75 rad). The dots without blue circles correspond to mirror updates driven by DNN. The single-molecule blinking frames with random and sudden wavefront changes were continuously acquired for 3 min from the immunofluorescence-labeled Tom20 specimen. See Supplementary Fig. 7 and Supplementary Videos 6 and 7 for more examples. PSFs in **b**–**d** and **g** were measured from 100-nm-diameter crimson beads nearby compensation areas. Scale bars in **b**–**d** and **g** are 3 μm. a.u., arbitrary units.

molecules have limited photons and stochastic positions, making them challenging to quantify, we further verified the PSF shape after correction by axially scanning fluorescence beads nearby the compensation areas. Through phase retrieval, we found DL-AO results share a highly similar and flat wavefront shape with the instrument optimum (Methods and Supplementary Note 4), with a residual of 0.29 ± 0.12 rad in $W_{rms}$

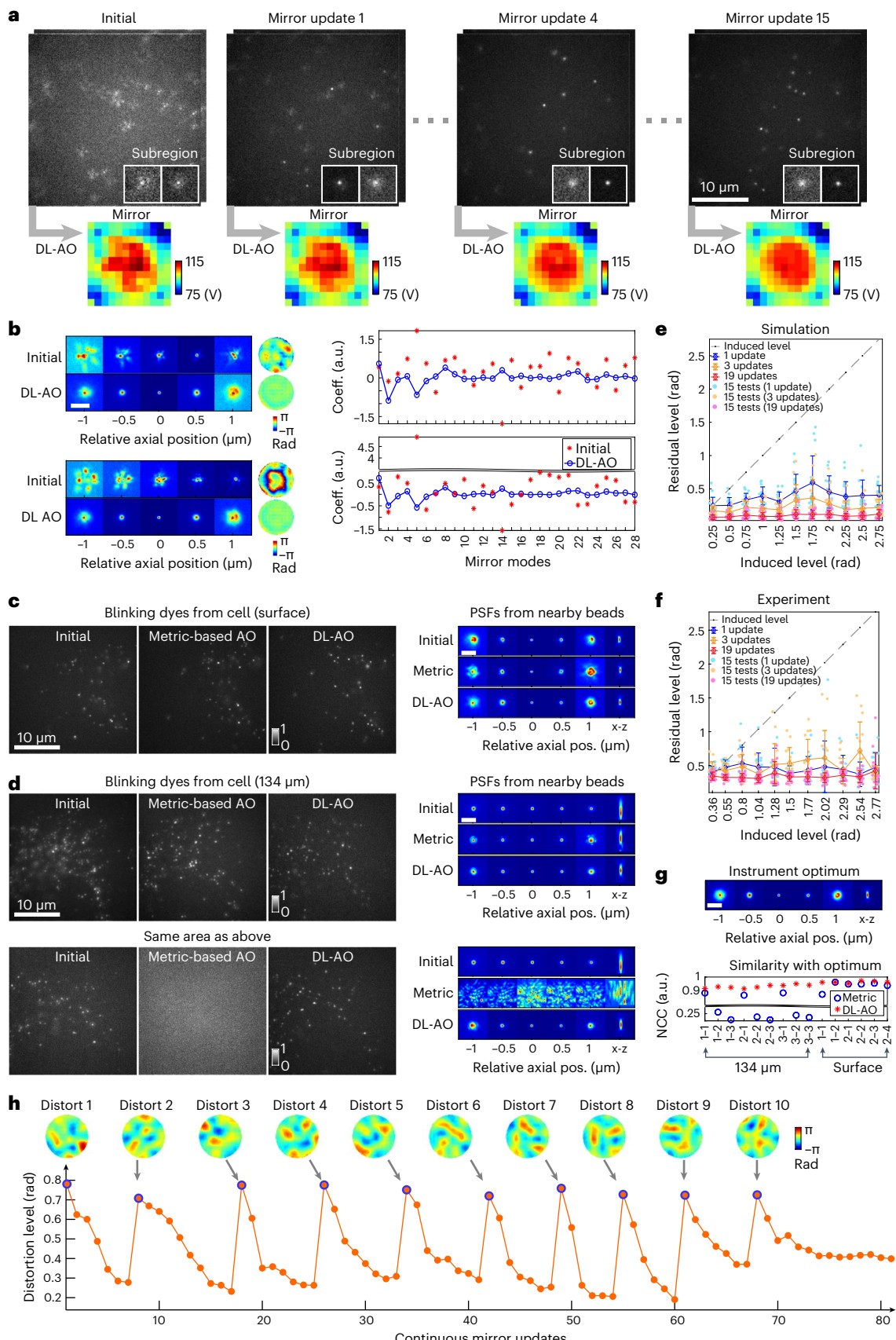

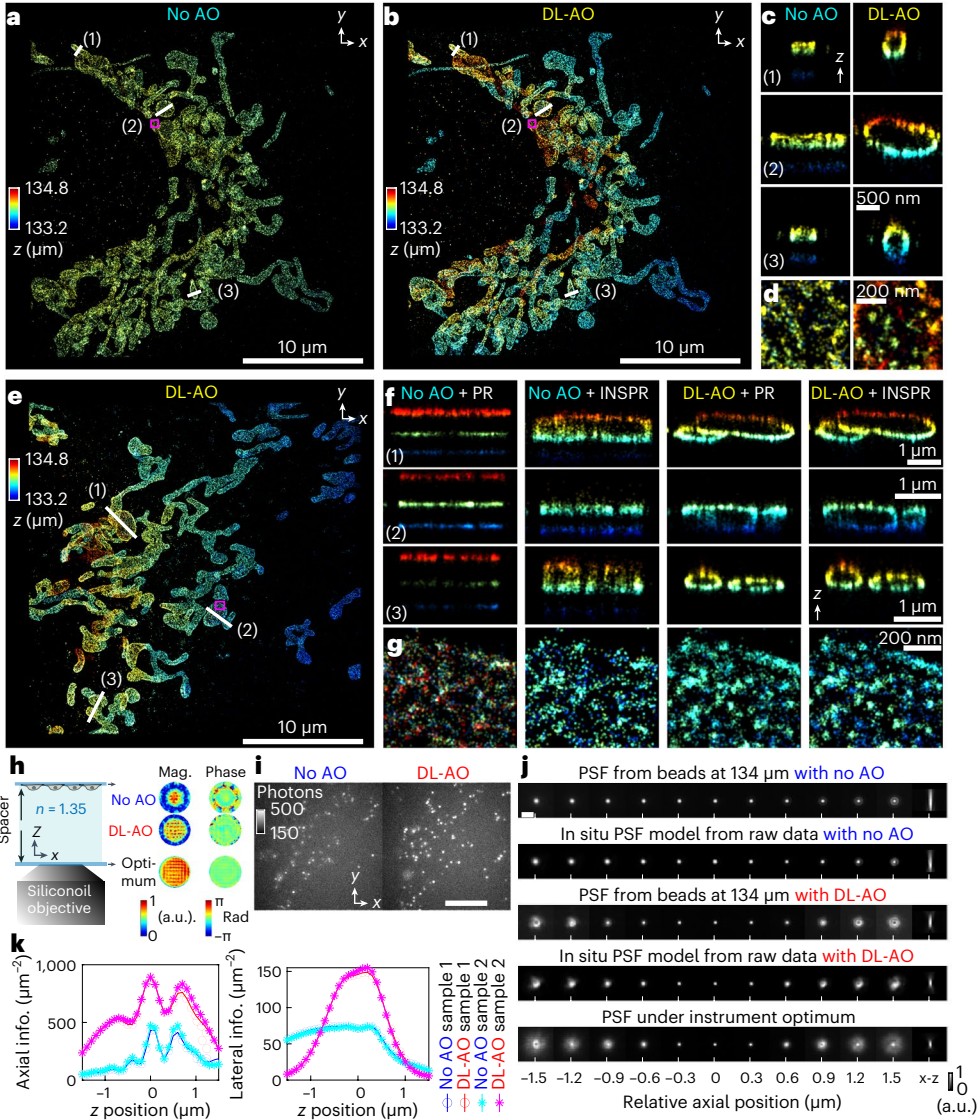

**Fig. 3 | Demonstrations of DL-AO correcting index mismatch-induced aberration by imaging Tom20 proteins in COS-7 cells through 134-μm water-based imaging media.** **a**, 3D SMLM reconstruction of Tom20 imaged through 134-μm water-based media without AO, then reconstructed with an in situ PSF model (INSPR). **b**, 3D SMLM reconstruction of Tom20 imaged through 134-μm water-based media with DL-AO, then reconstructed with INSPR. This depth was chosen based on the spacer we used during sample preparation (Methods). **c**, Axial cross-section of region in **a** and **b** compared without and with DL-AO. **d**, Enlarged regions in **a** and **b** comparing cases without and with DL-AO. **e**, 3D SMLM reconstruction of Tom20 imaged through 134-μm water-based media with DL-AO, then reconstructed with INSPR. **f**, Axial cross-sections in **a** and **b** comparing cases without and with DL-AO combined with reconstruction methods of either in vitro PSF model (PR) or INSPR. The PR PSF model for no-AO case was obtained from 100-nm-diameter crimson beads (referred to as beads hereafter) next to the imaged area. The in vitro model for DL-AO was obtained

from beads at the bottom coverslip surface. **g**, Enlarged regions in **a** and **b** comparing cases without and with DL-AO combined with reconstruction methods of either in vitro PR or INSPR. **h**, Cartoon of the constructed Tom20 specimen and visualization of pupil retrieved from beads at the top (no-AO and DL-AO) and bottom (optimum) surface of the coverslip. **i**, Raw blinking data (after converting the analog-to-digital unit readings in camera frames to the effective photoelectrons, referred as photon number, hereafter) of **a** and **b** compared without and with DL-AO. Scale bar, 10 μm. Results shown are representative of two datasets. **j**, Comparison of measured PSFs at 134 μm without and with DL-AO, in situ PSF models without and with DL-AO and the instrument optimum. Scale bar, 2 μm. **k**, Fisher information content without and with DL-AO was calculated based on PSF model built from beads nearby the imaged area. The values correspond to PSFs with 1,000 total photon counts and 10 background photons per pixel at axial positions of −1.5 μm to 1.5 μm.

(mean ± s.d., $N = 11$; Fig. 2b). Comparing the PSFs after DL-AO and the instrument optimum, high similarities of 0.95 ± 0.02 (mean ± s.d., $N = 11$) were consistently achieved, quantified by 3D NCC (Fig. 2b and Extended Data Fig. 6), and remained 0.96 ± 0.01 (mean ± s.d., $N = 11$ in NCC) for distortion levels from 0.25 to 2.75 radians in $W_{rms}$ (Extended Data Fig. 6). Often, this level of restoration was achieved with only 3–6 mirror updates (Extended Data Fig. 6b), and a single mirror update from DL-AO network reduced the wavefront error by 61.2% ± 24.2% (mean ± s.d, $N = 11$). To drive each mirror update, as few as two subregions containing

isolated single emitters were used for DL-AO network estimation, which spent an average of 0.1 s for forward propagation (Supplementary Table 3, Extended Data Fig. 6 and Supplementary Video 4) and made DL-AO suitable for real-time compensation during SMLM acquisition. Besides, simultaneously controlling a large number of mirror modes with high inference speed makes it possible to compensate aberrations in the presence of dynamic changes from the sample structures (Supplementary Video 8). In this direction, as a proof-of-principle demonstration for dynamic aberration correction, we showed that DL-AO can respond to

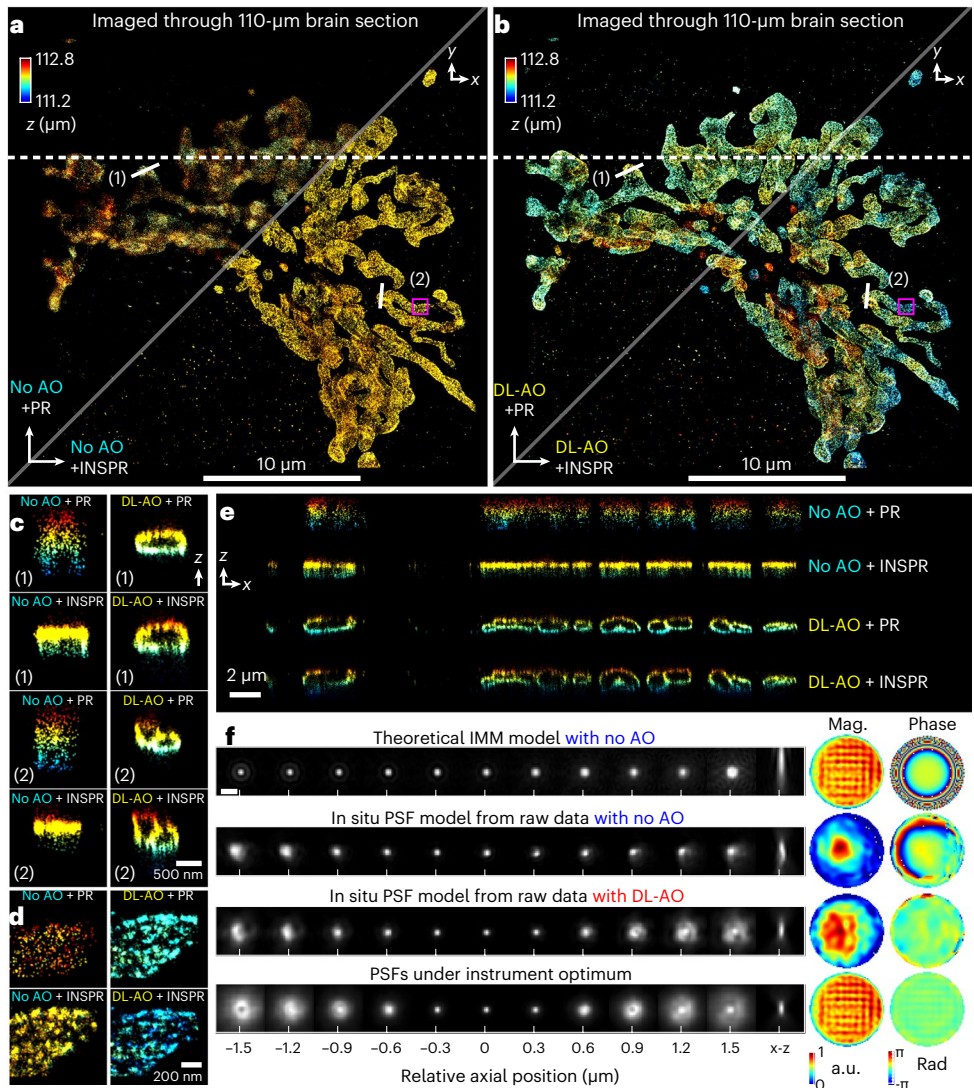

**Fig. 4 | Demonstrations of DL-AO correcting sample-induced aberrations by imaging Tom20 proteins in COS-7 cells through 110-µm unlabeled mouse brain section. a**, 3D SMLM reconstruction of Tom20 proteins imaged through unlabeled tissue without AO, reconstructed with in vitro PSF models: theoretical index mismatch model (PR, upper triangle) and in situ PSF models (INSPR, lower triangle). **b**, Tom20 proteins imaged through unlabeled tissue with DL-AO, reconstructed with in vitro PSF model (PR, upper triangle) and in situ PSF models (INSPR, lower triangle). **c**, Axial cross-sections in **a** and **b** comparing cases without and with DL-AO. **d**, Zoomed-in regions in **a** and **b** comparing cases with and without DL-AO. **e**, Axial cross-sections along the dashed line in **a** and **b**. **f**, Comparisons of PSFs and their pupil functions. The theoretical index mismatch model is based on a measured refractive index of 1.35 for sample media (Methods). Scale bar, 2 µm. Color coding in **a**–**e** indicates axial positions.

sudden wavefront changes and compensating for randomly induced wavefront distortions, while monitoring single-molecule blinking frames continuously in an autonomous manner (Fig. 2h, Supplementary Videos 6 and 7 and Supplementary Fig. 5).

Next, we evaluated the robustness of DL-AO on compensating different levels of wavefront distortion, from 0.25 to 2.75 radians in $W_{rms}$, by assessing the residual wavefront error after correction using both simulation and single-molecule blinking data. After one mirror update, we observed that 51.9% ± 9.3% and 64.3% ± 12.8% (mean ± s.d., $N = 165$) of the induced level was compensated for experimental and simulated data, respectively (Fig. 2e,f). After 19 mirror updates, the residual level was 0.32 ± 0.02 and 0.08 ± 0.03 (mean ± s.d., $N = 165$) radians for experimental and simulated data, respectively (Supplementary Fig. 3). This is a substantial improvement, as compared to existing metric-based methods[29–33], for example, Robust and Effective Adaptive Optics in Localization Microscopy (REALM)[33], which works up to 1 radian at the expense of 10 mirror updates per aberration mode, requiring a total of

330 updates to compensate 11 aberration types (3 rounds)[33]. In addition, metric-based AO is unstable when imaging volumetric cellular structures (Fig. 2c,d,g, Extended Data Fig. 1, Supplementary Figs. 4 and 5 and Supplementary Video 5). A detailed discussion and quantification of these intrinsic limitations of metric-based methods can be found in Supplementary Note 1. We note that when the PSF is in focus, metric-based AO works robustly to compensate aberrations, and thus metric-based AO was used in this work to perform system flattening in obtaining an instrument optimum pupil function for training DL-AO networks.

## Validation through tissue and cell specimens
Inhomogeneous refractive indices within cells and tissues redirect and scatter light. In particular, the mismatches between refractive indices in sample media and objective immersion media reduce the shape modulation of the single-molecule emission patterns axially and broaden the focus laterally (Fig. 2d), increasing the localization uncertainty in all directions and thus worsening the resolution of SMLM.

Such resolution deterioration becomes more drastic with an increasing imaging depth[18].

Here, we demonstrate the capacity of DL-AO in compensating index mismatch-induced aberrations using constructed specimens from 35 µm to 134 µm in thickness with water-based imaging media. Imaging immunofluorescence-labeled Tom20 in COS-7 cells through such thickness without AO correction, the super-resolution images of Tom20 proteins showed nearly no axial distributions (visualized by color differences; Fig. 3a and Extended Data Figs. 8a and 9a), a consequence of the severe lack of shape modulation along the axial direction due to the large imaging depth. While the raw data for both cases in the comparison were acquired in an interleaved manner without and with AO (Methods), DL-AO reconstruction showed the expected outer membrane contours of mitochondria, and without AO the reconstruction displayed notable artifacts (Fig. 3b,c). Zooming in on the lateral dimension, we observed the aggregations of Tom20 proteins, known to form clusters[52], when aberrations were corrected by DL-AO. In comparison, without DL-AO, the lateral reconstruction of Tom20 distribution is diffusive (Fig. 3d,g), as a result of deteriorated lateral resolution through the large imaging depth. These resolution contrasts without and with DL-AO are consistently observed with different samples (Fig. 3e–g and Extended Data Figs. 8 and 9).

Next, we illustrate the mechanism behind such resolution improvement (Fig. 3h–k) by looking at the PSFs and pupil function, which summarizes how the sample together with optical system modulates the collected light, before and after AO. In comparison to the near-uniform distribution of magnitude and phase in the pupil obtained from an in vitro bead, wavefront (phase in the retrieved pupil) showed substantial radial variations and increased phase wrappings at large radial positions (Fig. 3h and Extended Data Figs. 8d and 9d). As a result, the PSFs at different axial positions throughout a 2-µm axial range remained nearly invariant (Fig. 3j). Such loss of PSF shape modulation results in localization artifacts where identical axial positions are falsely assigned to molecules despite their axial distributions. In contrast, DL-AO restored the flatness of the wavefront, resulting in PSFs that are highly similar to the instrument optimum (Fig. 3h,j and Extended Data Figs. 8d and 9d). These improvements in PSF sharpness and modulation explain the resolution improvement after DL-AO (Fig. 3c,d,f,g and Extended Data Figs. 8c and 9c) and were further quantified statistically showing increased Fisher information content per photon upon DL-AO correction (Fig. 3k).

We further demonstrated DL-AO on arbitrary tissue-induced aberrations by imaging through 200-µm-cut unlabeled brain sections resolving membrane of mitochondria using immunofluorescence-labeled Tom20 in COS-7 cells (Fig. 4). Without DL-AO, our observation is consistent with that through water-based cavities where the information of Tom20's axial distribution is lost even with the in situ PSF model (Fig. 4a). Further deterioration was observed both laterally and axially (Fig. 4a,f) using an in vitro PSF model with theoretical index mismatch aberration incorporated. With DL-AO, the 3D reconstruction showed improved resolution, where such improvement could be visualized laterally by the distinct Tom20 protein clusters and axially by the mitochondria membrane contours (Fig. 4b–e).

### Amyloid-β fibrils in 125-µm-cut mouse brain sections

The 3D structures of amyloid-β (Aβ) fibrils are a focus of interest in the studies of Alzheimer's disease and are of particular importance with the success of amyloid-directed therapeutics[53,54]. Visualizing the formation and aggregation of these fibrils within the brain has been limited by the notable resolution loss when imaging through tissues. With DL-AO adaptively optimizing single-molecule emission patterns during SMLM imaging, we can now clearly resolve the organization of immunofluorescence-labeled Aβ fibrils in 125-µm-cut brain sections from 5XFAD mice, a transgenic Alzheimer's disease model that exhibits robust amyloid plaque pathology similar to that found in the human Alzheimer's disease brain[55] (Fig. 5). We imaged Aβ fibrils through these

thick brain tissues without and with DL-AO in an interleaved manner. We observed improved resolution in both axial and lateral directions with DL-AO compared with fibrils imaged without AO (Fig. 5b). Importantly, driven by DL-AO, SMLM reconstruction revealed the 3D organization of individual amyloid fibrils entangling and forming the plaque. However, while without DL-AO, the resolution deteriorates, making the intricate fibril ultrastructure look like blurry clusters (Fig. 5b,c). In addition, inspection of the axially color-coded lateral images and axial cross-section revealed that the fibril structures in the axial direction were distorted and flattened without DL-AO. A similar phenomenon was observed in the presence of spherical aberrations in the previous evaluation of mitochondrial membranes (Figs. 3, 4 and 5b,c). Interestingly, with DL-AO, our reconstructed super-resolution images using in vitro or in situ PSF models revealed highly similar results, suggesting that DL-AO restored the aberrated emission patterns approaching the instrument optimum. Combining DL-AO with INSPR, we imaged fibril structures in different plaque areas (Fig. 5d–l), and we were able to consistently resolve individual fibrils and revealed their 3D arrangements within plaques at various stages (Fig. 5f–l). Measuring the width of Aβ fibrils in tissues, we obtained an averaged width of about 52 ± 9 nm (mean ± s.d., N = 30) and 72 ± 19 nm (mean ± s.d., N = 30) in lateral and axial cross-sections, respectively (Fig. 5j). We note that these measured fibril widths have slight variations among different imaged plaques.

### Dendritic spines in 150–250-µm-cut mouse brain sections

Using DL-AO to correct sample-induced aberrations, and in situ PSF models to perform super-resolution reconstruction post-AO correction, we performed SMLM imaging through 150–250-µm-cut brain sections resolving dendritic spines, the 300–800-nm tiny protrusions from the dendrites whose morphology changes in response to neuronal activities associated with learning and memory[56,57]. Insufficient spatial resolution leads to an erroneous classification of spines[58,59] due to their miniature sizes. The capacity to resolve spines' ultrastructure within their tissue environment is critical in detecting morphological changes in the same area of the functional measurements. This technological advancement will allow electrophysiological and morphological mapping of the same neural circuits linking functional and structural synaptic plasticity with animal behavior[60]. We imaged Thy1-ChR2-EYFP transgenic mice, expressing Channelrhodopsin-2 enhanced yellow fluorescent protein (EYFP) fusion protein in cortical L5 Thy1+ pyramidal cells[61]. Through a 250-µm-cut brain section, we resolved the distinct membrane distribution of the fluorescently tagged target decorating the dendritic spines (Fig. 6 and Extended Data Fig. 10). Throughout the resolved volume of spines, we could observe the membrane-bounded structures as hollow tubes and blobs (Fig. 6d). Besides, the very thin neck of spines can be clearly visualized (Fig. 6g and Extended Data Fig. 10), which provides more accurate information about the dimension of spines. We also imaged 150-µm-cut mouse brain sections (Fig. 6c,f), where thinner sections provide a better signal-to-background ratio. Interestingly, we observed a few occurrences where dendrite membranes labeled ChR2-EYFP appeared to be twisted in the final reconstructed images (Fig. 6f), which may represent a type of physical substrate for decreasing gain for synaptic inputs[62,63]. We obtained an average localization precision of 13 nm and 57 nm in lateral and axial dimensions when imaging through the 250-µm-cut brain section, and 11–52 nm (lateral–axial) precision when imaging through the 150-µm-cut brain section. The capacity to resolve and accurately quantify the shape and size of dendritic spines through large tissue depths paves the way to link spine morphology and function and will facilitate studies of learning, memory and brain disorders.

### Discussion

Combing the power of single-molecule DNN with careful designs in network training, feedback and instrument control, we demonstrated that DL-AO optimizes PSFs approaching the instrument optimum during

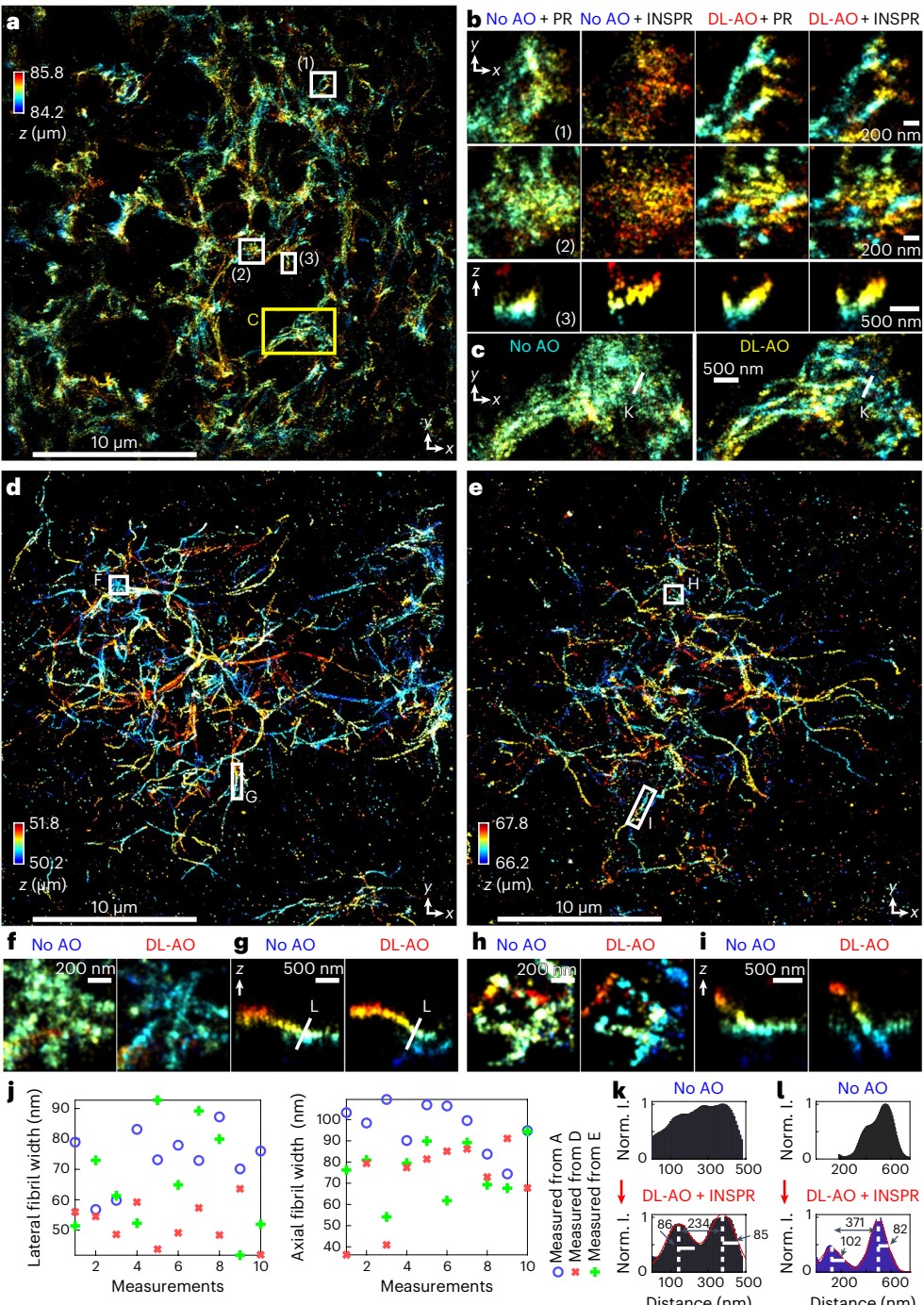

**Fig. 5 | Three-dimensional reconstruction of immunofluorescence-labeled amyloid-β fibrils in 125-μm brain sections of 7.5-month-old 5XFAD female mice. a**, Aβ fibrils imaged using SMLM with DL-AO and reconstructed with in situ PSF model (INSPR) at 85 μm from coverslip surface. Color coding indicates axial positions of single-molecule localizations. **b**, Subregions and cross-sections in **a** showing comparisons of Aβ fibrils imaged without and with DL-AO, reconstructed with either in vitro PSF model (PR) or in situ PSF models (INSPR). **c**, Comparison between fibrils imaged without and with AO, where data without AO were reconstructed using in vitro PR and data with AO used INSPR reconstruction. **d,e**, Aβ fibrils imaged with DL-AO and reconstructed with INSPR at 51 μm and 67 μm from the coverslip surface. **f**, Region in **d** comparing cases without and with DL-AO. **g**, Axial cross-sections in **d** comparing fibrils without and with DL-AO. **h**, Regions in **e** compared cases without and with DL-AO. **i**, Axial cross-sections in **e** comparing cases without and with DL-AO. **j**, Measurements of fibril widths in lateral and axial cross-sections in **a**, **d** and **e**. **k**, Comparison between intensity profiles along the white line in **c** without and with DL-AO. **l**, Comparison between intensity profiles along the white line in **g** without and with DL-AO. 'norm. I.' in **k** and **l** stands for normalized intensity, where intensity in the reconstructed image reflects counts of localized single molecules. The imaged structures were found at depths near the axial limit of tissue thicknesses. Optically measured tissue thicknesses vary among samples, which might be caused by variations in media volume between bottom and top coverslips.

SMLM experiments, and restores the resolution of 3D SMLM through a depth of >130 μm in brain tissues. DL-AO is demonstrated to work robustly in various types of data and specimens, including simulated SMLM frames (Fig. 2e, Supplementary Figs. 3a,b and 5), fluorescence beads (Figs. SS1 and SS2), mitochondrial networks in cells[8,40] (Figs. 3 and 4 and Extended Data Figs. 8 and 9), Aβ plaques[64] in the brains of

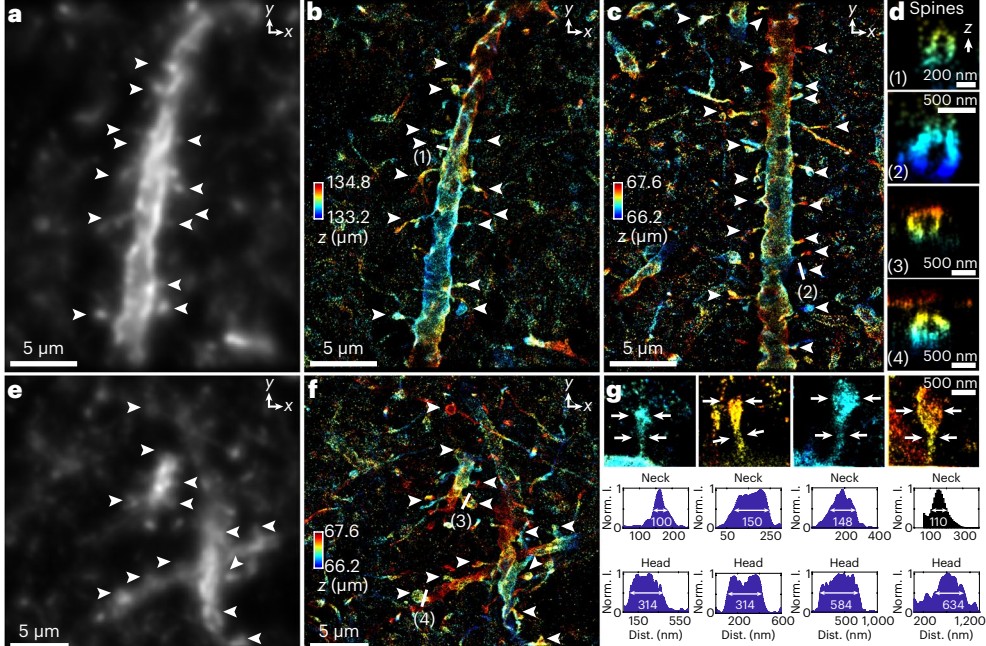

**Fig. 6 | Dendrites and spines in immunofluorescence-labeled Thy1-ChR2-EYFP in 150–250-μm-cut brain sections of 7-week-old mice. a,e,** Diffraction-limited images of Thy1-ChR2-EYFP. Images in **a** and **e** were generated by replacing single-molecule localization points in **b** and **f** with their corresponding PSFs without aberration. **b,** Super-resolution reconstruction of Thy1-ChR2-EYFP using SMLM with DL-AO through a 250-μm-cut brain section. **c,f,** Super-resolution reconstructions of Thy1-ChR2-EYFP using SMLM with DL-AO through 150-μm-cut brain sections. This depth was chosen based on the spacer we used during sample preparation (Methods). **d,** Axial cross-sections identified spines in **b, c** and **f. g,** Identified spines in **b, c** and **f,** and the corresponding size measurements of their necks and heads. 'Norm. I.' stands for normalized intensity, where intensity in the reconstructed image reflects counts of localized single molecules. 'dist.' indicates distance. The histograms show the raw intensity counts along the lines indicated by white arrows in **g.** Sizes are measured at the full widths at the half-maximum intensity. Color coding indicates axial positions. White arrows in **a–f** indicate identified spines. The imaged structures were found at depths near the axial limit of tissue thicknesses. Optically measured tissue thicknesses vary among samples, which might be caused by variations in media volume between bottom and top coverslips. The datasets shown are representative of seven datasets of dendrites with depths of 68–134 μm.

mouse models of Alzheimer' disease (Fig. 5, Supplementary Fig. 6 and Fig. SS23), as well as dendrites and spines[40,65,66] in cortical L5 Thy1+ pyramidal cells in the brains of Thy1-ChR2-EYFP transgenic mice (Fig. 6 and Extended Data Fig. 10). For these data acquired at an imaging depth of 35–134 μm, a lateral resolution of 14–31 nm and a 3D resolution of 41–81 nm on average were measured using decorrelation analysis[67] and Fourier shell correlation[68], respectively (Supplementary Fig. 8). Throughout all these demonstrations, we have kept the DL-AO network parameters unchanged including architecture and training range. The key to the consistent performances despite the distinct sample variations lies in the detection of single molecules because these emission patterns bear no influences from the underlying structures and thus provide a unique and pure source for aberration measurements, invariant across sample types.

However, DL-AO requires at least two isolated and detectable PSFs to start compensation, and this requirement might be challenging to meet when the aberration level or imaging depth is drastically higher than the demonstrated cases where single-molecule emissions are no longer identifiable. In those cases, an initial compensation with the conventional metric-based AO method would serve as a good start while DL-AO provides subsequent and continuous fine aberration corrections for high-resolution single-molecule reconstruction. Because measuring aberrations from single-molecule-containing subregions bears no influences from the underlying sample structures, DL-AO is capable of robustly compensating aberrations despite the dynamic changes in the underlying sample structure (Supplementary Video 8). We further demonstrated that the improved compensation speed (Supplementary Videos 4 and 5) makes DL-AO capable of monitoring and compensating for random and sudden aberration changes (Supplementary Videos 6 and 7). Some of these cellular and tissue structures have been shown

previously in thinner sections or on coverslip surfaces[8,69–71]. Imaging these well-characterized structures helps us in identifying the potential artifact and provides visual assessments of the achievable resolution through the complex tissue and cell environments tested here.

Further, we performed an initial investigation on controlling 50 mirror modes simultaneously with DL-AO (Supplementary Note 8). We observed that DL-AO network responded toward individual mirror deformations mostly in a one-to-one manner, a behavior observed with both beads samples and blinking single molecules from immunofluorescence-labeled cell specimens (Figs. SS19 and 20). We expect that future development in designing training data and neural network architecture will improve the inference accuracy of DL-AO through a large compensation range, ultimately enabling single-shot compensation during SMLM imaging. Additionally, the demonstrated DL-AO applications are limited by the working distance of the silicone oil objective, and thus the imaging depth could potentially be extended when combined with long working distance objectives, if permissible by tissue scattering and fluorescence background. Besides, the current implementation of DL-AO only corrects aberration shared within the field of view, because a deformable mirror is placed at the common pupil plane of the entire FOV. For the residual wavefront differences, analytical methods, such as INSPR[40], can be applied to retrieve region-specific PSF models to localize molecules at different segments of the field of view (Fig. SS22). To compensate field-of-view-dependent aberrations, DL-AO could be potentially combined with the multi-pupil adaptive optics approach[72]. To further improve the achievable resolution and imaging fidelity, we expect that DL-AO can be combined with light-sheet illumination[73,74] for an increased signal-to-background ratio of single-molecule detections, tissue clearing[75] for labeling penetration and reduced aberration level and expansion methods[76] for further

improved spatial resolution, opening doors to observe ultrastructural organizations and colocalizations in tissues and small animals.

Finally, we note a very exciting demonstration in the previous AO-SMLM work by Siemons et al. ('REALM')[33], which demonstrates the possibility of correlating AO-SMLM and functional measurement in brain sections. Although SMLM experiments were performed in fixed tissue, the possibility of accessing tissue nanoscale features in the context of its function illustrated an impactful direction of SMLM in neuroscience. We expect that the demonstrated capacity of DL-AO makes it a central player in connecting our understanding of the brain's ultrastructure and function. SMLM in live tissue, however, has major challenges. Tissue-induced aberration and scattering, the limited temporal resolution, live-tissue compatible probes and its labeling strategy represent barriers in revealing the ultrastructural dynamics in living tissues and animals. DL-AO allows robust compensation of complex wavefront through tissues in near real-time. We believe it represents one solid step toward this grand challenge of live-tissue nanoscopy.

## Online content

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

## Methods

### Preparation of fluorescent beads on coverslips

We cleaned 25-mm-diameter coverslips (CSHP-No1.5-25, Bioscience Tools) successively in ethanol (2701, Decon) and HPLC-grade water (W5-4, Fisher Chemical) three times and then dried them with compressed air. To promote fluorescent bead adhesion on the coverslip, 200 µl of poly-L-lysine solution (P4707, Sigma-Aldrich) was added to one coverslip and incubated for 20 min at room temperature (RT). Following that, the coverslip was rinsed with deionized water. For bead incubation, we first diluted 100-nm-diameter crimson beads (custom-designed, Invitrogen) to 1:1,000,000 in deionized water. Then we added 200 µl of the diluted bead solution to the center of the coverslip and incubated for 20 min at RT. The coverslip was subsequently rinsed with deionized water. The treated coverslip was placed on a custom-made holder[13], and 20 µl of 38% 2,2′-thiodiethanol (166782, Sigma-Aldrich) in 1× PBS (10010023, Gibco) was added to its center. Another 25-mm-diameter coverslip (cleaned using the above protocol) was placed on top of this coverslip. This coverslip sandwich was sealed with two-component silicone dental glue (Twinsil speed 22, Dental-Produktions und Vertriebs).

### Cell culture

COS-7 cells (CRL-1651, American Type Culture Collection (ATCC)) were grown on coverslips placed in six-well plates and cultured in DMEM (30-2002, ATCC) with 10% FBS (30-2020, ATCC) and 1% penicillin–streptomycin (15140122, Gibco) at 37 °C with 5% $CO_2$. The cells were passaged when their confluence reached 80%. The cells were fixed when their confluence reached about 30%.

### Fixation and labeling of Tom20 in COS-7 cells

Cultured cells were first fixed with 37 °C pre-warmed 3% formaldehyde aqueous solution (Formalin) diluted in 1× PBS from 16% formalin (15710, Electron Microscopy Sciences (EMS)) and 0.5% glutaraldehyde aqueous solution (diluted in 1× PBS from 8% glutaraldehyde aqueous solution, 16019, EMS), with gently rocking at RT for 15 min. After fixation, cells were rinsed twice with 1× PBS and then quenched for 7 min with freshly prepared 0.1% sodium borohydride (452882, Sigma-Aldrich) in 1× PBS. The cells were rinsed three times with 1× PBS and blocked with 3% BSA (001-000-162, Jackson ImmunoResearch) and 0.2% Triton X-100 in 1× PBS, with gently rocking at RT for 1 h. After blocking, the cells were incubated at 4 °C overnight with primary antibody (sc-11415, Santa Cruz Biotechnology), diluted at 1:500 in antibody dilution buffer (1% BSA and 0.2% Triton X-100 in 1× PBS). We then washed cells three times for 5 min each time in 0.05% Triton X-100 in 1× PBS, and incubated cells at RT for 5 h with goat anti-rabbit IgG (H + L), Alexa Fluor 647-conjugated secondary antibody (A21245, Invitrogen), diluted at 1:500 in antibody dilution buffer (1% BSA and 0.2% Triton X-100 in 1× PBS). After being washed three times with 5 min each time in 0.05% Triton X-100 in 1× PBS, cells were post-fixed with 4% formalin (diluted at 1:4 with 1× PBS from 16% formalin, 15710, EMS) at RT for 10 min. Cells were then rinsed three times with 1× PBS and stored in 1× PBS at 4 °C.

### Fixation and labeling of amyloid-β in mouse brain sections

The 5xFAD Alzheimer's disease mouse model was used for immunostaining Aβ. Mice were maintained on the C57BL/6J (B6) background (strain 000664), which were purchased from the Jackson Laboratory (JAX MMRRC, 034848). The 5xFAD transgenic mice overexpress five familial Alzheimer's disease (FAD) mutations under control of the Thy1 promoter: the *APP* (695) transgene containing the Swedish (p.Lys670Asn, p.Met671Leu), Florida (p.Ile716Val) and London (p.Val717Ile) mutations, and the *PSEN1* transgene containing the p.Met146Leu and p.Leu286Val *FAD* mutations[32].

Up to five mice were housed per cage with SaniChip bedding and LabDiet 5K52/5K67 (6% fat) feed, with 40–60% humidity at 20–26 °C. The colony room was kept on a 12:12-h light–dark schedule with the lights on from 7:00 to 19:00 daily. The mice were bred and housed in specific-pathogen-free conditions.

Female mice were euthanized by perfusion with ice-cold PBS following full anesthetization with Avertin (125–250 mg per kg body weight intraperitoneal injection)[77]. Animals used in the study were housed in the Stark Neurosciences Research Institute Laboratory Animal Resource Center, Indiana University School of Medicine. All animals were maintained and experiments performed in accordance with the recommendations in the Guide for the Care and Use of Laboratory Animals of the National Institutes of Health. The protocol was approved by the Institutional Animal Care and Use Committee at Indiana University School of Medicine.

Perfused brains from mice at 7.5 months of age were fixed in 4% formalin (1:4 dilution with 1× PBS from 16% formalin, 15710, EMS) for 24 h at 4 °C. Following fixation, brains were cryoprotected in 30% sucrose at 4 °C, and then cut into sections of 150 µm by a vibratome (7000smz-2, Campden Instruments). For immunostaining, free-floating sections were washed and permeabilized with 0.1% Triton X-100 in 1× PBS (PBST), and antigen retrieval was subsequently performed using 1× Reveal Decloaker (Biocare Medical) at 85 °C for 10 min. Sections were blocked in 5% normal donkey serum (D9663, Sigma-Aldrich) in PBST for 1 h at RT. The sections were then incubated with Aβ antibody (Cell Signaling Technology, 2454, rabbit) at a 1:1,000 dilution in 5% normal donkey serum in PBST at 4 °C overnight. Sections were washed and stained for 1 h at RT with donkey anti-rabbit IgG (H + L), Alexa Fluor 647-conjugated secondary antibody (A31573, Invitrogen) diluted at 1:1,000 in 5% normal donkey serum in PBST[78].

### Fixation and labeling of Thy1+ pyramid cells in mouse brain sections

To obtain mice expressing the proper amount of ChR2-EYFP in Thy1+ pyramidal cells, the litters of Thy1-ChR2-EYFP (B6.Cg-Tg (Thy1-COP4/EYFP)18Gfng/J, Jackson Laboratory) mice crossed with B6 (C57BL/6, Jackson Lab) mice were used for the labeling (mouse strain 000664; mouse species: *Mus musculus*). The humidity for mouse housing is 44%, and the temperature is 22 °C. The colony room was kept on a 12/12-h light–dark cycle with the light on from 6:00 to 18:00 daily.

To extract the brains for sectioning, the litters of 7-week-old mice were first anesthetized by intraperitoneal injections of a mix of 90 mg per kg body weight ketamine (59399-114-10, Akron) and 10 mg per kg body weight xylazine (343750, HVS). After confirmation of deep anesthesia, the abdomen was open to expose the diaphragm. The chest cavity was then opened by cutting through the diaphragm and ribs to expose the heart. The trans-cardiac perfusion was performed by inserting a needle into the left ventricle and a small incision into the right atrium. Mice were perfused with 1× PBS (1:10 dilution from DSP32060, Dot Scientific). After the liver was pale, mice were continuously perfused with 4% formalin (1:8 dilution with 1× PBS from 32% formalin, 15714, EMS) to pre-fix the brain until the muscle turned stiff. Brains were carefully collected and post-fixed with 4% formalin at 4 °C overnight. The fixed brains were trimmed for coronal slicing. The trimmed brains were fixed and cut into sections of 150 µm, 200 µm and 250 µm by a vibratome (1000 Plus, TPI Vibratome).

The brain sections were washed three times, for 15 min each time, in wash buffer (0.1% Triton X-100 in 1× PBS) with a gentle shake (120 r.p.m., Orbi-Shaker, Benchmark), and then were incubated in blocking butter (5% BSA (A9647, Sigma-Aldrich) in 1× PBS) for 1.5 h with a gentle shake. The blocked brain sections were incubated with chicken anti-GFP antibody (ab13970, Abcam; diluted to 1:1,000 in blocking buffer) at 4 °C overnight. After washing three times in the wash buffer as in the first step, the slices were incubated with goat anti-chicken IgY (H + L), Alexa Fluor 647-conjugated antibody (A21449, Invitrogen; diluted to 1:600 in wash buffer) at RT for 2 h with gentle rocking.

All animals were maintained and experiments performed in accordance with the recommendations in the Guide for the Care and Use of Laboratory Animals of the National Institutes of Health.

The protocol was approved by the Institutional Animal Care and Use Committee at Purdue University.

## Imaging buffer and sample mounting for single-molecule localization microscopy

Immediately before SMLM imaging, the coverslip with specimens was placed on a custom-made holder[13]. Imaging buffer[79] (10% (wt/vol) glucose in 50 mM Tris, 50 mM sodium chloride, 10 mM 2-mercaptoethylamine, 50 mM 2-mercaptoethanol, 2 mM cyclooctatetraene, 2.5 mM protocatechuic acid and 50 nM proto-catechuic dioxygenase, pH 8.0) was added to the coverslip. Then, another cleaned coverslip was placed on top. This coverslip sandwich was sealed with two-component silicone dental glue. Samples with immunofluorescence-labeled cells on the top coverslips were prepared as described below: 200 μl of poly-L-lysine solution was added to the bottom coverslip, incubated for 20 min and subsequently rinsed with deionized water. Then, 20 μl of microsphere suspension (134 μm in diameter, 7640A, Thermo Scientific) was spread around the outer ring area of the coverslip, and incubated at RT until the coverslip was dried. Then, we placed this coverslip with microspheres at the bottom, added 50–80 μl imaging buffer without touching the microspheres, and added the coverslip with cells on top, with the cell-side surface facing down. The refractive indices of sample media and immersion oil were 1.35 and 1.406, respectively, measured by Abbe refractometer (334610, Thermo Scientific).

## Microscope setup

All experimental data were recorded on a custom-designed SMLM setup built around an Olympus IX-73 microscope stand (Olympus America). This system is equipped with a ×100/1.35-NA (numerical aperture) silicone oil-immersion objective lens (UPLSAPO100XS, Olympus America), a PIFOC objective positioner (ND72Z2LAQ, Physik Instrumente), a three-axis piezo nano-positioning system (Nano-LP200, Mad City Labs) and a manual XY stage (MicroStage-LT, Mad City Labs). A continuous-wave laser at a wavelength of 642 nm (2RU-VFL-P-2000-642-B1R, MPB Communications) was coupled with a polarization-maintaining single-mode fiber (PM-S405-XP, Thorlabs) after passing through an acousto-optic tunable filter (AOTFnC-400.650-TN, AA Opto-electronic) for power modulation. The excitation light coming out of the fiber was focused onto the pupil plane of the objective lens after passing through a filter cube holding a quadband dichroic mirror (Di03-R405/488/561/635-t1, Semrock). The emission fluorescence was split with a 50/50 non-polarizing beam splitter (BS016, Thorlabs) mounted on a kinematic base (KB25/M, Thorlabs). The separated fluorescence signals were delivered by two mirrors onto a 90° specialty mirror (47-005, Edmund Optics), passed through a band-pass filter (FF01-731/137-25), and were then projected on an sCMOS camera (Orca-Flash4.0v3, Hamamatsu) with an effective pixel size of 119 nm on the sample plane. The detection planes that received the signals transmitted and reflected by the beam splitter were referred to as plane 1 and plane 2, respectively. The pupil plane of the objective lens was imaged onto a deformable mirror (Multi-3.5, Boston Micromachines). The imaging system was controlled by a custom-written program in LabVIEW (National Instruments).

## Measurement of mirror deformation modes

The experimental mirror deformation modes[47] (Supplementary Note 1) were measured using the fluorescence bead sample described above. We introduced positive and negative (unit amplitude) mirror changes for each mirror deformation mode. For each mirror shape setting, we acquired PSFs at z-positions from −1.5 to 1.5 μm, with a step size of 100 nm, a frame rate of 10 Hz and three frames per z-position. Pupil phase was extracted through a phase retrieval algorithm. To obtain the experimental mirror deformation bases without the influences of instrument-induced or sample-induced aberrations, we calculated

the differences of the retrieved pupil phases between the positive and negative unit changes of mirror modes and divided them by two. The actual distortion level introduced by each experimental mirror mode was quantified through root-mean-square wavefront error[48] (Methods and Supplementary Note 3).

## Measurement of instrument optimum

We define instrument optimum as the status where optical hardware was optimized to limit the inherent system aberrations. To obtain this optimized status, we followed a previously described method[6], where the deformable mirror was adjusted as follows. Starting from the flat voltage map (provided by the manufacturer) of the deformable mirror, 28 mirror modes (Fig. SS6) were applied sequentially. For each mirror mode, 11 different amplitudes were applied while recording the corresponding fluorescence signal from an in-focus 100-nm crimson bead sample. To extract the fluorescence signal from individual beads, the symmetry center of each imaged bead was obtained using the radial symmetry method[80]. Subsequently, a symmetric two-dimensional Gaussian was generated at the symmetry center and was multiplied by the isolated emission pattern from the fluorescence bead, generating a Gaussian-masked image, and then the total intensity of the masked image was calculated to extract the center peak signal of the beads in focus. For each mirror mode, images of the bead were acquired at 11 different mirror mode amplitudes and the corresponding center peak signals of the bead were extracted as described above. The optimal amplitude (that is, the amplitude providing the highest center peak signal from the beads) was determined from a quadratic fit of these 11 signal measurements versus mirror mode amplitudes. After identifying optimal amplitudes for each of the 28 modes, these amplitudes were added to the flat voltage map (provided by the manufacturer), serving as the starting point for another iteration. This iterative process was repeated five times to achieve optimal system aberration correction. PSFs under instrument optimum were measured using the fluorescence beads sample described above. Data were acquired at a series of z-positions from −1.5 to 1.5 μm, with a step size of 100 nm, a frame rate of 10 Hz and three frames per z-position. A phase retrieval algorithm was then performed on the bead stack to obtain the pupil function under instrument optimum. The instrument optimum can be further verified by decomposing the pupil phase into Zernike mode[48] and checking whether the absolute values of the first 64 Zernike coefficients (Wyant order[48]) are smaller than $0.2\ \lambda/2\pi$.

## Calculation of mean square wavefront error

The root-mean-square wavefront error ($W_{rms}$) values were calculated by the root mean square among all pixels within the image of pupil phase angle. $W_{rms}$ values for experimental wavefronts were either calculated using the pupil phase obtained by phase retrieval from fluorescence beads (Fig. SS6) or calculated using the wavefront images composed of a linear combination of experimental mirror deformation modes as estimated by DL-AO (Fig. 2e,f, Extended Data Figs. 2, 4, 5 and 7 and Supplementary Figs. 3, 5 and 7).

## Measurement of network responses to individual mirror deformation modes

The aberrated PSFs for characterizing network responses (Extended Data Figs. 2 and 4 and Supplementary Fig. 1) were measured using either Tom20 specimens or fluorescence bead samples described above. The samples were first excited with the 642-nm laser at a low intensity of ~50 W/cm² to find regions of interest. Then, data containing single-molecule blinking events were collected at a laser intensity of 2–6 kW/cm² and a frame rate of 50 Hz. The aberrated PSFs from the fluorescence bead samples were measured the same way as we measured PSFs under the instrument optimum. A set of PSF measurements was performed under positive and negative unit changes of each mirror

deformation mode. The differences of network output between positive and negative mirror changes were calculated and divided by two giving the final response vector for each mirror deformation mode.

### Single-molecule localization microscopy acquisition with deep learning-driven adaptive optics

In SMLM data acquisition, the fluorescently labeled samples were first excited with a 642-nm laser at a low intensity of ~50 W/cm$^2$ to find a region of interest. Imaging depths of mitochondrial specimens were measured by the differences of PIFOC readings between the apparent focus of the region of interest and the bottom coverslip surface. The imaging depths for immunofluorescence-labeled tissue specimens were measured by the differences of PIFOC readings between apparent focus signals of the region of interest and the fluorescence signal closest to bottom coverslip surface. The optically measured tissue thicknesses vary among samples that contain brain sections of the same machine-cut thickness. This mismatch between machine-cut thickness (for example, 250-um-cut brain sections mentioned above) and optically measured thickness might be caused by variations in media volume between bottom and top coverslips. Before SMLM experiments, bright-field images of this region were recorded over an axial range from −1 to 1 µm with a step size of 100 nm as reference images for focus stabilization[81]. Then, the blinking data were collected at a laser intensity of 2–6 kW/cm$^2$ and a frame rate of 50 Hz, where the first 3–20 cycles were used for DL-AO, with 20–100 frames per cycle. In the case where high levels of background photons were observed ( 100 per pixel per frame), a temporal median filter was used to estimate structured background for each pixel. This background map was subtracted from each camera frame before the frames were segmented into subregions for DL-AO processing. After DL-AO correction, 2,000 frames were collected per cycle, and 20–120 cycles (50,000–236,000 frames; Supplementary Table 1) were collected for each imaging area. For the interleaved SMLM imaging without and with AO, deformable mirror shape was set to switch between the DL-AO compensated shape and the shape used for instrument optimum (Methods) for each imaging cycle (2,000 frames). Acquisition of no-AO data was performed first in the interleaved sequence for fair comparison. Upon each switch between no-AO and DL-AO acquisitions, the PIFOC objective positioner was moved to compensate apparent focal shift in the case of index mismatch-induced aberration[82]. The focal shifts were determined by an estimated linear relationship between the apparent focus shift and the amplitudes of two radially symmetric mirror deformation modes. The shifts per unit amplitude changes were empirically estimated to be −0.3 µm for mirror mode 5 and −0.2 µm for mirror mode 15 (Fig. SS6). Here, a negative movement of the PIFOC objective positioner corresponds to shifting the imaging plane closer to the bottom coverslip surface.

### Structure size quantification in the reconstructed images

The neck sizes of dendritic spines are measured as follows. First, we selected a profile line at the location where measurement was to be made. A rectangular box was then cropped along the line, with its width ranging from 50 to 500 nm (depending on the spine neck length and the number of localizations). The localization result inside this rectangular box was isolated and rendered into an image with a 3-nm pixel size. Each point in the rendered image was blurred with a Gaussian kernel of 3 pixels in width. Intensity profile was generated along the profile line by sum projection and subsequently the histogram was normalized by dividing its maximum value. The spine neck/head sizes were calculated by the full width at the half maximum of the intensity histogram. The widths of the Aβ fibrils were measured the same way as that for the spine necks, except for a Gaussian function was used to fit the line profile ('fit', Curve Fitting Toolbox 2020a, MATLAB R2020a, MathWorks), with Gaussian function switched between 'gauss1' (single Gaussian fit) and 'gauss2' (two Gaussians) depending on the number of peaks observed in the intensity histogram. The half width at the half maximum of the fitted Gaussian curve was treated as the width of each fibril.

### Reporting summary

Further information on research design is available in the Nature Portfolio Reporting Summary linked to this article.

### Data availability

The results of molecular localizations for cell/tissue structures are available in Figshare at https://doi.org/10.6084/m9.figshare.23823438. Example training and testing data for DL-AO are available in Supplementary Software packages. Complete training and testing datasets can be generated through the shared codes. Other data that support the findings of this study are available from the corresponding authors upon request.

### Code availability

PyTorch scripts for training, validating and testing DL-AO and MATLAB codes for generating single-molecule training datasets are available as Supplementary Software, and further updates will be made available in the GitHub repository. We also include a Jupyter Notebook in the Supplementary Software for demonstrating one-time wavefront inference using a trained DL-AO network. The code in this notebook is executable in a web browser with Google Colaboratory, which provides free access to essential computing hardware such as GPUs or TPUs used in DL-AO. Custom code used in this work is available at https://github.com/HuanglabPurdue/DL-AO.

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

### Acknowledgements

We thank Fan Xu (Beijing Institute of Technology) for suggestions on PSF segmentation and the super-resolution reconstruction process. We thank Sheng Liu (The University of New Mexico) for suggestions on the phase retrieval algorithm and the PSF generation process. We thank Yue Zheng (Purdue University) for suggestions on the manuscript. This work was supported by the US National Institutes of Health (grants GM119785 to F.H., MH123401 to F.H. and A.A.C. and RF1AG074566 to G.E.L.).

### Author contributions

P.Z. and F.H. conceived the project and designed the experiments for DL-AO characterization. P.Z. developed the DL-AO workflow, wrote the DL-AO instrument control, performed experiments and analyzed the data. D.M. developed the microscope setup. P.Z., D.M. and H.-C.G.

performed and optimized deformable mirror calibration. X.C. and A.P.T. optimized the staining procedure for tissue specimens. P.Z., X.C., A.P.T., Y.T., L.F., C.B., A.A.C. and F.H. designed the experiments and prepared biological samples. G.E.L., A.A.C. and F.H. supervised the study. All authors wrote the paper.

## Competing interests

P.Z. and F.H. are inventors on a patent application submitted by Purdue University that covers the basic principles of DL-AO. All other authors have no competing interests.

## Additional information

**Extended data** is available for this paper at https://doi.org/10.1038/s41592-023-02029-0.

**Correspondence and requests for materials** should be addressed to Gary E. Landreth, Alexander A. Chubykin or Fang Huang.

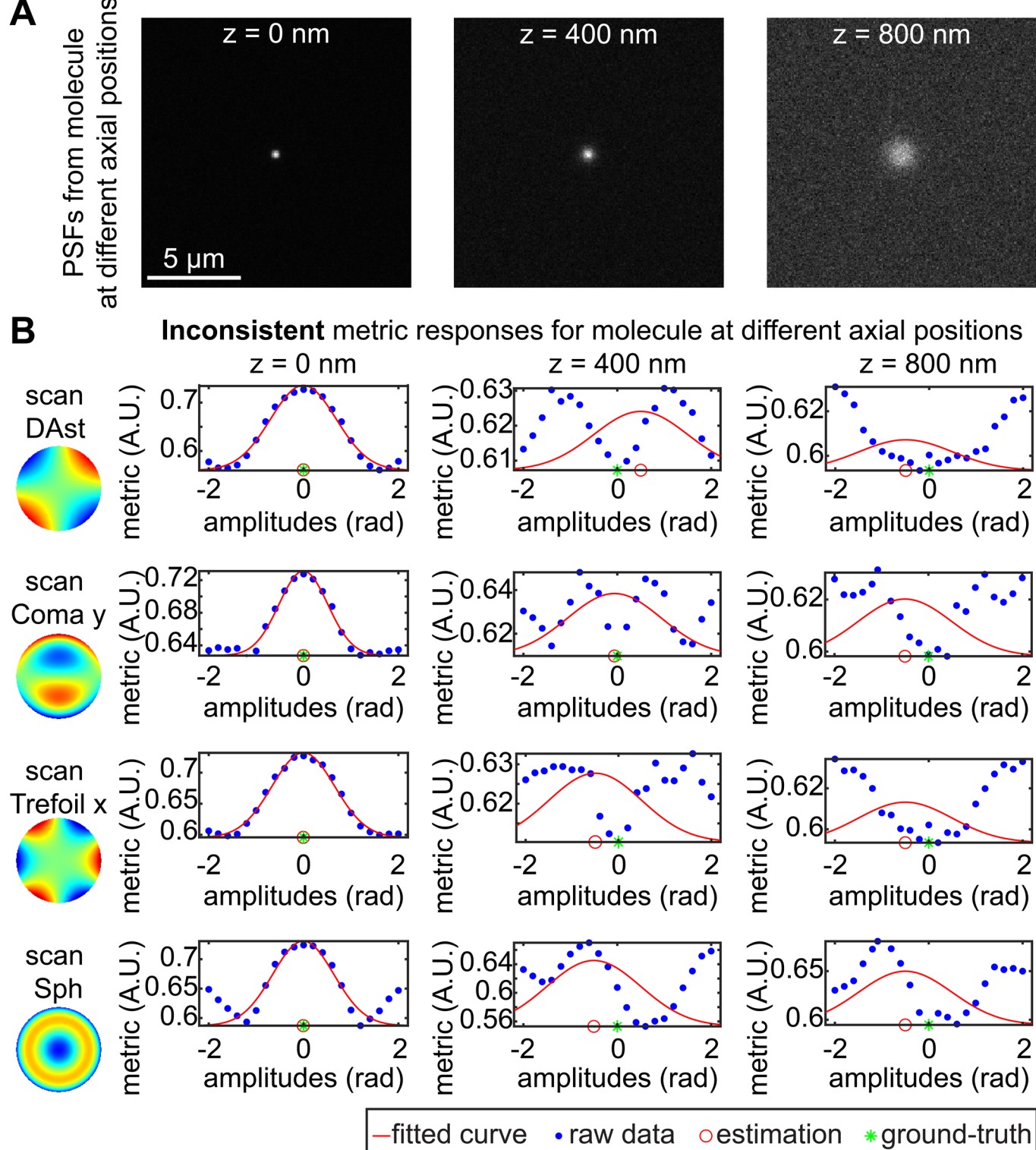

**Extended Data Fig. 1 | Inconsistent responses in metric-based AO.**
(**A**) Simulated emission patterns from single molecules at different axial positions. (**B**) Metric values vs. amplitudes of mirror shapes. Each metric value was calculated through a weighted sum of the Fourier transform of an acquisition under certain mirror shape, the weighting factor of which is defined in previous work[32,33]. The acquisitions were simulated as images with 400 × 400 pixels and

65 nm pixel size. Each acquisition contains one PSF with 10000 photon counts and 10 background photon counts. Each row shows the changes in metric values when scanning the amplitudes of a mirror shape shown on the left. Different columns show metrics calculated from PSFs at different axial positions. 'DAst' and 'Sph' stand for Diagonal Astigmatism and Primary Spherical in Zernike polynomials[48].

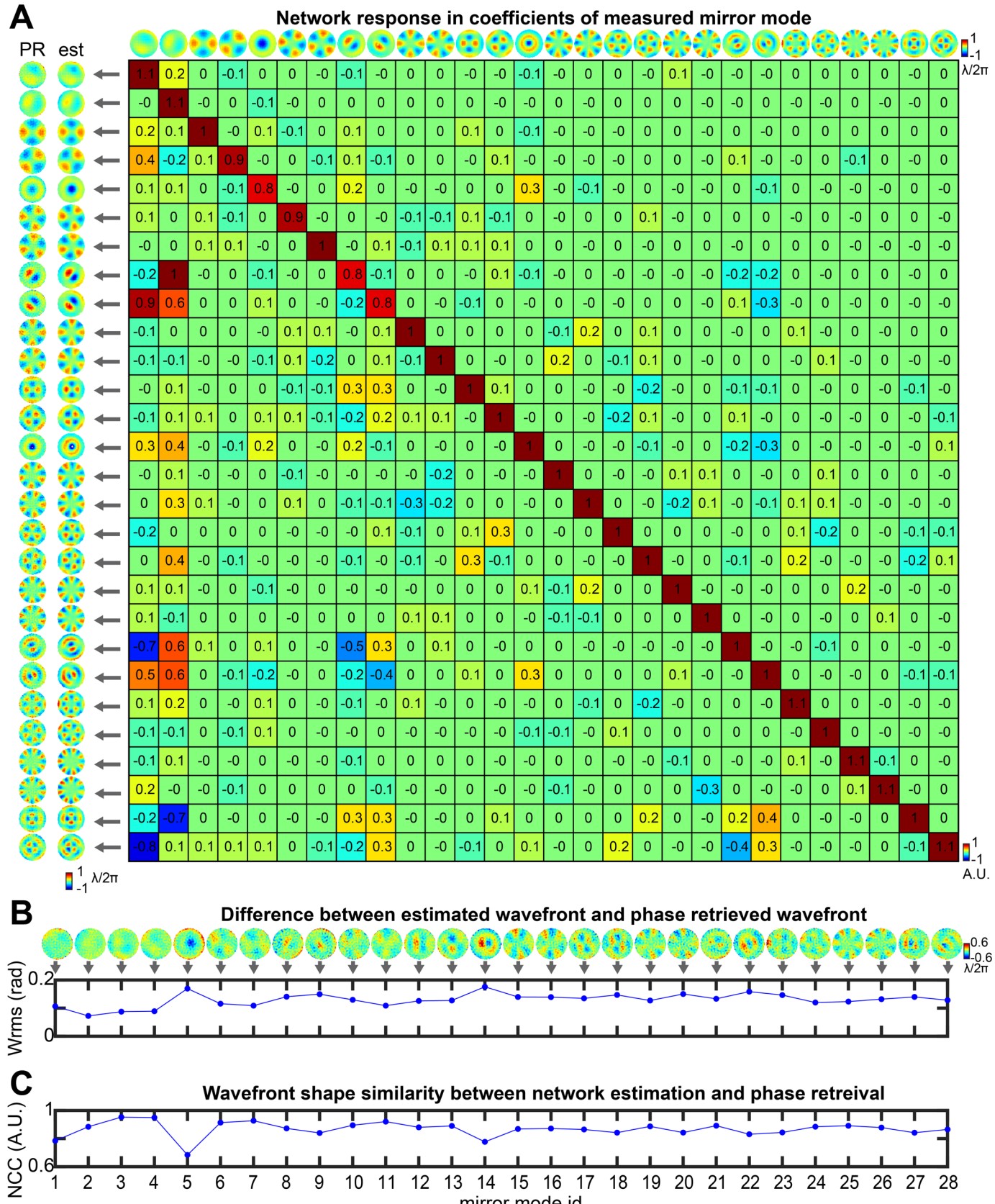

Extended Data Fig. 2 | See next page for caption.

**Extended Data Fig. 2 | Characterizing neural network responses to mirror mode changes using PSFs measured from fluorescent beads. (A)** Network response to individual mirror mode changes. Each row of the response matrix shows the network responded mirror coefficients under a unit change of each mirror deformation mode. After linear combining measured mirror modes (images below the title) with network responded coefficients, we obtained network estimated wavefront shape w.r.t. individual mirror mode changes (the 2nd column). The 1st column shows phase retrieved wavefronts from beads imaged individual mirror mode changes. The PSFs were measured with 100-nm-diameter crimson beads. PSFs from −1.5 μm to 1.5 μm around the focus, with 0.1 μm step size, were collected for characterizing network responses. **(B)** Difference between network estimated wavefront and phase retrieved wavefront (the first two columns in A). The top row shows the pixel-wise differences between wavefronts obtained from network estimation and that obtained from phase retrieval. The plot below shows the root mean square wavefront error[48] ($W_{rms}$, Methods) of each wavefront difference. **(C)** Similarity between network estimated wavefront and phase retrieved wavefront. The similarity is quantified with 2D normalized cross correlation (NCC).

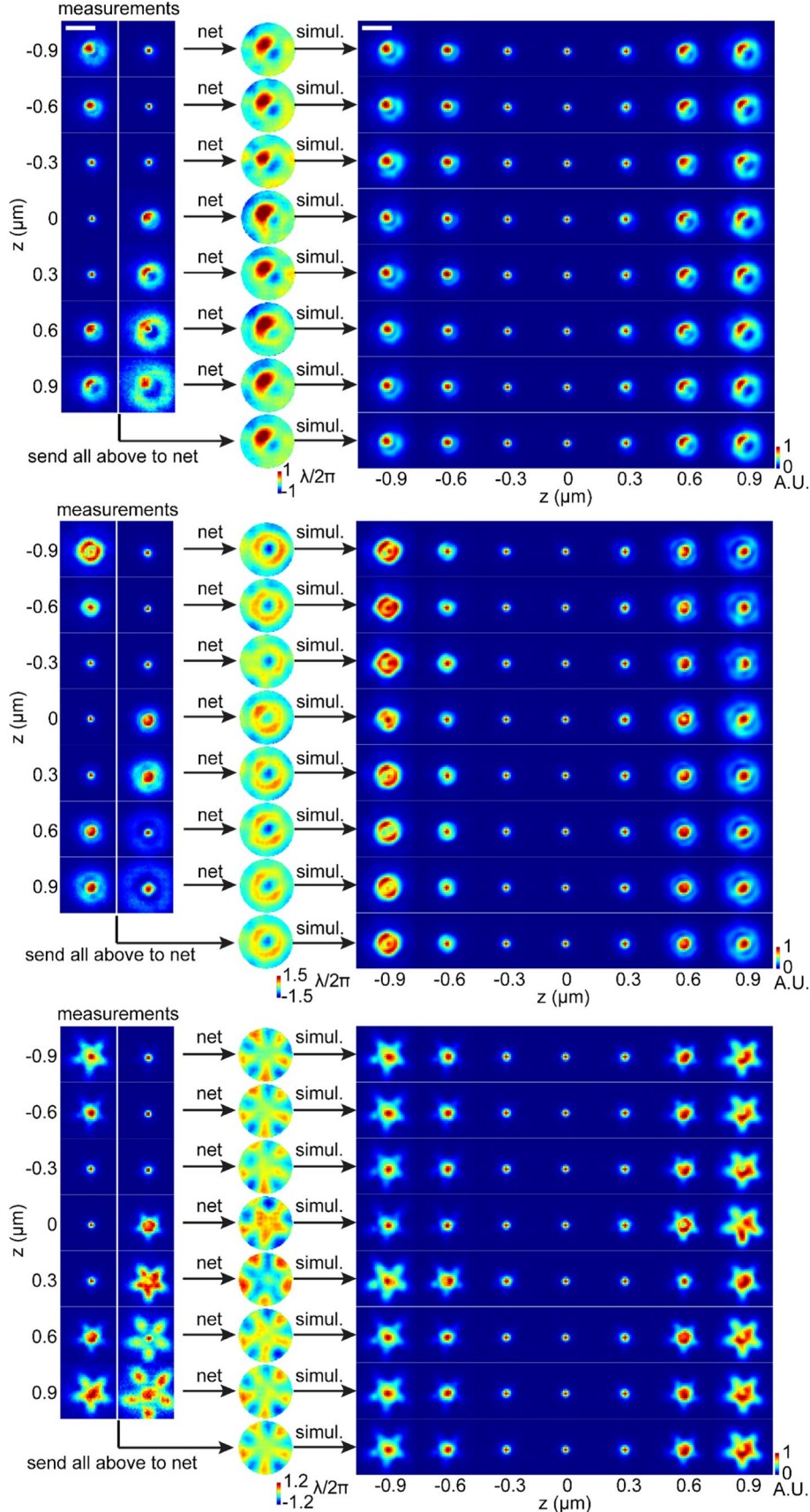

**Extended Data Fig. 3 | See next page for caption.**

**Extended Data Fig. 3 | Comparison between measured PSFs and PSFs simulated from network estimations based on a single measurement of an isolated molecule.** The left column shows measured PSFs from 100-nm-diameter crimson beads when scanning Piezo stage at different axial positions. The measured PSFs in biplane sub-regions were sent to neural network, which outputs a vector of mirror mode coefficients for each sub-region. The measured mirror modes (Supplementary Note 3) were linear combined with mirror mode coefficients output from network, which result in wavefronts shown in the middle column. The wavefront for network estimations based on all measured PSFs were generated with an averaged value among network outputs w.r.t. PSFs at different axial positions. The wavefront was then used to simulate PSFs at different axial positions to check the similarity between measured PSFs and PSFs simulated from network estimations. PSFs were simulated without background and noise for visualization. Scale bars: 2 μm.

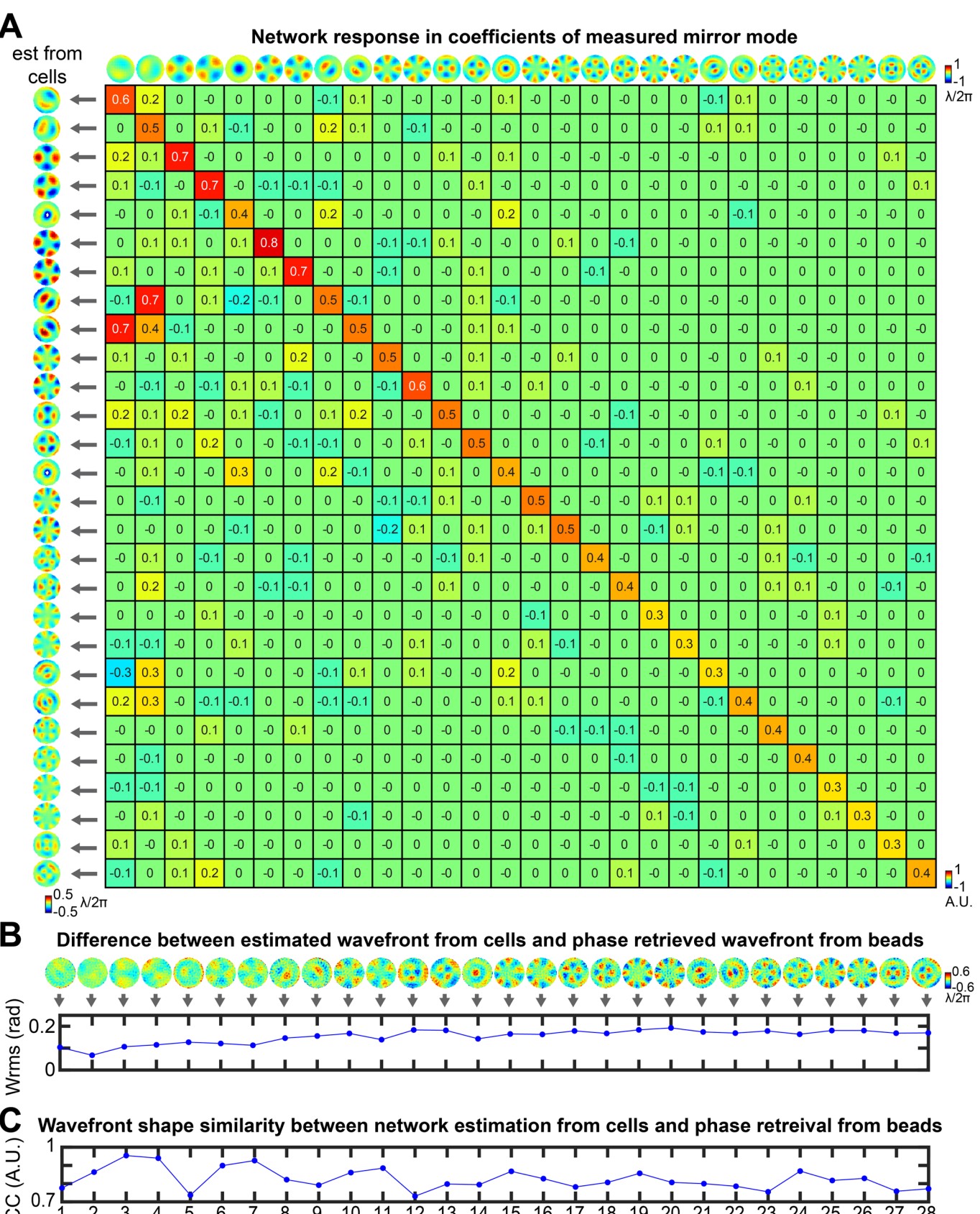

**A** Network response in coefficients of measured mirror mode

est from cells

**B** Difference between estimated wavefront from cells and phase retrieved wavefront from beads

**C** Wavefront shape similarity between network estimation from cells and phase retreival from beads

**Extended Data Fig. 4 | See next page for caption.**

**Extended Data Fig. 4 | Characterizing neural network responses to mirror mode changes using PSFs measured from blinking molecules.** (**A**) Network response to individual mirror mode changes. Each row of the response matrix shows the network responded mirror coefficients under a unit change of each mirror deformation mode. After linear combining measured mirror modes (images below the title) with network responded coefficients, we obtained network estimated wavefront shape w.r.t. individual mirror mode changes. The PSFs were measured experimental blinking frames from immune-fluorescence-labeled Tom20 specimen. 100 PSFs were used for calculating each network response. (**B**) Difference between network estimated wavefront (left column in A) and phase retrieved from beads (left column in Extended Data Fig. 2A). The top row shows the pixel-wise differences between wavefronts obtained from network estimation and that obtained from phase retrieval. The plot below shows the root mean square wavefront error[48] ($W_{rms}$, Methods) of each wavefront difference. (**C**) Similarity between network estimated wavefront and phase retrieved wavefront. The similarity is quantified with 2D normalized cross correlation (NCC).

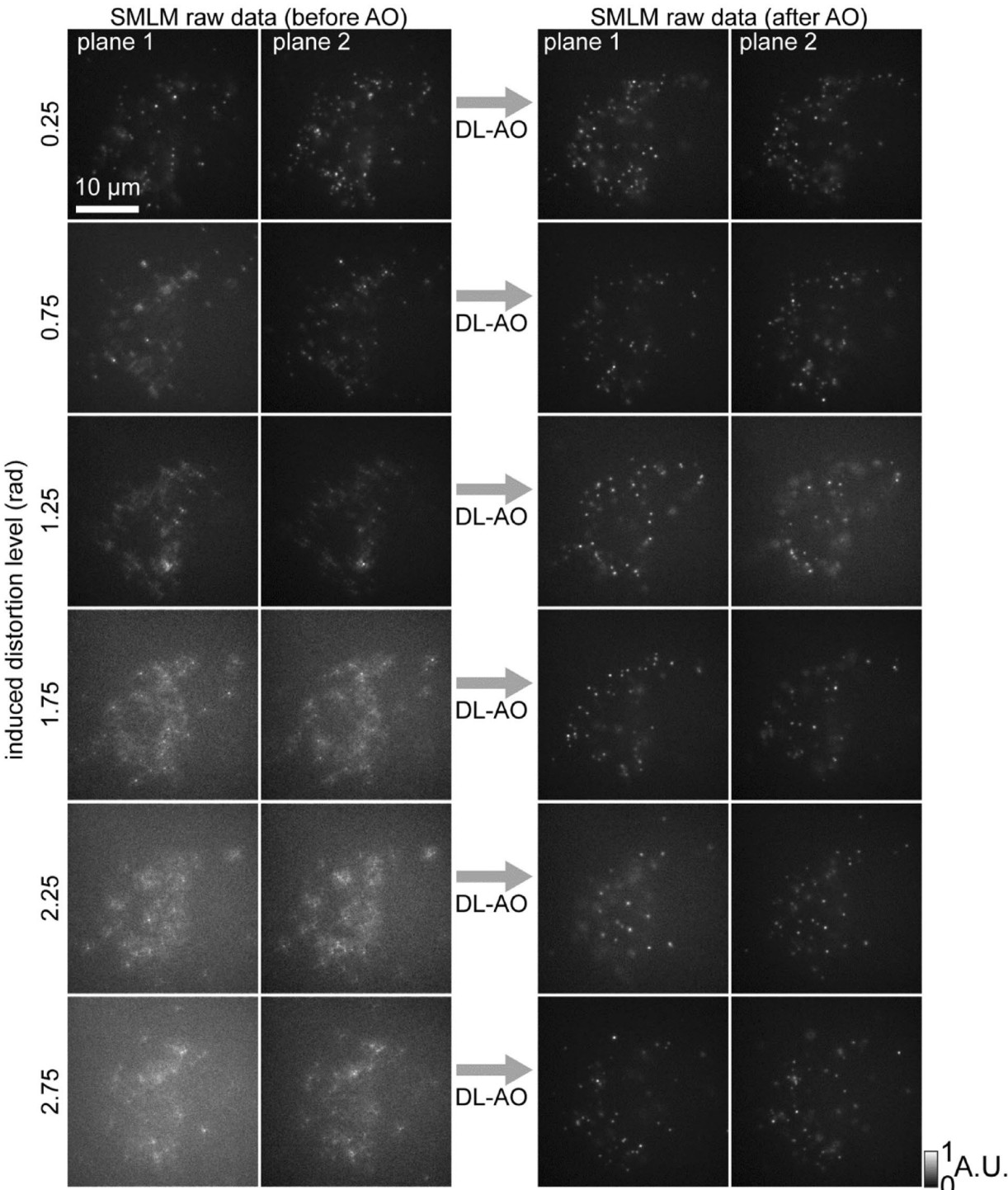

**Extended Data Fig. 5 | SMLM frames before and after DL-AO compensating various amount of induced aberrations.** DL-AO was compensating aberrations in different levels (in $W_{rms}$, Methods) based on experimental blinking frames from immune-fluorescence-labeled Tom20 specimen. 20 camera frames were used for DL-AO estimation before each mirror update. The blinking data after DL-AO were compensation results after 19 mirror updates. The results shown are representatives of 15 tests.

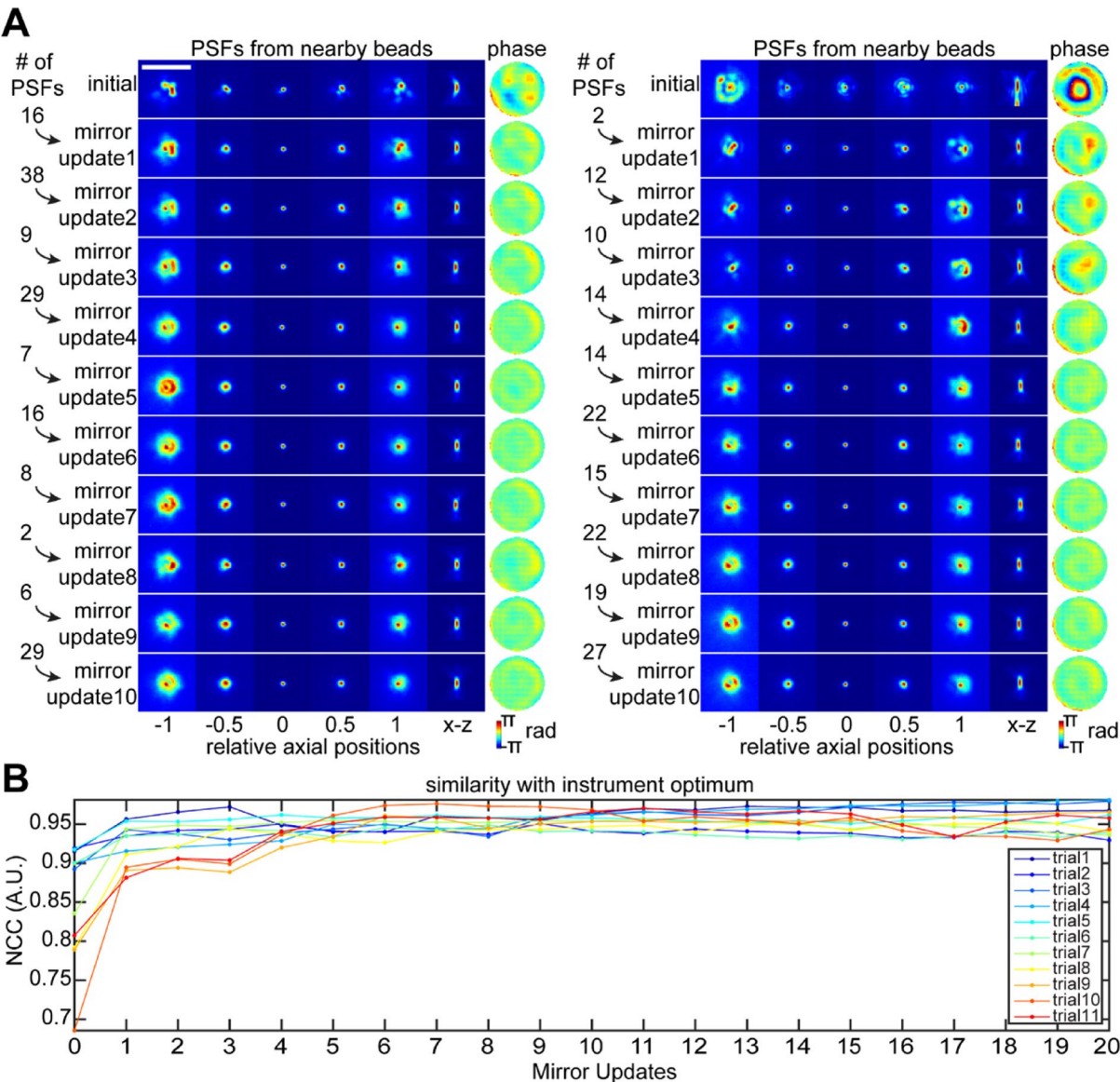

**Extended Data Fig. 6 | PSF shape before and after each mirror update during DL-AO compensation.** (**A**) Examples of PSFs before and after each mirror update, when compensating artificially induced aberrations with DL-AO. Compensations were performed in real time during SMLM experiments. The results shown are representatives of 11 tests. The SMLM blinking frames for compensation were acquired from immune-fluorescence-labeled Tom20 specimen. PSFs were measured from 100-nm-diameter crimson beads near the compensation area post SMLM acquisition. Scale bar: 5 µm. 'PSF #' stands for number of sub-regions used for each DL-AO network estimation. 'phase' stands for pupil phase obtained by phase retrieval on the measured PSFs from beads. Each PSF was normalized to maximum equals to 1. (**B**) Quantitative comparisons between PSFs measured under instrument optimum and those measured before and after each mirror update using 3D normalized cross correlation (NCC).

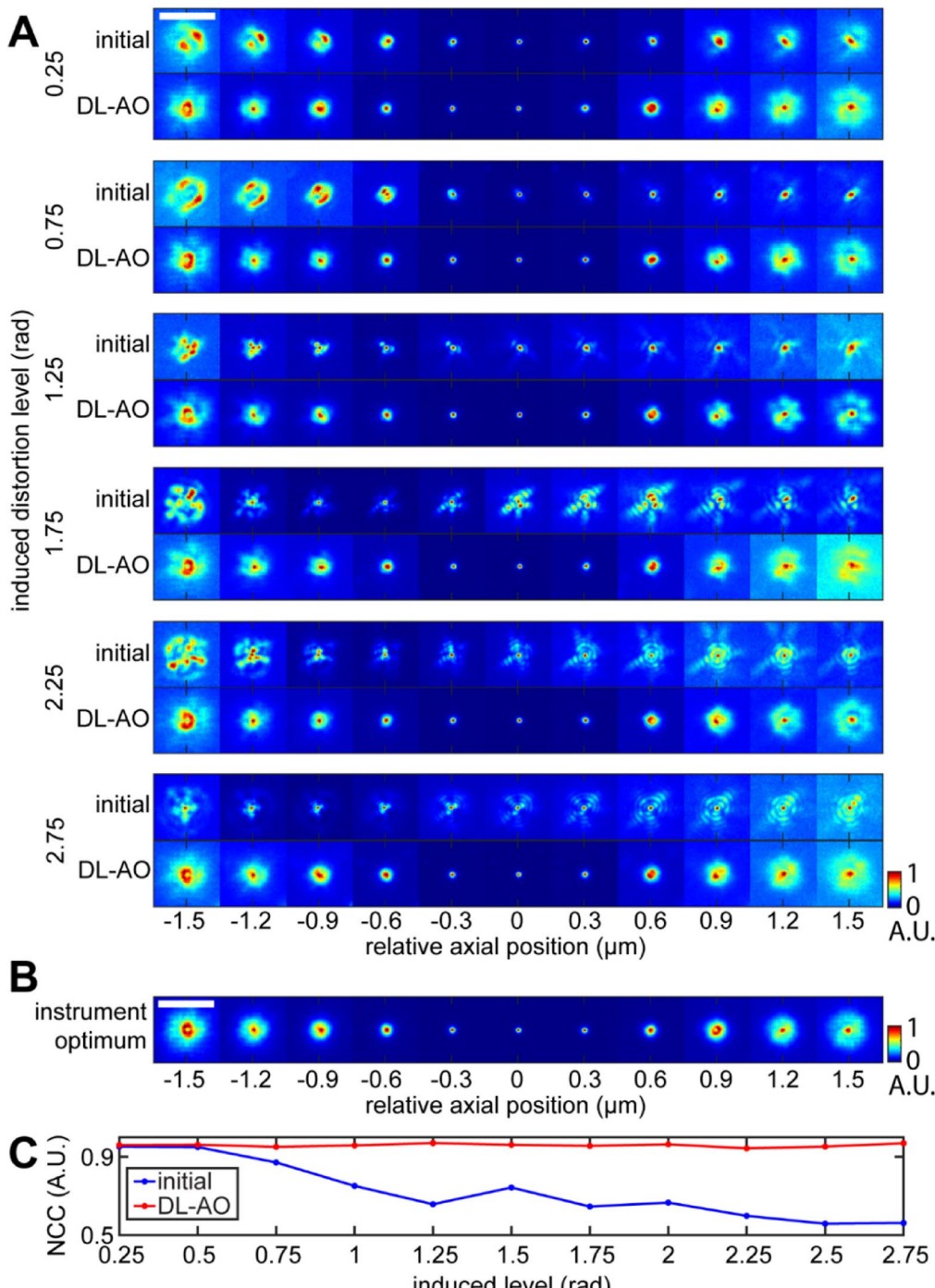

**Extended Data Fig. 7 | PSFs before and after DL-AO at various number of induced aberrations.** (**A**) Examples of PSFs before and after DL-AO, when compensating artificially induced aberrations. Compensations are performed in real time during SMLM experiments shown in Extended Data Fig. 5. PSFs are measured from 100-nm-diameter crimson beads near the compensation area post SMLM acquisition. Scale bar: 5 µm. (**B**) PSFs are measured under instrument optimum (Methods) from 100-nm-diameter crimson beads. Scale bar: 5 µm. (**C**) Quantitative comparisons between PSFs measured under instrument optimum and those measured before and after DL-AO using 3D normalized cross correlation (NCC).

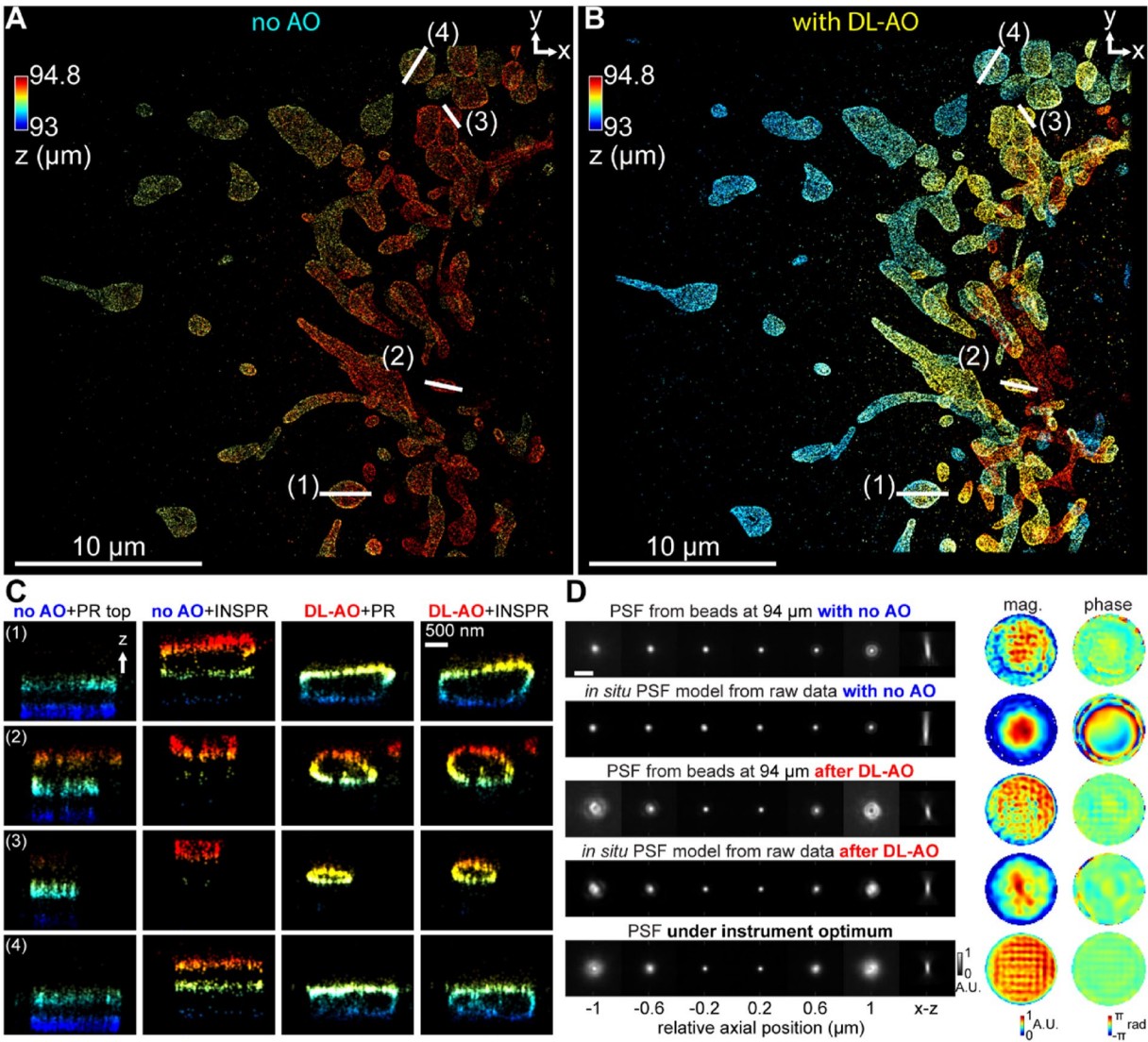

**Extended Data Fig. 8 | Demonstrations of DL-AO correcting index mismatch induced aberration by imaging Tom20 proteins in COS-7 cells through 94 μm water-based imaging media.** (**A**) 3D SMLM reconstruction of Tom20 imaged through 94 μm water-based media without AO, then reconstructed with *in situ* PSF model (INSPR) (**B**) 3D SMLM reconstruction of Tom20 imaged through 94 μm water-based media with DL-AO, then reconstructed with INSPR. (**C**) Axial cross-sections in A and B comparing cases without and with DL-AO combined with reconstruction methods of either *in vitro* PSF model (PR) or *in situ* PSF models (INSPR). The PR PSF model for no AO case was obtained from 100-nm-diameter crimson bead next to the imaged area. The *in vitro* model for DL-AO was obtained from beads at the bottom coverslip surface. Choices of *in vitro* model are made in order to find the closest match with corresponding experimental conditions. (**D**) Comparison of measured PSFs at 94 μm without and with DL-AO, *in situ* PSF models without and with DL-AO, and the instrument optimum. Scale bar: 2 μm.

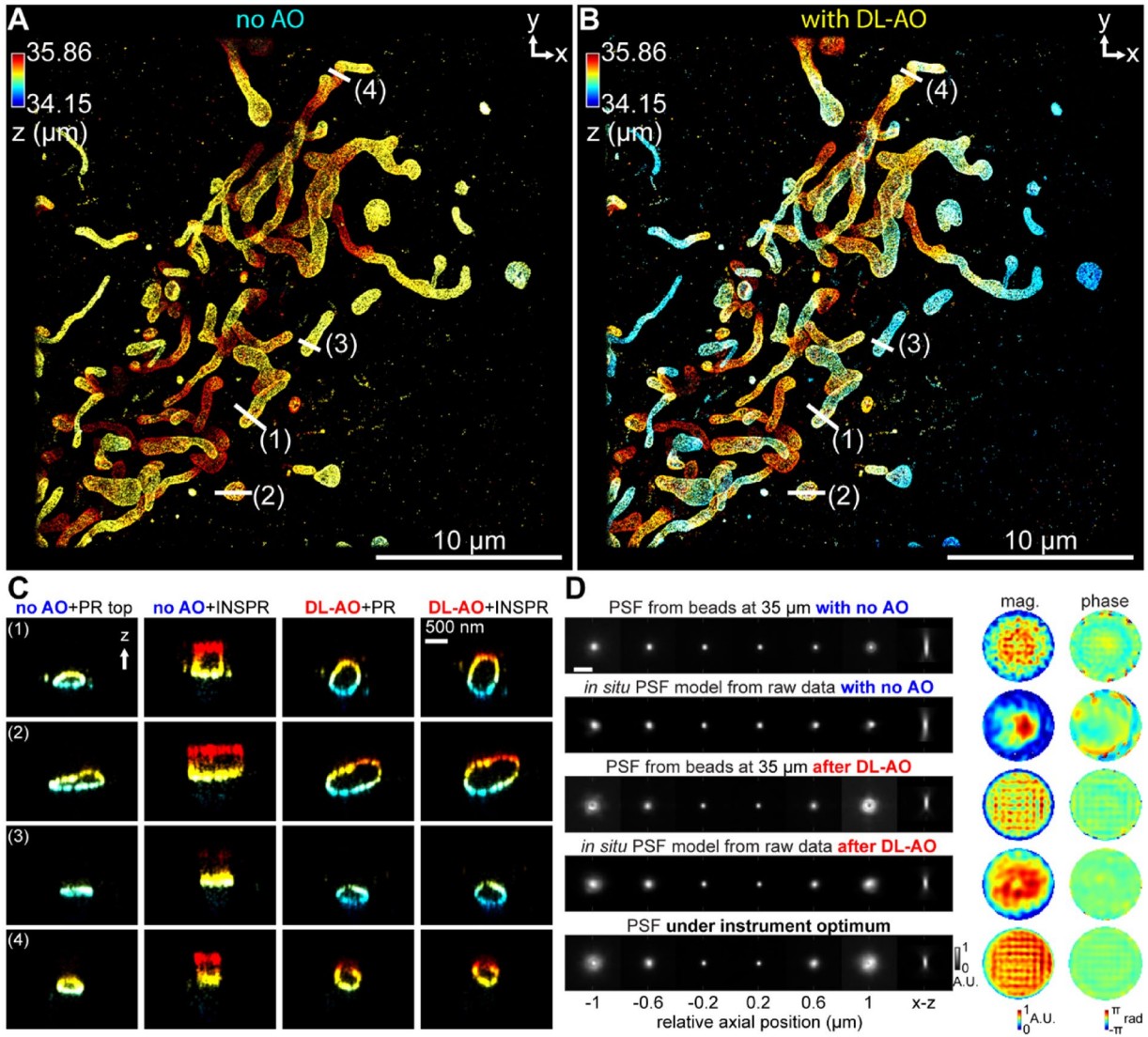

**Extended Data Fig. 9 | Demonstrations of DL-AO correcting index mismatch induced aberration by imaging Tom20 proteins in COS-7 cells through 35 μm water-based imaging media.** (**A**) 3D SMLM reconstruction of Tom20 imaged through 35 μm water-based media without AO, then reconstructed with *in situ* PSF model (INSPR) (**B**) 3D SMLM reconstruction of Tom20 imaged through 35 μm water-based media with DL-AO, then reconstructed with INSPR. (**C**) Axial cross-sections in A and B comparing cases without and with DL-AO combined with reconstruction methods of either *in vitro* PSF model (PR) or *in situ* PSF models (INSPR). The PR PSF model for no AO case was obtained from 100-nm-diameter crimson bead next to the imaged area. The *in vitro* model for DL-AO was obtained from beads at the bottom coverslip surface. Choices of *in vitro* model are made to find the closest match with corresponding experimental conditions. (**D**) Comparison of measured PSFs at 35 μm without and with DL-AO, *in situ* PSF models without and with DL-AO, and the instrument optimum. Scale bar: 2 μm.

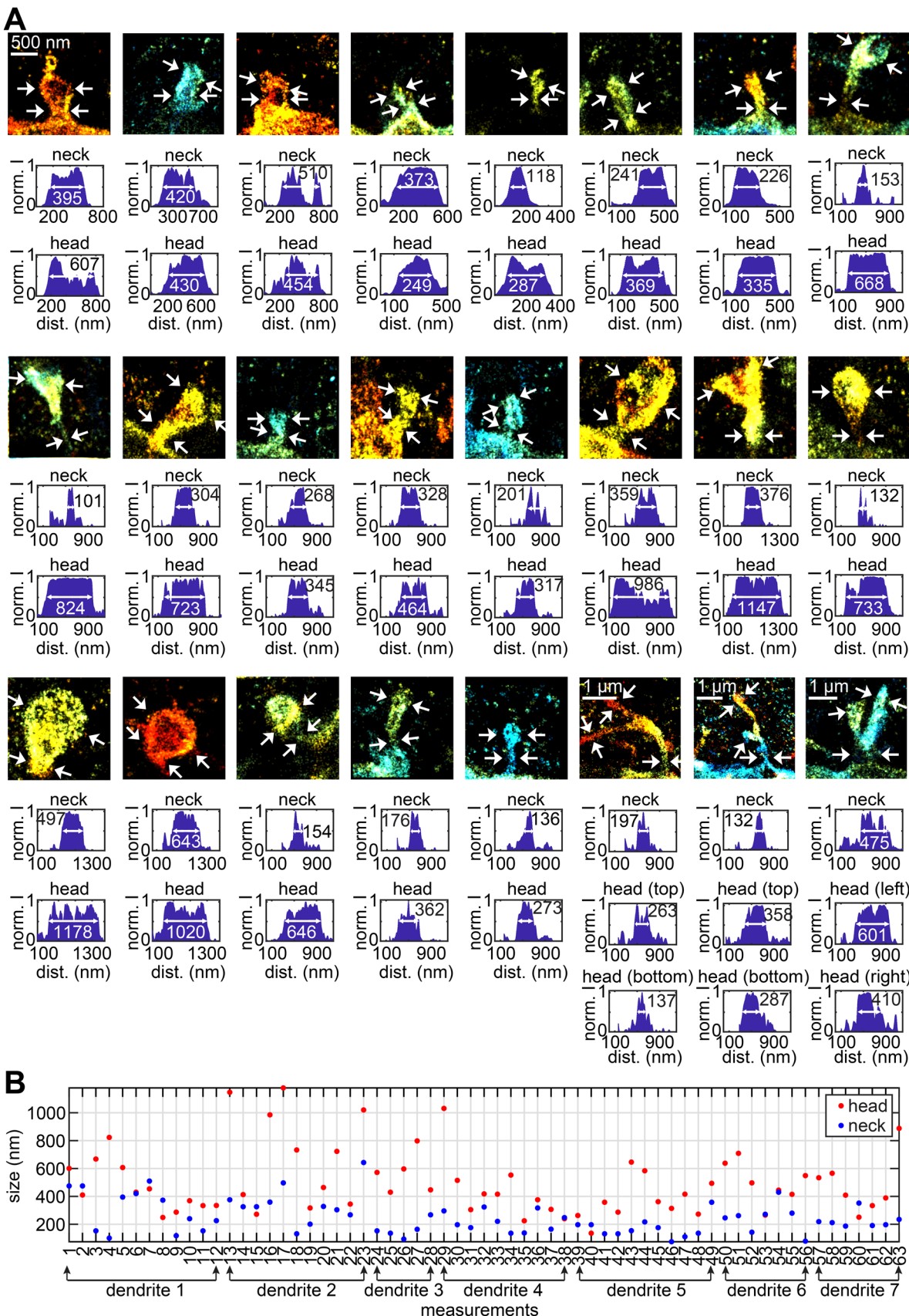

Extended Data Fig. 10 | See next page for caption.

**Extended Data Fig. 10 | Size measurements of spines' heads and necks. (A)** Identified spines in Fig. 6A–C, and the corresponding size measurements of their necks and heads. 'Norm. I.' stands for normalized intensity, where intensity in reconstructed image reflects counts of localized single molecules. 'dist.' stands for distance. The histograms show the raw intensity counts along the lines indicated by white arrows. Sizes are measured at the full widths at the half maximum intensity (Methods). The images of spines share the same scale bar as the first image, unless labeled specifically. **(B)** Size measurements of spines' heads and necks from seven dendrites in immune-fluorescence-labeled Thy1-ChR2-EYFP in 150–250 μm brain sections of 7-week-old mice. The sizes were measured from super-resolution reconstructions of Thy1-ChR2-EYFP using SMLM with DL-AO through 150-μm-cut brain sections (Dendrite 1-3) and 250-μm-cut brain sections (Dendrite 4-7). Dendrite 1-7 were reconstructed at 67 μm, 67 μm, 83 μm, 134 μm, 134 μm, 126 μm, 132 μm from coverslip surface. The imaged structures were found at depths near the axial limit of tissue thicknesses. Optically measured tissue thicknesses vary among samples, which might be caused by variations in media volume between bottom and top coverslips.

|---|---|

# Reporting Summary

## Statistics

For all statistical analyses, confirm that the following items are present in the figure legend, table legend, main text, or Methods section.

| n/a | Confirmed | |
|---|---|---|
| ☐ | ☒ | The exact sample size (*n*) for each experimental group/condition, given as a discrete number and unit of measurement |
| ☐ | ☒ | A statement on whether measurements were taken from distinct samples or whether the same sample was measured repeatedly |
| ☒ | ☐ | The statistical test(s) used AND whether they are one- or two-sided *Only common tests should be described solely by name; describe more complex techniques in the Methods section.* |
| ☒ | ☐ | A description of all covariates tested |
| ☒ | ☐ | A description of any assumptions or corrections, such as tests of normality and adjustment for multiple comparisons |
| ☐ | ☒ | A full description of the statistical parameters including central tendency (e.g. means) or other basic estimates (e.g. regression coefficient) AND variation (e.g. standard deviation) or associated estimates of uncertainty (e.g. confidence intervals) |
| ☒ | ☐ | For null hypothesis testing, the test statistic (e.g. *F*, *t*, *r*) with confidence intervals, effect sizes, degrees of freedom and *P* value noted *Give P values as exact values whenever suitable.* |
| ☒ | ☐ | For Bayesian analysis, information on the choice of priors and Markov chain Monte Carlo settings |
| ☒ | ☐ | For hierarchical and complex designs, identification of the appropriate level for tests and full reporting of outcomes |
| ☒ | ☐ | Estimates of effect sizes (e.g. Cohen's *d*, Pearson's *r*), indicating how they were calculated |

*Our web collection on statistics for biologists contains articles on many of the points above.*

## Software and code

Policy information about availability of computer code

| Data collection | Data was collected on a custom-built SMLM microscope which is described in Online Methods. The microscope is controlled by custom-made controlling software written in LabVIEW 2015 (National Instruments). The program is available from the corresponding author upon request. |
|---|---|
| Data analysis | Data analysis was performed using custom single molecule localization algorithm in MATLAB R2020a (MathWorks) which was shared in 'INSPR toolbox' available at https://github.com/HuanglabPurdue/INSPR. The in situ PSF model was obtained with INSPR toolbox, the detailed parameter of which was described in Supplementary Notes. The in vitro model was obtained using phase retrieval algorithm, which was described in Supplementary Notes and was shared at: https://github.com/HuanglabPurdue/smNet. The training data generation process used in this study was written in MATLAB R2020a (MathWorks), which is shared in supplementary software. The training and testing scripts used in this study was written in Python3.6.9 with Pytorch0.4.0 and CUDA10.1, which is shared in supplementary software with example training and testing data. The supplementary software for reproducing this study as well as its future updates will be available at GitHub: https://github.com/HuanglabPurdue/DL-AO. Additional datasets are available from the corresponding authors upon request. |

For manuscripts utilizing custom algorithms or software that are central to the research but not yet described in published literature, software must be made available to editors and reviewers. We strongly encourage code deposition in a community repository (e.g. GitHub). See the Nature Portfolio guidelines for submitting code & software for further information.

## Data

Policy information about <u>availability of data</u>

All manuscripts must include a <u>data availability statement</u>. This statement should provide the following information, where applicable:

- Accession codes, unique identifiers, or web links for publicly available datasets
- A description of any restrictions on data availability
- For clinical datasets or third party data, please ensure that the statement adheres to our <u>policy</u>

> The results of molecular localizations for cell/tissue structures are available in Figshare, doi: 10.6084/m9.figshare.23823438. Example training and testing data for DL-AO are available in supplementary software packages. Complete training and testing datasets can be generated through the shared codes. Other data that support the findings of this study are available from the corresponding authors upon request.

## Human research participants

Policy information about <u>studies involving human research participants and Sex and Gender in Research.</u>

| | |
|---|---|
| Reporting on sex and gender | N/A |
| Population characteristics | N/A |
| Recruitment | N/A |
| Ethics oversight | N/A |

Note that full information on the approval of the study protocol must also be provided in the manuscript.

# Field-specific reporting

Please select the one below that is the best fit for your research. If you are not sure, read the appropriate sections before making your selection.

☒ Life sciences  ☐ Behavioural & social sciences  ☐ Ecological, evolutionary & environmental sciences

For a reference copy of the document with all sections, see <u>nature.com/documents/nr-reporting-summary-flat.pdf</u>

# Life sciences study design

All studies must disclose on these points even when the disclosure is negative.

| | |
|---|---|
| Sample size | For biological data, the number of sub-regions analyzed is determined by the number of emission events obtained from single molecule switching nanoscopy experiments. Number of localizations per dataset is listed in Supplementary Table 1. For robustness test, the sample size was determined by the number of experiments performed using our technique. The number of repeated experiments was labeled in figures or figure captions. For simulated data, the sample size was determined by the simulated emission patterns of single molecules for different imaging conditions. |
| Data exclusions | Qualitative exclusion criteria for accepting or rejecting imaged samples were pre-established based on comparisons to previously published images and preliminary experiments. Single molecule localizations were statistically tested and rejected/accepted based on their log-likelihood ratio (as goodness of fit metric), theoretical uncertainty, emitted photon as well as their convergence during fitting. |
| Replication | All attempts at replication were successful. The number of replications were provided in the figure legends. |
| Randomization | In this study, there is no allocation of sample into groups. |
| Blinding | In this study, there is no allocation of sample into groups. |

# Reporting for specific materials, systems and methods

We require information from authors about some types of materials, experimental systems and methods used in many studies. Here, indicate whether each material, system or method listed is relevant to your study. If you are not sure if a list item applies to your research, read the appropriate section before selecting a response.

## Materials & experimental systems

| n/a | Involved in the study |
|-----|----------------------|
| ☐ | ☒ Antibodies |
| ☐ | ☒ Eukaryotic cell lines |
| ☒ | ☐ Palaeontology and archaeology |
| ☐ | ☒ Animals and other organisms |
| ☒ | ☐ Clinical data |
| ☒ | ☐ Dual use research of concern |

## Methods

| n/a | Involved in the study |
|-----|----------------------|
| ☒ | ☐ ChIP-seq |
| ☒ | ☐ Flow cytometry |
| ☒ | ☐ MRI-based neuroimaging |

# Antibodies

| | |
|---|---|
| Antibodies used | Primary antibodies:<br>Tom20 antibody (Santa Cruz Biotechnology, Cat#sc-11415, used at 1:500)<br>β-Amyloid antibody (Cell Signaling Technology, Cat#2454, used at 1:1000)<br>Anti-GFP antibody (Abcam, Cat#ab13970, used at 1:1000)<br><br>Secondary antibodies:<br>Goat anti-rabbit IgG (H+L) highly cross-adsorbed secondary antibody, Alexa Fluor 647 (Invitrogen, Cat#A21245, used at 1:500)<br>Donkey anti-rabbit IgG (H+L) highly cross-adsorbed secondary antibody, Alexa Fluor 647 (Invitrogen, Cat#A31573, used at 1:1000)<br>Goat anti-chicken IgY (H+L) secondary antibody, Alexa Fluor 647 (Invitrogen, Cat#A21449, used at 1:600) |
| Validation | All the antibodies used here are commercial products. They have been extensively used by us and others in the past, and have been validated through our previous experiments as well as the manufacturer's own test and validations. Below we list one reference for each antibody:<br>Tom20 antibody - doi: 10.1016/j.cell/2016/06/016<br>β-Amyloid antibody - doi: 10.1038/s41592-018-0053-8<br>Anti-GFP antibody - doi: 10.1038/s41467-018-08146-1<br>Goat anti-rabbit IgG (H+L) highly cross-adsorbed secondary antibody, Alexa Fluor 647 - doi: 10.1016/j.cell/2016/06/016<br>Donkey anti-rabbit IgG (H+L) highly cross-adsorbed secondary antibody, Alexa Fluor 647 - doi: 10.1016/j.cell/2016/06/016<br>Goat anti-chicken IgY (H+L) secondary antibody, Alexa Fluor 647 - doi: 10.1016/j.stem.2016.11.019 |

# Eukaryotic cell lines

Policy information about cell lines and Sex and Gender in Research

| | |
|---|---|
| Cell line source(s) | COS-7 cells (CRL-1651 from ATCC) |
| Authentication | Cell line was purchased from ATCC and was not independently authenticated. |
| Mycoplasma contamination | Not applicable. |
| Commonly misidentified lines<br>(See ICLAC register) | No commonly misidentified cell lines were used. |

# Animals and other research organisms

Policy information about studies involving animals; ARRIVE guidelines recommended for reporting animal research, and Sex and Gender in Research

| | |
|---|---|
| Laboratory animals | For amyloid β plaques in mouse brains, 5xFAD mice were reported previously (PMID: 17021169). The mouse strain #: 000664. Briefly, this transgenic mouse coexpress five FAD mutations [APP K670N/M671L (Swedish) + I716V (Florida) + V717I (London) and PS1 M146L + L286V], introduced into APP and PS1 cDNAs by site-directed mutagenesis, and then subcloned into exon 2 of the mouse Thy1 cassette (PMID: 17021169). In this study, female 7.5-month-old mice were used and weighed between 20-30g. The mice were housed with 40-60% humidity at 20-26 °C.<br>For dendrites of neurons in the mouse primary visual cortex, male postnatal day 89 and 273 Ai32 mice were used (Jackson Lab, stock#012569). Briefly, the mice express ChR2-eYFP following exposure to Cre-recombinase, allowing for conditional optogenetic activation. The mice were housed with 44% humidity at 22 °C. Mice strain#: 000664. Mice species: Mus musculus. |
| Wild animals | The study did not involve wild animals |
| Reporting on sex | This information has not been collected |
| Field-collected samples | The study did not involve samples collected from the field |
| Ethics oversight | All animal procedures associated with mice were approved by the Indiana University School of Medicine Institutional Animal Care and Use Committee (IACUC) and Purdue Animal Care and Use Committee (PACUC), and complied with all relevant ethical regulations. |

Note that full information on the approval of the study protocol must also be provided in the manuscript.

