## [Peer Review File · Nature Methods]

Peer Review Information

Manuscript Title: Deep Learning Driven Adaptive Optics for Single Molecule Localization Microscopy

Corresponding author name(s): Fang Huang

Editorial Notes: n/a

Reviewer Comments & Decisions:

Decision Letter, initial version:

Dear Fang,

Please let me begin by apologizing for our delays in making a decision on your manuscript. We were hoping to hear from a third reviewer, but ended up making the decision without them.

Your Brief Communication, "Deep Learning Driven Adaptive Optics for Single Molecule Localization Microscopy", has now been seen by two reviewers. As you will see from their comments below, although the reviewers find your work of considerable potential interest, they have raised a number of concerns. We are interested in the possibility of publishing your paper in Nature Methods, but would like to consider your response to these concerns before we reach a final decision on publication.

We therefore invite you to revise your manuscript to address these concerns. We ask that you address the calls for clarification, add additional quantitative assessments of performance, add relevant comparisons when necessary, and appropriately cite and discuss other approaches for 3D SMLM.

[Redacted] This URL links to your confidential home page and associated information about manuscripts you may have submitted, or that you are reviewing for us. If you wish to forward this email to co-authors, please delete the link to your homepage.

We hope to receive your revised paper within three months. If you cannot send it within this time, please let us know. In this event, we will still be happy to reconsider your paper at a later date so long as nothing similar has been accepted for publication at Nature Methods or published elsewhere.

OPEN SCIENCE REQUIREMENTS

REPORTING SUMMARY AND EDITORIAL POLICY CHECKLISTS

Please note that these forms are dynamic ‘smart pdfs’ and must therefore be downloaded and completed in Adobe Reader. We will then flatten them for ease of use by the reviewers. If you would like to reference the guidance text as you complete the template, please access these flattened versions at <http://www.nature.com/authors/policies/availability.html>.

DATA AVAILABILITY

All novel DNA and RNA sequencing data, protein sequences, genetic polymorphisms, linked genotype and phenotype data, gene expression data, macromolecular structures, and proteomics data must be deposited in a publicly accessible database, and accession codes and associated hyperlinks must be provided in the “Data Availability” section.

Please include a “Data availability” subsection in the Online Methods. This section should inform readers about the availability of the data used to support the conclusions of your study, including accession codes to public repositories, references to source data that may be published alongside the paper, unique identifiers such as URLs to data repository entries, or data set DOIs, and any other statement about data availability. At a minimum, you should include the following statement: “The data that

support the findings of this study are available from the corresponding author upon request”, describing which data is available upon request and mentioning any restrictions on availability. If DOIs are provided, please include these in the Reference list (authors, title, publisher (repository name), identifier, year). For more guidance on how to write this section please see: <http://www.nature.com/authors/policies/data/data-availability-statements-data-citations.pdf>

CODE AVAILABILITY

Please include a “Code Availability” subsection in the Online Methods which details how your custom code is made available. Only in rare cases (where code is not central to the main conclusions of the paper) is the statement “available upon request” allowed (and reasons should be specified).

MATERIALS AVAILABILITY

ORCID

Nature Methods is committed to improving transparency in authorship. As part of our efforts in this direction, we are now requesting that all authors identified as ‘corresponding author’ on published papers create and link their Open Researcher and Contributor Identifier (ORCID) with their account on the Manuscript Tracking System (MTS), prior to acceptance. This applies to primary research papers

only. ORCID helps the scientific community achieve unambiguous attribution of all scholarly contributions. You can create and link your ORCID from the home page of the MTS by clicking on 'Modify my Springer Nature account'. For more information please visit www.springernature.com/orcid.

Sincerely,
Rita

Rita Strack, Ph.D.
Senior Editor
Nature Methods

Reviewers' Comments:

Reviewer #1:

Remarks to the Author:

In the manuscript titled "Deep Learning Driven Adaptive Optics for Single Molecule Localization Microscopy", Peiyi Zhang et al. developed a deep-learning network (DL-AO) to direct inference of wavefront distortion for AO-SMLM, instead of the previous iterative trial-then-evaluate strategy. The DL-AO is the combination of the previously reported smNet1 and Kalman filter, which enhances the network stability compared with the previous smNet. The effectiveness of deep learning is built on a reasonable training dataset. Therefore, the authors load a linear combination of Zernike polynomials on the DM in a custom-designed SMLM setup, and they detect corresponding 3D PSF on the sample plane (3- μ m axial depth). The distorted PSFs and known ground-truth wavefronts (corresponding to Zernike polynomial coefficients) are set as the input and output of the DL-AO network. The average forward transmission time of the network is 0.1 s. It should be noted that the initial network parameters (illustrated in Supplementary Table 4), the number of network updates (3-20), and the network depth (smNet characteristics) need to be adjusted in different experiments. Ultimately, SMLM based on DL-AO can achieve ultra-high-resolution 3D imaging ($\sim 30 \times 30 \times 200 \mu$ m) in Tom20 proteins in COS-7 cells, immune-fluorescence-labeled amyloid- β fibrils in 125- μ m brain sections, and dendrites and spines in Thy1-ChR2-EYFP in 150-250 μ m brain sections.

1. Originality and significance

This manuscript mainly proposes a direct-inference AO-SMLM method, which combines the published smNet1 and Kalman filter to design a new DL-AO network for detecting sample aberrations in thick biological tissues (<200 μm). And the AO-SMLM based on DL-AO is able to efficiently correct the axial aberrations in biological tissues to achieve an axial resolution of tens of nanometers. However, we have some doubts about the innovativeness which are given as below.

- ① The innovative point of this manuscript is to propose a DL-AO network for directly predicting wavefront distortion and to use a DM for aberration compensation. However, the network architecture of DL-AO (Fig. 1 and Supplementary Notes 2.3) and sub-regions (Supplementary Notes 2.2) is the same as that of the published smNet1. Although the DL-AO employs the Kalman filter to reduce the estimation uncertainty, several publications have proposed that the Kalman filter is a powerful tool to limit network prediction error²⁻⁴. In addition, the transmission matrix for two axial positions is calculated by the Matlab function 'irregtform', which is the same as the published literature¹. Therefore, DL-AO is not sufficient to become a completely new network model for predicting specimen aberrations.
- ② Multiples of mirrors updates provide precise wavefront correction for all experiments presented in this manuscript, but such deep learning networks with multiple iterations to improve prediction accuracy have been reported already⁵. The manuscript suggests that the traditional image quality-based iterative algorithm can only perform on fixed target tissue structures within a plane or with a small axial extent. But there have been publications about 3D imaging by AO-SMLM⁶, so it is not possible to conclude whether multiple DL-AO iterations are fundamentally different from the traditional iterative method.

The real-time correction may be a proof to suggest DL-AO is different from the traditional iterative methods. However, the forward propagation of DL-AO network estimation is about 0.1 second, so 3-20 mirror updates for every experiment will spend hundred milliseconds or even seconds. Does the SMLM based on DL-AO still have a time advantage over the traditional methods?

In addition, according to current reports, super-resolution fluorescence imaging for observing organelle activity requires a high framerate with tens or hundreds Hz^{7, 8}. Thus, the advantages and new applications of DL-AO-SMLM over these super-resolution methods need further explanation.

- ③ The results of fixed static specimens cannot reflect the unique advantages of the DL-AO-SMLM, because the axial hollow distribution of Tom20 proteins and the structures of amyloid- β fibrils^{9, 10}, dendrites¹¹ and spines^{12, 13} have been already reported. And the fine imaging results by the DL-AO-SMLM do not show different and new structures.

2. Data and methodology

High axial-resolution imaging by the custom-designed SMLM setup based on DL-AO has a certain degree of confidence because the training dataset of the DL-AO network is generated from the SMLM setup. Types of specimens are diverse, including fluorescent beads, fixed in-vitro cells, and fixed brain slices. The demonstration focuses on the high axial resolution and reasons for the high axial resolution of DL-AO. However, I have the following questions:

① Testing and training datasets appear vastly different, so how can authors prove the results are authentic? 3D PSFs in the training dataset have significant differences in the axial distribution (Supplementary Fig. 2 and Supplementary Fig. 2), but the PSFs at different axial positions throughout a 2 μm axial range remained nearly invariant in specimen imaging (first paragraph on Page 9). It is puzzling whether such a large difference between the training and test datasets requires complex tuning operations before actual specimen imaging and whether these tuned parameters require some a priori knowledge to narrow the test range.

② The benefits of SMLM based on DL-AO are not fully demonstrated by designed experiments. The axial distribution of Tom20 proteins and the structures of amyloid- β fibrils⁹, ¹⁰, dendrites¹¹, and spines¹², ¹³ have been already reported, even with imaging results in vivo¹². It is not stated in figures and representation of the results whether the AO-SMLM based on DL-AO finds different structures from conventional methods because of its characteristic of high resolution. Besides, the feature of real-time correction is not shown in this manuscript.

3. The process and the presentation conform to the general adaptive optics process, showing comparison results before and after AO. And the effectiveness of AO is further demonstrated by the normalized intensity distribution. This manuscript uses Fisher information to demonstrate the axial high-resolution imaging after AO, which illustrates the high resolution of the SMLM based on DL-AO imaging from multiple perspectives. My doubt in this section is:

Why is it possible to show imaging results in Fig. 5a with z-axis positions at 133.2-134.8 μm , when the system images amyloid- β fibrils in a fixed 125- μm -depth brain slice? The z-axis position marked in the figure is the position of the imaging plane from the sample surface or the coverslip when the sample is in the water-based media?

4. The SMLM based on DL-AO has shown excellent 3D resolution in both in-vitro cellular imaging and brain slice tissue structure imaging, verifying the generality of the method in some applications. In the discussion, the manuscript states the method is no longer applicable when single molecule emissions are no longer identifiable in the cases of high aberration level or deep imaging depth. Regarding the conclusions, I think that the following issues need to be addressed:

① The test scenario of imaging depth needs to be stated in the conclusion. The manuscript gives a conclusion that the DL-AO advances the depth of 3D SMLM imaging up to $>130 \mu\text{m}$ in tissue, but this conclusion is derived from the imaging of dendrites and spines in immune-fluorescence-labeled Thy1-ChR2-EYFP mouse brain sections. In contrast, imaging of immune-fluorescence-labeled amyloid- β fibrils in brain slices is only at 50-70 μm . Whether the imaging depth is related not only to the AO methods and scattering the tissue scattering but also to the type of fluorescence, imaging target, etc. If it is as speculated above, it is not reasonable to conclude that 3D SMLM imaging with DL-AO can be through $>130 \mu\text{m}$ depth of tissue, while authors should further clarify the test scenario for this conclusion.

② The resolution of 3D SMLM with DL-AO needs to be quantitatively stated, which is not enough to describe as 'super resolution' in this manuscript. Although the distinguished minimum distances are shown in amyloid- β fibrils imaging and spines imaging, the best indication of the resolution of 3D SMLM

with DL-AO is in fluorescent-beads experiments. The resolution of the custom-designed system can be calculated from the size of the beads in actual values and imaging values.

5. The following experiments should be added to prove the conclusions in the manuscript:

① Dynamic sample imaging to support real-time calibration. It is recommended to supplement dynamic imaging results like organelles activity in some publications about super-resolution real-time imaging⁷, 8. The capability of real-time correction will be fully demonstrated if dynamic results are shown in a video.

② The limit imaging depth in biological tissues. It is recommended to supplement imaging results before and after AO in different-depth tissue, and further count SNRs variation or the resolution variation with depth for the same specimen. Thus, there will be an exact penetration depth of 3D SMLM with DL-AO in tissue, instead of the current plausible penetration depth of $>130\ \mu\text{m}$.

③ Quantifying 3D resolution. It is suggested that supplementary analysis of the lateral and axial normalized intensity distributions of fluorescence beads before and after AO illustrate the resolution of 3D SMLM with DL-AO, which paves the way for clarifying the irreplaceable application scenarios of the system.

④ A more reasonable initial value setting for the network to reduce the number of mirror updates and to be distinguished from the conventional methods. In this manuscript, DL-AO contains three networks, and raw data should be input into network 1 for the initial compensation, and the compensation result is considered whether to continue to use network 2 or network 3 for further compensation. The switching of multiple networks will take plenty of time for training and increase the difficulty and duration for judgment in the actual operation, although it can improve the precision of the output results. I wonder whether the robustness of a single network can be improved by a more reasonable initial value setting (for example, coefficients of the piston and tilt Zernike polynomials are set to 014, 15) to reduce the number of network switches and mirror updates.

6. References:

The literature review of AO-SMLM is insufficient in this manuscript. The authors should revise the following points in the introduction:

① The manuscript briefly describes three super-resolution microscopes that can achieve fluorescence imaging beyond the diffraction limit resolution but does not further cite the literature to illustrate the application scenarios of the three fluorescence microscopes. Explaining the unique application scenarios of SMLM helps the subsequent discussion of the irreplaceability of 3D SMLM with DL-AO and highlights the technical advantages.

② The manuscript uses an entire paragraph to describe the background of adaptive optics techniques. However, the most relevant to this paper is the adaptive optics technology associated with super-resolution microscopy, while only one sentence in the text summarizes all the articles that have been reported on AO-SMLM, indicating the current limitation of AO-SMLM is that it can only image in planar specimens. Not only the existing methods of adaptive optics in super-resolution microscopy are not introduced, but also the smNet, as important as the basis of DL-AO, is not described in the introduction.

7. Clarity and context:

- ① The abstract/summary is absent in the manuscript.
- ② In the introduction, the development, applications, and current limitations about super-resolution microscopy with adaptive optics need to be supplemented to clarify the significance of the SMLM with DL-AO.
- Besides, in the third paragraph, "the wavefront variations induced by lateral and axial positions from a collection of emitters in a volume." is different from the published articles on adaptive optics^{14, 15}. Because the lateral and axial offsets of the system are consistent for all sampling points, the piston and tilt terms are not considered in adaptive optics.

Reference:

1. Zhang, P. et al. Analyzing complex single-molecule emission patterns with deep learning. *Nat Methods* 15, 913-916 (2018).
2. Angeli, A., Desmet, W. & Naets, F. Deep learning of multibody minimal coordinates for state and input estimation with Kalman filtering. *Multibody System Dynamics* 53, 205-223 (2021).
3. Lu G, Ouyang W & Xu D in Deep kalman filtering network for video compression artifact reduction (Proceedings of the European Conference on Computer Vision 2018).
4. Roy, S.K., Nicolson, A. & Paliwal, K.K. in *Interspeech 2020* 2692-2696 (2020).
5. Hu, L., Hu, S., Gong, W. & Si, K. Deep learning assisted Shack-Hartmann wavefront sensor for direct wavefront detection. *Optics letters* 45, 3741-3744 (2020).
6. Siemons, M.E., Hanemaaijer, N.A.K., Kole, M.H.P. & Kapitein, L.C. Robust adaptive optics for localization microscopy deep in complex tissue. *Nat Commun* 12, 3407 (2021).
7. Dong, D. et al. Super-resolution fluorescence-assisted diffraction computational tomography reveals the three-dimensional landscape of the cellular organelle interactome. *Light Sci Appl* 9, 11 (2020).
8. Zhao, W. et al. Sparse deconvolution improves the resolution of live-cell super-resolution fluorescence microscopy. *Nat Biotechnol* 40, 606-617 (2022).
9. Schmidt, R. et al. Spherical nanosized focal spot unravels the interior of cells. *Nat Methods* 5, 539-544 (2008).
10. Gu, L. et al. Molecular-scale axial localization by repetitive optical selective exposure. *Nat Methods* 18, 369-373 (2021).
11. Lühns T, Ritter C & M, A. 3D structure of Alzheimer's amyloid- β (1-42) fibrils. *Proceedings of the National Academy of Sciences* 102, 6 (2005).
12. Pfeiffer, T. et al. Chronic 2P-STED imaging reveals high turnover of dendritic spines in the hippocampus in vivo. *Elife* 7 (2018).
13. Yuste, R. & Bonhoeffer, T. Genesis of dendritic spines: insights from ultrastructural and imaging studies. *Nat Rev Neurosci* 5, 24-34 (2004).
14. Liu, R., Li, Z., Marvin, J.S. & Kleinfeld, D. Direct wavefront sensing enables functional imaging of infragranular axons and spines. *Nature Methods* 16, 615-618 (2019).

15. Wang, K. et al. Rapid adaptive optical recovery of optimal resolution over large volumes. Nature Methods 11, 625-628 (2014).

Reviewer #3:

Remarks to the Author:

The manuscript describes an adaptive optics technique that is tailored for single molecular localization microscopy based on Deep learning. The proposed technique employs experimentally obtained sub-regions of the filtered raw images to train the neural network and directly infer the wavefront aberration for correction using mirror modes. The technique is further compared to model based method in simulation and experiments. The technique is successfully demonstrated in imaging a number of cell and tissue samples for correcting sample induced aberrations.

The study is clear and rather comprehensive. Simulation and experimental data are present at a high level. The I do have a couple of concerns related to the described method.

1. About the comparison to mode based AO method, clearly DL-AO can reach the optimal correction with less mirror updates and might overperform when there is significant extended volumetric structures; however one large advantage of the model based method is that it doesn't require single emission PSF, i.e. wide-field image without clearly separated image structure can also be used to estimate and reduce aberrations, which give more opportunities for AO correction, for example, initial correction before reaching the optimal SMLM imaging status. The author didn't mention this though they point out in the discussion section that the DL-AO strictly requires PSF images.

2. Because of the relatively long imaging time of SMLM, the DM update number or AO correction time is not as crucial as in some other methods, such as SIM and STED. Clearly DL-AO, as an AO method is effective, as shown in the cell and tissue imaging examples. However the comparison will be more interesting to be carried out between DL-AO and modal AO, especially about the highest localisation precision. I find the report is a bit inadequate in this aspect.

There are also a small number of minor issues in the manuscript, listed below.

-Statement in Line 20 appears to contradict to that in line 21 and 22.

-Line 216 suggest brain tissue is 200 um, but Fig. 4 suggest 110 nm (Line 369).

-Argument about post-processing irreversibility Line 32 is arguable because the statement in Line 30-32 is not clear.

Author Rebuttal to Initial comments

Details of Revision to Address Reviewers' Comments
Manuscript: N METH-BC49343, "Deep Learning Driven Adaptive Optics for Single Molecule Localization Microscopy"

We would like to thank the reviewers for their time and effort in providing us with comments and suggestions on how to improve our work. Based on these constructive comments, we have now revised and expanded the manuscript including new data, figures and videos, for which we are very excited to share. We believe that we have addressed the reviewers' comments and these revisions have significantly strengthened the manuscript. Please find below a point-by-point response to our reviewers' comments.

Reviewer #1 (Remarks to the Author):

In the manuscript titled "Deep Learning Driven Adaptive Optics for Single Molecule Localization Microscopy", Peiyi Zhang et al. developed a deep-learning network (DL-AO) to direct inference of wavefront distortion for AO-SMLM, instead of the previous iterative trial-then-evaluate strategy. The DL-AO is the combination of the previously reported smNet⁰⁰ and Kalman filter, which enhances the network stability compared with the previous smNet. The effectiveness of deep learning is built on a reasonable training dataset. Therefore, the authors load a linear combination of Zernike polynomials on the DM in a custom-designed SMLM setup, and they detect corresponding 3D PSF on the sample plane (3- μm axial depth). The distorted PSFs and known ground-truth wavefronts (corresponding to Zernike polynomial coefficients) are set as the input and output of the DL-AO network. The average forward transmission time of the network is 0.1 s.

It should be noted that the initial network parameters (illustrated in Supplementary Table 4), the number of network updates (3-20), and the network depth (smNet characteristics) need to be adjusted in different experiments.

Response: We thank the reviewer for the detailed summary of our work. We would like to clarify that we used the same initial network parameters throughout all data presented in the manuscript. This has been an important development criterion for us since its achievement represents the generalization of our proposed method toward various types of specimens. The parameters shown in **Supplementary Table 4** are for generating data used in training the three cascaded networks used in DL-AO. The number of mirror updates depends on the stopping criterion of DL-AO, which is explained in **Supplementary Note 2.5**. Since DL-AO monitors isolated single molecule emission patterns and proposes mirror changes when emission patterns deviate from the system optimum, it is possible to run DL-AO continuously without stopping (**Fig. R1**). We added **Supplementary Video 4** and **Supplementary Video 5**, which are examples of DL-AO continuously monitoring and compensating for >700 mirror updates without diverging, given a consistent signal-to-noise level. Moreover, we have kept the network depth (shown in **Supplementary Table 2**) the same throughout this manuscript. We observed that DL-AO works robustly in various types of specimens, including simulated SMLM frames, fluorescent beads, mitochondria networks in cells, amyloid- β plaques in the brains of mouse models of Alzheimer's disease, as well as

dendritic spines in cortical L5 Thy1+ pyramidal cells in the brains of Thy1-ChR2-EYFP transgenic mice. We revised the manuscript to clarify and highlight this aspect of DL-AO (Line 320-331).

Fig. R1: An example of DL-AO continuously compensating aberration for >700 mirror updates without diverging. The grey levels indicate the photon counts per pixel. The timestamp of each camera frame during imaging with AO is displayed on the top right corner of each panel. The bottom left panel shows the deformable mirror voltage map w.r.t current detection. This figure comes from screenshots of two frames in **Supplementary Video 4**.

*Ultimately, SMLM based on DL-AO can achieve ultra-high-resolution 3D imaging (~30*30*200μm) in Tom20 proteins in COS-7 cells, immune-fluorescence-labeled amyloid-β fibrils in 125-μm brain sections, and dendrites and spines in Thy1-ChR2-EYFP in 150-250 μm brain sections.*

1. Originality and significance

This manuscript mainly proposes a direct-inference AO-SMLM method, which combines the published smNet¹⁰ and Kalman filter to design a new DL-AO network for detecting sample aberrations in thick biological tissues (<200 μm). And the AO-SMLM based on DL-AO is able to efficiently correct the axial aberrations in biological tissues to achieve an axial resolution of tens of nanometers. However, we have some doubts about the innovativeness which are given as below.

① The innovative point of this manuscript is to propose a DL-AO network for directly predicting wavefront distortion and to use a DM for aberration compensation. However, the network architecture of DL-AO (Fig. 1 and Supplementary Notes 2.3) and sub-

regions (Supplementary Notes 2.2) is the same as that of the published smNet⁶⁰. Although the DL-AO employs the Kalman filter to reduce the estimation uncertainty, several publications have proposed that the Kalman filter is a powerful tool to limit network prediction error^{67,68}. In addition, the transmission matrix for two axial positions is calculated by the Matlab function 'irregtform', which is the same as the published literature⁶¹. Therefore, DL-AO is not sufficient to become a completely new network model for predicting specimen aberrations.

Response: The proposed deep learning driven adaptive optics represents a significant departure from the inference and image processing-focused deep learning development for microscopy. The manuscript demonstrates an important capacity of deep learning in driving and optimizing a hardware in microscopy instruments in pursuit of higher resolution and imaging depth. Leveraging the network design of smNet, DL-AO proposed novel approaches in deep learning training data augmentation—generating millions of experimental PSFs with highly accurate ground truth and cascaded network design to compensate large scale aberrations and stably converge approaching the level untouched by a specimen. Furthermore, our new data demonstrate the potential of DL-AO in compensating dynamic aberrations in near real-time (Fig. 2H, Supplementary Fig.17, Supplementary Videos 6 and 7). An example is shown in Fig. R2 in this letter. These unique designs and capacities make us believe DL-AO is an important milestone for the development of single molecule imaging through deep tissues and, of great potential, for dynamic specimens. We agree with the reviewer that Kalman filter is a widely used method in many fields, such as signal processing and system control, and we incorporated the Kalman filter in DL-AO because it helps stabilize our instrument control process, especially when the PSF quality in single molecule blinking dataset is uncontrollable (Supplementary Notes 2.4 and 5).

Fig. R2: DL-AO compensates for random and sudden wavefront changes during continuous SMLM acquisition. Images in the top row are the distorted wavefronts introduced during continuous imaging. A dot with a blue circle corresponds to a mirror update that introduces a random wavefront distortion (targeted level of 0.75 rad in W_{rms}). Each grey arrow points from an induced wavefront distortion to its corresponding mirror update. The dots without blue circles correspond to mirror updates driven by deep neural network. The single molecule blinking frames with random and sudden wavefront changes were continuously acquired for three minutes from the immune-fluorescence-labeled Tom20 specimen. See Supplementary Video 6 for the compensation process. See Supplementary Fig. 17 and Supplementary

Video 7 for more examples. This figure comes from **Fig. 2H**.

② Multiples of mirrors updates provide precise wavefront correction for all experiments presented in this manuscript, but such deep learning networks with multiple iterations to improve prediction accuracy have been reported already¹⁰. The manuscript suggests that the traditional image quality-based iterative algorithm can only perform on fixed target tissue structures within a plane or with a small axial extent. But there have been publications about 3D imaging by AO-SMLM¹⁰, so it is not possible to conclude whether multiple DL-AO iterations are fundamentally different from the traditional iterative method.

Response: We thank the reviewer for the comment. Please allow us to clarify. The deep learning work mentioned by the reviewer relies on the Shack-Hartmann wavefront sensor to function. However, implementing wavefront sensors for SMLM requires a bright and stationary point source¹. SMLM achieves super resolution by using photo-switchable or photo-convertible probes, which blink stochastically with limited photons, making it difficult to measure wavefront with the Shack-Hartmann wavefront sensor. DL-AO measures wavefront directly from the raw data acquired in SMLM, and it does not rely on additional wavefront sensing hardware (e.g. Shack-Hartmann) to help with the measurement. We added a discussion about this in **Supplementary Note 5**. As for traditional metric-based senseless AO methods, the image quality is extracted through a human-design metric. While intensity or sharpness metric may work robustly for confocal, two-photon, SIM, and STED, it is difficult to design an image quality metric that summarizes aberration-related information from a single molecule blinking frame, while ignoring irrelevant variations, such as intensity, background, and molecules' positions. As demonstrated, the current state-of-art metric design responds inconsistently to single molecules at different axial positions (**Supplementary Fig. 1**). Tested in practice, we found there are still chances to observe converging examples with traditional methods, but this level of performance is not stable in experiments involving three-dimensional structures (**Fig. 2D**, **Supplementary Fig. 12**). This observation is consistent with the demonstrated convergence problem for traditional methods based on simulated PSFs (**Supplementary Fig. 11**). In addition to the above-mentioned figures, we added further discussion on this topic in main text (**Lines 59-81**) and **Supplementary Note 1**.

The real-time correction may be a proof to suggest DL-AO is different from the traditional iterative methods. However, the forward propagation of DL-AO network estimation is about 0.1 second, so 3-20 mirror updates for every experiment will spend hundred milliseconds or even seconds. Does the SMLM based on DL-AO still have a time advantage over the traditional methods?

Response: We thank the reviewer for the comment. We added **Supplementary Video 4** to show the compensation process of DL-AO and the conventional metric-based AO (**Fig. R3**). To highlight the speed difference between DL-AO and the traditional method, we displayed the timestamp of each camera frame recorded during the compensation process. We observed that DL-AO finished compensation within two seconds, while metric-based AO took four minutes when compensating for the same aberration

(Supplementary Video 4). In addition, our new data demonstrate the potential of DL-AO in compensating dynamic aberrations in near real-time (Fig. 2H, Supplementary Fig. 17, and Supplementary Videos 6-7).

Fig. R3: An example of compensation process using DL-AO and the conventional metric-based AO. The left and right panel shows the raw data during the metric-based AO compensation and DL-AO compensation, respectively. The timestamp of each camera frame during imaging with AO is displayed on the top right corner of each panel. The grey levels indicate the photon counts per pixel. The time stamps were obtained from recorded files in each mirror updating cycle. This figure comes from Supplementary Video 4.

In addition, according to current reports, super-resolution fluorescence imaging for observing organelle activity requires a high framerate with tens or hundreds Hz^(7, 8). Thus, the advantages and new applications of DL-AO-SMLM over these super-resolution methods need further explanation.

Response: We thank the reviewer for the comment. The super-resolution fluorescence imaging technique mentioned by the reviewer is Structured Illumination Microscopy (SIM). The unique advantage of SMLM comparing to other super resolution techniques lies in measuring individual molecules without ensemble averaging^{3,3}, its potential in achieving ultra-high resolution, and molecular counting. There are review articles^{4,5} about this cross-modality comparison between these super resolution techniques. We have now added references that illustrate the application scenarios of the three fluorescence microscopes in the Introduction (**Line 42**). Live cell single molecule super-resolution imaging has been demonstrated in the past⁶⁻⁸. Although currently limited to thin specimens and cellular structures near the coverslip, the potential of performing live SMLM imaging through tissues and small animals holds enormous potential. Toward this

target, DL-AO aims at restoring the performance of SMLM for imaging structures in thick tissue specimens. Our newly added data shows the proof-of-principle of DL-AO's capacity in monitoring and compensating dynamic aberration changes (**Fig. 2H, Supplementary Fig. 17, Supplementary Videos 6-7**) and at the same time, continuous aberration compensation through dynamic structure changes (**Supplementary Video 8**). In our opinion, these capacities are solid steps toward the final target of enabling nanoscale observation through live tissues and animals.

③ The results of fixed static specimens cannot reflect the unique advantages of the DL-AO-SMLM, because the axial hollow distribution of Tom20 proteins and the structures of amyloid- β fibrils^{69, 68}, dendrites⁶⁹ and spines^{69, 68} have been already reported. And the fine imaging results by the DL-AO-SMLM do not show different and new structures.

Response: We thank the reviewer for the comment. Our DL-AO aims at restoring the performance of SMLM for imaging structures in the tissue specimen. Here we presented images of well-characterized cellular structures to demonstrate and validate the developed approach. The existing knowledge about these cellular organelles helps us in identifying the potential artifacts and provides visual assessments of the achievable resolution through the complex tissue environments using the developed method. The reports mentioned by the reviewer focus on *in vitro* purified protein samples, cells on the coverslip, or thin tissue specimens. DL-AO-SMLM is designed to allow ultra-high resolution imaging through tissues at large depths by compensating and restoring the precision and accuracy of the SMLM approach, expanding the capacity of conventional AO-SMLM approach for extended 3D tissue structures (**Supplementary Figs. 1, 11, 12**). Our amyloid- β fibrils, dendrites, and spines provide practical demonstrations of this new capacity of high-resolution imaging focusing on the depths that the images are taken from, and the precision and accuracy achieved in these thick specimens. In the manuscript, we also present well-understood and commonly imaged specimens for easy assessment of our aberration compensation effectiveness. For example, although the membrane contour of TOM20 protein-labeled mitochondria has been resolved previously, the achievable resolution and reconstruction accuracy deteriorate significantly when imaging through 35-135 μm water-based media (**Fig. 3 and Supplementary Figs. 13-14**). DL-AO-SMLM restores the significantly aberrated PSF and enables the capacity of resolving the sharp mitochondria membrane contour decorated by TOM20 proteins at $>130 \mu\text{m}$ through water-based media (**Fig. 3**). Moreover, DL-AO allows, for the first time, resolving the membrane contour of dendritic spines through 100-250 μm cut brain slices using light (**Fig. 6**), whereas all previous demonstrations are limited to $<40 \mu\text{m}$ tissue depths. The capacity to resolve and accurately quantify the shape and size of dendritic spines throughout large tissue thickness paves the way to link spine morphology and function and will facilitate studies of learning, memory, and brain disorders.

2. Data and methodology

High axial-resolution imaging by the custom-designed SMLM setup based on DL-AO has a certain degree of confidence because the training dataset of the DL-AO network is generated from the SMLM setup. Types of specimens are diverse, including fluorescent beads, fixed in-vitro cells, and fixed brain slices. The demonstration focuses on the high axial resolution and reasons for the high axial resolution of DL-AO. However, I have the following questions:

① Testing and training datasets appear vastly different, so how can authors prove the results are authentic? 3D PSFs in the training dataset have significant differences in the axial distribution (Supplementary Fig. 2 and Supplementary Fig. 2), but the PSFs at different axial positions throughout a 2 μm axial range remained nearly invariant in specimen imaging (first paragraph on Page 9). It is puzzling whether such a large difference between the training and test datasets requires complex tuning operations before actual specimen imaging and whether these tuned parameters require some a priori knowledge to narrow the test range.

Response: We apologize for the confusion from the reviewer about the figure and thank the reviewer for pointing this out. We have now reworded the figure caption of **Supplementary Fig. 2** to clarify this confusion. **Supplementary Fig. 2** is not a demonstration of how our training data should look like. **Supplementary Fig. 2** is a demonstration of the accuracy of single iteration inference using the DL-AO network. The arrows under “acqui.” point to PSF stacks measured under a specific deformable mirror voltage map. These measured PSFs input to the network for estimating aberrations. After obtaining the estimated wavefronts from the network, we simulated PSF stacks based on this estimation, which can be referred to as the network-estimated PSFs. We displayed the network estimated PSFs under the corresponding measured input PSFs in **Supplementary Fig. 2**, to visually demonstrate the one-time inference accuracy with the DL-AO network. We further quantified the similarity between experimentally obtained PSFs and the PSFs generated based on DL-AO estimation by calculating normalized cross-correlation (NCC) between them (**Supplementary Fig. 2C**). As for the confusing point in our descriptions “different axial positions throughout a 2 μm axial range remained nearly invariant in specimen imaging”, we were referring to the refractive index mismatch induced aberration. We demonstrated that DL-AO can compensate for this type of aberration in **Figs. 2C, 2D, 2J, Supplementary Figs. 11, 13, 14, and Supplementary Video 2**. We kept the same setting in the manuscript. Besides, our training data contains 6 million examples, which are generated based on the experimentally measured wavefront basis. Throughout all these demonstrations, we have kept the same network setting including architecture and training range, etc., and, in fact, it is exactly the same trained DL-AO networks that performed all the inference and updates throughout the manuscript including data on beads, simulations, cells and various types and thickness of tissues. The key here is the detection of single molecules since these emission patterns bear no influences from the underlying structures and thus provide a unique and pure source for aberration measurements, invariant across sample structures. We added the above descriptions in **Lines 326-331 and Lines 337-340**.

② The benefits of SMLM based on DL-AO are not fully demonstrated by designed experiments. The axial distribution of Tom20 proteins and the structures of amyloid- β fibrils^{69, 68}, dendrites⁶⁹, and spines^{69, 68} have been already reported, even with imaging results in vivo⁶⁹. It is not stated in figures and representation of the results whether the AO-SMLM based on DL-AO finds different structures from conventional methods because of its characteristic of high resolution. Besides, the feature of real-time correction is not shown in this manuscript.

Response: We thank the reviewer for this comment, DL-AO is developed to restore the resolution of SMLM imaging in tissue. In regards to DL-AO demonstrations through deep tissues and related demonstrations from prior works, we have provided a detailed response above and we have added these references to the main text and highlighted the emphasis on DL-AO's capacity of imaging through large depths while maintaining the ultra-high resolution capacity through tissues. We thank the reviewer for the suggestion of demonstrating dynamic aberration compensation using DL-AO. We have now included videos showing time-stamped AO compensation process using REALM and DL-AO (**Supplementary Videos 4-5**) to highlight the significantly improved compensation speed. Furthermore, we now included exciting proof-of-principle demonstrations on DL-AO in autonomously compensating random and abrupted aberration changes during a continuous single molecule imaging session (**Fig. 2H, Supplementary Fig. 17, Supplementary Videos 6-7**), an example figure of which is shown in **Fig. R2**. We hope these demonstrations will pave the way towards real-time aberration correction through living specimens and its application towards cellular/tissue targets that are key in their functions.

3. The process and the presentation conform to the general adaptive optics process, showing comparison results before and after AO. And the effectiveness of AO is further demonstrated by the normalized intensity distribution. This manuscript uses Fisher information to demonstrate the axial high-resolution imaging after AO, which illustrates the high resolution of the SMLM based on DL-AO imaging from multiple perspectives.

My doubt in this section is:

Why is it possible to show imaging results in Fig. 5a with z-axis positions at 133.2-134.8 μm , when the system images amyloid- β fibrils in a fixed 125- μm -depth brain slice? The z-axis position marked in the figure is the position of the imaging plane from the sample surface or the coverslip when the sample in the water-based media?

Response: We thank the reviewer for pointing this out. This is a typographical error in the manuscript. The correct thickness of the specimen is 85 μm , which is written in the figure caption. We updated **Fig. 5** with the correct labels. The z-axis position is the position of the imaging plane. We described this measurement in **SMLM acquisition with DL-AO** section in **Methods**: The imaging depths for immunofluorescence-labeled tissue specimens were measured by the differences of PIFOC or piezo nanopositioning system readings between apparent focuses of the region-of-interest and the fluorescent signal closest to the bottom coverslip surface.

4. The SMLM based on DL-AO has shown excellent 3D resolution in both in-vitro cellular imaging and brain slice tissue structure imaging, verifying the generality of the method in some applications.

Response: We thank the reviewer for positive evaluations of the achieved resolution in cells and tissue specimens and the appreciation of general applicability to the demonstrated specimen types.

In the discussion, the manuscript states the method is no longer applicable when single molecule emissions are no longer identifiable in the cases of high aberration level or deep imaging depth. Regarding the conclusions, I think that the following issues need to

be addressed:

① *The test scenario of imaging depth needs to be stated in the conclusion. The manuscript gives a conclusion that the DL-AO advances the depth of 3D SMLM imaging up to >130 μm in tissue, but this conclusion is derived from the imaging of dendrites and spines in immune-fluorescence-labeled Thy1-ChR2-EYFP mouse brain sections. In contrast, imaging of immune-fluorescence-labeled amyloid- β fibrils in brain slices is only at 50-70 μm . Whether the imaging depth is related not only to the AO methods and scattering the tissue scattering but also to the type of fluorescence, imaging target, etc. If it is as speculated above, it is not reasonable to conclude that 3D SMLM imaging with DL-AO can be through >130 μm depth of tissue, while authors should further clarify the test scenario for this conclusion.*

Response: We understand the reviewer's point of view. In this revision, we added more examples of 3D SMLM images for immune-fluorescence-labeled amyloid- β fibrils in brain slices in **Fig. SS10**. We also added **Fig. SS9** to show the representative images before and after DL-AO in different depth imaging through tissue specimens (**Fig. R4**). Certainly, the scattering of photons is increasingly dominant with increasing depth as shown in **Fig. SS9**. At the same time, we fully agree that the complex optical properties of tissues vary drastically among different tissue types. Liver, intestine, and muscle tissues have significantly higher scattering and autofluorescence than brain tissues, for example. We have now revised the manuscript to emphasize that the demonstrated depth is within brain tissues. To provide readers with sufficient information on the photon statistics of the detected single molecules at various depths and specimen types, we also reported photon counts and background counts obtained from the raw SMLM data in this manuscript in **Supplementary Table 1**.

Fig. R4: Representative images of raw data before and after DL-AO at different depth imaging through brain section. (A, B) Raw data of imaging immune-fluorescence-labeled amyloid- β fibrils in 200- μm -cut brain sections of 7.5-month-old 5XFAD female mouse. (C) Raw data of imaging Tom20 proteins in COS-7 cells using SMLM through unlabeled mouse brain section. (D, E, F) Raw data of imaging Thy1-ChR2-EYFP using SMLM in mouse brain section. This figure comes from Fig. SS9.

② *The resolution of 3D SMLM with DL-AO needs to be quantitatively stated, which is not enough to describe as 'super resolution' in this manuscript. Although the distinguished minimum distances are shown in amyloid- β fibrils imaging and spines imaging, the best indication of the resolution of 3D SMLM with DL-AO is in fluorescent-beads experiments. The resolution of the custom-designed system can be calculated from the size of the beads in actual values and imaging values.*

Response: We understand the reviewer's concern. Quantifying resolution in SMLM remains an active area of research due to its complexity^{9,10}. SMLM resolution not only relies on the localization precision of individual emitters, but also on labeling density, localization density, as well as localization biases. DL-AO is developed to restore the performance of SMLM in tissue by monitoring and correcting sample-induced aberrations and storing the emission pattern of a single molecule back to the instrument optimum. Effectively, this process restores the localization precision (by reducing PSF blur, especially in the axial direction), minimizes localization bias (by reducing PSF distortion, such as non-radially symmetrical wavefront shapes), and increases localization density (through restored PSF compactness). As a result, DL-AO-SMLM result in significantly higher quality reconstructions than without AO or conventional AO counterparts. However, labeling density and fluorophore brightness are sample dependent meaning SMLM, at its best, can only resolve the structures that have been labeled and its practical resolution can vary drastically depending on the type of fluorophore imaged and its photophysical property within the labeled specimen. CRLB¹¹ here provided a useful tool for evaluating localization precision. The Square root of CRLB represents the precision lower bound of an estimated parameter given an aberrated or DL-AO compensated PSF model with the consideration of the photon detection noise model. Importantly, we would like to note the other two factors leading to DL-AO's high-resolution results – minimized localization bias (thus artifacts) and increased localization density, cannot be reflected using CRLB, and thus these factors are demonstrated through PSF stacks and pupil phase before and after AO correction. With this consideration, we reported the theoretically achievable lateral and axial *precision* for super resolution images in this manuscript in **Supplementary Table 1**. Furthermore, we added an analysis of the lateral and axial normalized intensity distributions of fluorescence bead before and after AO in **Fig. SS2C-D (Fig. R5)**. From this figure, we can observe that the resulting PSFs and pupils from DL-AO, metric-AO, and no-AO together with their supported localization precision limit laterally and axially.

Fig. R5: Examples of the inconsistency between intensity distributions of fluorescent beads and theoretical precision in SMLM. (A) PSFs measured from 100-nm-diameter crimson beads and the corresponding phase retrieved pupil functions without AO, with metric-AO and with DL-AO. Scale bar: 2 μm . (B) Theoretically achievable localization precision without AO, with metric-AO and with DL-AO was calculated based on PSF model built from fluorescent beads measurements. The values correspond to PSFs with 1000 total photon counts and 50 background photons per pixel at axial positions of $-0.5 \mu\text{m}$ to $0.5 \mu\text{m}$. (C) Comparison on the in-focus PSF shapes and their lateral normalized intensity profiles before and after AO. The PSFs are zoom-in views of regions inside the orange box of A. Line profiles are calculated from the sum projection along the dashed white lines. (D) Comparison on the axial PSF shapes and their normalized intensity profiles before and after AO. The PSFs are zoom-in views of regions inside the magenta box of A. Line profiles are calculated from the sum projection along the dashed white lines. This example comes from Fig. SS2

5. The following experiments should be added to prove the conclusions in the manuscript:

(1) *Dynamic sample imaging to support real-time calibration. It is recommended to supplement dynamic imaging results like organelles activity in some publications about super-resolution real-time imaging^{71, 72}. The capability of real-time correction will be fully demonstrated if dynamic results are shown in a video.*

Response: We thank the reviewer for the comment. We have now included videos showing time-stamped AO compensation process using REALM and DL-AO (Supplementary Videos 4-5) to highlight the significantly improved compensation speed. Furthermore, we now included proof-of-principle demonstrations on DL-AO in autonomously compensating random and abrupt aberration changes during a continuous single molecule imaging session (Fig. 2H, Supplementary Fig. 17, Supplementary Videos 6-7). In addition, we have included a demonstration of continuous aberration compensation through dynamic structure changes (Supplementary Video 8). We hope the reviewer would share our excitement on these novel possibilities of DL-AO for dynamic aberration correction and we hope, these

demonstrations will pave the way towards real-time aberration correction through living specimens and its application towards cellular/tissue targets that are key in their functions but less abundant in their copy numbers.

② *The limit imaging depth in biological tissues. It is recommended to supplement imaging results before and after AO in different-depth tissue, and further count SNRs variation or the resolution variation with depth for the same specimen. Thus, there will be an exact penetration depth of 3D SMLM with DL-AO in tissue, instead of the current plausible penetration depth of >130 μm .*

Response: We thank the reviewer for the comment. As mentioned above, we have now added Fig. SS9 to show the representative images before and after DL-AO in different depth imaging through tissue specimens. In addition, we reported the photon counts and background counts in the raw data of our super resolution images in this manuscript in **Supplementary Table 1**. As observed in these data frames, photon scattering increases with increasing imaging depths. The scattering induced by the specimen can potentially be reduced by tissue clearing technologies¹²⁻¹⁵ or by reducing the local differences in refractive index between the subcellular content and the mounting medium¹⁶. We have now included this discussion in **Lines 345-353**. Furthermore, we added clarification on our conclusion as follows “and restores the resolution of 3D SMLM through >130 μm depth in brain tissues.” (**Line 319**).

③ *Quantifying 3D resolution. It is suggested that supplementary analysis of the lateral and axial normalized intensity distributions of fluorescence beads before and after AO illustrate the resolution of 3D SMLM with DL-AO, which paves the way for clarifying the irreplaceable application scenarios of the system.*

Response: We thank the reviewer for the comment. SMLM reconstructs a super-resolution image with molecular centers. The resolution of SMLM depends on multiple factors such as the accuracy and precision for estimating the emitters' center coordinates. Quantifying the resolution in SMLM remains an active research field. Instead of claiming a subjective resolution number in terms of nanometers, we reported the localization *precision* limit in both lateral and axial directions (square root of Cramer-Rao Lower Bound¹¹) for SMLM reconstructions in **Supplementary Table 1**. To further address the reviewer's concern, we added an analysis of the lateral and axial normalized intensity profiles of fluorescence bead before and after AO in **Fig. SS2C-D (Fig. R5C-D)**. From this figure, we can see that the size of PSFs from fluorescent beads cannot reflect the difference in Cramer-Rao Lower Bound, nor can it reflect the wavefront distortion (**Fig. SS2A**). We further added a comparison between SMLM reconstructions with DL-AO and their corresponding diffraction-limited images (**Fig. R6, Supplementary Fig. 16**), where the diffraction-limited images are generated by replacing single molecule localization points with their corresponding PSFs without aberration (i.e. instrument optimum PSF from beads). We hope these new data will provide readers with additional opportunities in evaluating the potential resolution achieved at various specimens and depths.

Fig. R6: Immune-fluorescence-labeled amyloid- β fibrils in 125- μm -cut brain sections of 7.5-month-old 5XFAD female mouse. (A-F) Comparison between images obtained from conventional diffraction-limited microscopy and 3D SMLM reconstruction with DL-AO. Color code indicates axial positions of single molecules. Conventional microscopy cannot resolve the 3D organizations of amyloid- β fibrils, e.g. single fibrils, bundle-like structures of amyloid- β fibrils. Conventional diffraction-limited images are generated by replacing single molecule localization points with their corresponding PSFs without aberration. The capacity to identify individual fibril nanostructures in the thick tissues with high resolution benefits amyloid pathology studies in determining the degree of fibril compaction at different brain areas. This comes from **Supplementary Fig. 16.**

④ *A more reasonable initial value setting for the network to reduce the number of mirror updates and to be distinguished from the conventional methods. In this manuscript, DL-AO contains three networks, and row data should be input into network 1 for the initial compensation, and the compensation result is considered whether to continue to use network 2 or network 3 for further compensation. The switching of multiple networks will take plenty of time for training and increase the difficulty and duration for judgment in the actual operation, although it can improve the precision of the output results. I wonder whether the robustness of a single network can be improved by a more*

reasonable initial value setting (for example, coefficients of the piston and tilt Zernike polynomials are set to 0^{th}) to reduce the number of network switches and mirror updates.

Response: We thank the reviewer for this comment. We added **Supplementary Video 4** to highlight the compensation speed difference between DL-AO and the traditional metric-based method. In this video, we displayed the timestamp of each camera frame during AO-SMLM compensation. DL-AO finished compensation within two seconds while compensating the same aberration using metric-based AO took four minutes (**Supplementary Video 4**). In terms of stopping criteria, DL-AO finishes compensation when the proposed wavefront change by Network3 has a maximum displacement smaller than $1/20 \lambda$ for three consecutive compensations (described in **Supplementary Note 2.5.2**). **Supplementary Video 4** shows that the entire DL-AO compensation routine finished within two seconds including data acquisition, segmentation, inference, mirror updates, and network switches through multiple iterations. Such speed achieved by DL-AO further allowed us to compensate for random and abrupt changes of wavefront in near real-time. Our new data (**Fig. R2**, **Fig. 2H**, **Supplementary Videos 6-7**, and **Supplementary Fig. 17**) show an exciting proof-of-principle demonstration of DL-AO's capacity in compensating dynamic aberration changes within tissue and cell specimens. We agree with the reviewer that it is imperative to further increase the speed of DL-AO to ideally allow subsecond compensation. To achieve this goal, a fully integrated deep learning control, sensor, and instrument control system is necessary, e.g. without needing a general operating system and storage. We are very interested in this future direction of DL-AO development.

6. References:

The literature review of AO-SMLM is insufficient in this manuscript. The authors should revise the following points in the introduction:

① The manuscript briefly describes three super-resolution microscopes that can achieve fluorescence imaging beyond the diffraction limit resolution but does not further cite the literature to illustrate the application scenarios of the three fluorescence microscopes.

Explaining the unique application scenarios of SMLM helps the subsequent discussion of the irreplaceability of 3D SMLM with DL-AO and highlights the technical advantages.

Response: We thank the reviewer for this comment. Please allow us to clarify. SIM, STED, and SMLM are parallel approaches that break the fundamental diffraction limit of light using different principles. The unique advantage of SMLM lies in measuring individual molecules without ensemble averaging^{2,3}, its potential in achieving ultra-high resolution in both live and fixed specimens, and molecular counting. Further improving the resolution and imaging depths of SIM, STED, and SMLM in cells and tissues remain ongoing research fields. There are review articles^{4,5} about this cross-modality comparison between these super resolution techniques. We added references that illustrates the application scenarios of the three fluorescence microscopes in the Introduction (**Line 42**). We have now reworded our explanation of SMLM to show the unique advantage of SMLM in measuring single molecules without ensemble averaging and described its potential in achieving nanometer resolution (**Lines 45-47**).

② *The manuscript uses an entire paragraph to describe the background of adaptive optics techniques. However, the most relevant to this paper is the adaptive optics technology associated with super-resolution microscopy, while only one sentence in the text summarizes all the articles that have been reported on AO-SMLM, indicating the current limitation of AO-SMLM is that it can only image in planar specimens. Not only the existing methods of adaptive optics in super-resolution microscopy are not introduced, but also the smNet, as important as the basis of DL-AO, is not described in the introduction.*

Response: We thank the reviewer for this comment. Please allow us to clarify. SIM and SMLM are both wide-field fluorescent microscopes for super resolution imaging. For adaptive optics in SIM, guide stars were generated by multiphoton excitation^{17,18}, or by embedding a fluorescent bead in specimen, providing a stable wavefront measurable both directly and indirectly^{19,20}. In contrast, signals from individual molecules blink stochastically with limited photons in SMLM experiments, providing no bright and stable guide stars¹. Besides, introducing an external guide star, such as a fluorescent bead, overshadows the single molecule emission patterns and introduces fluorescent backgrounds that are detrimental to the achievable resolution of SMLM. While intensity or sharpness metric may work robustly for confocal²¹, two-photon²², SIM^{23–25}, and STED²⁶, it is difficult to design an image quality metric that summarizes aberration-related information from a single molecule blinking frame, while ignoring irrelevant variations, such as intensity, background, and molecules' positions. We thank the reviewer for pointing out this confusing point. To clarify this, we added the above descriptions in **Introduction**. smNet is our recent development, which was demonstrated in its capacity to infer wavefront distortions from individual PSFs in simulation and its responsiveness in experimental datasets. We agree with the reviewer that smNet serves as the basis of DL-AO, therefore we introduced smNet in the **Design of DL-AO** section.

7. *Clarity and context:*

① *The abstract/summary is absent in the manuscript.*

Response: We thank the reviewer for the comment. We have now moved the abstract from the manuscript submission table to the main text so it is now showing together with the manuscript in one document.

② *In the introduction, the development, applications, and current limitations about super-resolution microscopy with adaptive optics need to be supplemented to clarify the significance of the SMLM with DL-AO.*

Besides, in the third paragraph, "the wavefront variations induced by lateral and axial positions from a collection of emitters in a volume." is different from the published articles on adaptive optics^{90, 99}. Because the lateral and axial offsets of the system are consistent for all sampling points, the piston and tilt terms are not considered in adaptive optics.

Response: We thank the reviewer for the comment. Please allow us to clarify, both articles mentioned by the reviewer use two-photon excitation (TPE) generated guide-stars. Due to the non-linear property of TPE, the generated guide stars have well-confined axial distribution—out-of-focus fluorescent probes have a small probability of being excited through TPE. During single molecule imaging, the detected fluorescence signals are from both in-focus and out-of-focus emitters, the wavefronts of which reaching a wavefront sensor are not the same. DL-AO focuses on single molecule emission patterns that are segmented from the single molecule blinking frames. These patterns, although originated from different axial and lateral positions, share a similar wavefront distortion (without the consideration of tip, tilt and defocus) because their confined lateral (within FOV) and axial location (generally $\pm 1 \mu\text{m}$) when detected and segmented by DL-AO. We thank the reviewer for pointing out this confusing point. To clarify this, we reworded this sentence into “Besides, wavefronts, if measured directly, are composed of not only the aberrated wavefront induced by the specimen, but also the wavefront variations from both in-focus and out-of-focus single emitters with different lateral and axial positions, making it difficult to infer wavefront distortion specific for the SMLM imaging volume.”.

Reference:

- (1) Zhang, P. et al. Analyzing complex single-molecule emission patterns with deep learning. *Nat Methods* 15, 913-916 (2018).
- (2) Angeli, A., Desmet, W. & Naets, F. Deep learning of multibody minimal coordinates for state and input estimation with Kalman filtering. *Multibody System Dynamics* 53, 205-223 (2021).
- (3) Lu G, Ouyang W & Xu D in Deep kalman filtering network for video compression artifact reduction (*Proceedings of the European Conference on Computer Vision* 2018).
- (4) Roy, S.K., Nicolson, A. & Paliwal, K.K. in *Interspeech 2020* 2692-2696 (2020).
- (5) Hu, L., Hu, S., Gong, W. & Si, K. Deep learning assisted Shack-Hartmann wavefront sensor for direct wavefront detection. *Optics letters* 45, 3741-3744 (2020).
- (6) Siemons, M.E., Hanemaaijer, N.A.K., Kole, M.H.P. & Kapitein, L.C. Robust adaptive optics for localization microscopy deep in complex tissue. *Nat Commun* 12, 3407 (2021).
- (7) Dong, D. et al. Super-resolution fluorescence-assisted diffraction computational tomography reveals the three-dimensional landscape of the cellular organelle interactome. *Light Sci Appl* 9, 11 (2020).
- (8) Zhao, W. et al. Sparse deconvolution improves the resolution of live-cell super-resolution fluorescence microscopy. *Nat Biotechnol* 40, 606-617 (2022).
- (9) Schmidt, R. et al. Spherical nanosized focal spot unravels the interior of cells. *Nat Methods* 5, 539-544 (2008).
- (10) Gu, L. et al. Molecular-scale axial localization by repetitive optical selective exposure. *Nat Methods* 18, 369-373 (2021).
- (11) Lühns T, Ritter C & M, A. 3D structure of Alzheimer's amyloid- β (1-42) fibrils. *Proceedings of the National Academy of Sciences* 102, 6 (2005).
- (12) Pfeiffer, T. et al. Chronic 2P-STED imaging reveals high turnover of dendritic spines

in the hippocampus in vivo. Elife 7 (2018).

(13) Yuste, R. & Bonhoeffer, T. Genesis of dendritic spines: insights from ultrastructural and imaging studies. Nat Rev Neurosci 5, 24-34 (2004).

(14) Liu, R., Li, Z., Marvin, J.S. & Kleinfeld, D. Direct wavefront sensing enables functional imaging of infragranular axons and spines. Nature Methods 16, 615-618 (2019).

(15) Wang, K. et al. Rapid adaptive optical recovery of optimal resolution over large volumes. Nature Methods 11, 625-628 (2014).

Reviewer #3 (Remarks to the Author)

The manuscript describes an adaptive optics technique that is tailored for single molecular localization microscopy based on Deep learning. The proposed technique employs experimentally obtained sub-regions of the filtered raw images to train the neural network and directly infer the wavefront aberration for correction using mirror modes. The technique is further compared to model based method in simulation and experiments. The technique is successfully demonstrated in imaging a number of cell and tissue samples for correcting sample induced aberrations.

Response: We thank the reviewer for the positive evaluations of our work.

The study is clear and rather comprehensive. Simulation and experimental data are present at a high level. The I do have a couple of concerns related to the described method.

1. About the comparison to mode based AO method, clearly DL-AO can reach the optimal correction with less mirror updates and might overperform when there is significant extended volumetric structures; however one large advantage of the model based method is that it doesn't require single emission PSF, i.e. wide-field image without clearly separated image structure can also be used to estimate and reduce aberrations, which give more opportunities for AO correction, for example, initial correction before reaching the optimal SMLM imaging status. The author didn't mention this though they point out in the discussion section that the DL-AO strictly requires PSF images.

Response: We thank the reviewer for this comment. We agree with the reviewer that the traditional sensorless-AO has this advantage and can perform an initial compensation before using DL-AO when there are no identifiable PSFs. We updated our **Discussion** as follows: "However, DL-AO requires at least two isolated and detectable PSFs to start compensation, and this requirement might be challenging to meet when the aberration level or imaging depth is significantly higher than the demonstrated cases where single molecule emissions are no longer identifiable. In those cases, an initial compensation with the conventional metric-based AO method would serve as a good start while DL-AO provides subsequent and continuous fine aberration corrections for high resolution single molecule reconstruction."

2. Because of the relatively long imaging time of SMLM, the DM update number or AO correction time is not as crucial as in some other methods, such as SIM and STED.

Clearly DL-AO, as an AO method is effective, as shown in the cell and tissue imaging examples. However the comparison will be more interesting to be carried out between DL-AL and modal AO, especially about the highest localisation precision. I find the report is a bit inadequate in this aspect.

Response: We thank the reviewer for this comment. We added **Fig. SS1** to show the comparisons on the achievable localization precisions for both methods. To satisfy the optimal condition on traditional methods, we avoided using volumetric structures and performed comparison on compensating aberration with in-focus PSF from a fluorescent bead through unlabeled brain section. Compared to conventional metric-based AO, we observed that DL-AO achieves higher localization precision (calculated through Fisher information). We also added comparisons with PSFs are slightly out-of-focus to illustrate the focus-dependent compensation performance of conventional metric-based methods in **Fig. SS2**. In addition, we have now included videos showing time-stamped AO compensation process using REALM and DL-AO (**Supplementary Videos 4-5**) to highlight the significantly improved compensation speed. Furthermore, we now included exciting proof-of-principle demonstrations on DL-AO in autonomously compensating random and abrupt aberration changes during a continuous single molecule imaging session (**Fig. 2H, Supplementary Fig. 17, Supplementary Videos 6-7**), and at the same time, continuous aberration compensation through dynamic structure changes (**Supplementary Video 8**). We hope the reviewer would share our excitement on these novel possibilities of DL-AO for dynamic aberration correction and we hope, these demonstrations will pave the way towards real-time aberration correction through living specimens and its application towards cellular/tissue targets that are key in their functions but less abundant in their copy numbers.

There are also a small number of minor issues in the manuscript, listed below.

-Statement in Line 20 appears to contradict to that in line 21 and 22.

Response: We thank the reviewer for pointing this out. To clarify this, we reworded our first sentence into “Fluorescent microscopy is an indispensable tool in visualizing cellular and tissue machinery with molecular specificity, however, in its conventional form, the resolution is limited to 250-700 nm laterally and axially due to the diffraction of light.” (**Lines 36-38**)

-Line 216 suggest brain tissue is 200 um, but Fig. 4 suggest 110 nm (Line 369).

Response: We thank the reviewer for this comment. The brain tissue was cut into 200 μm sections by a vibratome (1000 Plus, TPI Vibratome). **Fig. 4** shows Tom20 proteins imaged through this 200 μm machine-cut section. The reported depth in the figure is the imaging depth, which is measured by the axial position difference between the apparent focus of the region of interest and the fluorescent signal closest to the coverslip surface holding the tissue (**Methods**). The optically measured tissue thicknesses vary among samples that contain brain sections of the same machine-cut thickness. This mismatch between machine-cut thickness and optically measured thickness might be caused by variations in media volume between the bottom and top coverslips. We added a description of this observation in **Methods**.

-Argument about post-processing irreversibility Line 32 is arguable because the statement in Line 30-32 is not clear.

Response: We thank the reviewer for pointing out this clarity issue. We reworded these sentences as “One major reason is the distortion and blurring of single molecule emission patterns (i.e. PSFs) caused by the inhomogeneous refractive indices of the specimen. Such alterations of the detected emission patterns often reduce the information content carried by each detected photon, worsen the theoretically achievable localization precision, and thus causes significant resolution loss, which is irreversible by post-processing.”

References

1. Liu, S., Huh, H., Lee, S.-H. & Huang, F. Three-Dimensional Single-Molecule Localization Microscopy in Whole-Cell and Tissue Specimens. *Annu. Rev. Biomed. Eng.* **22**, 155–184 (2020).
2. Moerner, W. E., Shechtman, Y. & Wang, Q. Single-molecule spectroscopy and imaging over the decades. *Faraday Discuss.* **184**, 9–36 (2015).
3. Von Diezmann, A., Shechtman, Y. & Moerner, W. E. Three-Dimensional Localization of Single Molecules for Super-Resolution Imaging and Single-Particle Tracking. *Chemical Reviews* **117**, 7244–7275 (2017).
4. Valli, J. *et al.* Seeing beyond the limit: A guide to choosing the right super-resolution microscopy technique. *J. Biol. Chem.* **297**, 100791 (2021).
5. Wegel, E. *et al.* Imaging cellular structures in super-resolution with SIM, STED and Localisation Microscopy: A practical comparison. (2016). doi:10.1038/srep27290
6. Jones, S. A., Shim, S. H., He, J. & Zhuang, X. Fast, three-dimensional super-resolution imaging of live cells. *Nat. Methods* **8**, 499–505 (2011).
7. Huang, F., Schwartz, S. L., Byars, J. M. & Lidke, K. A. Simultaneous multiple-emitter fitting for single molecule super-resolution imaging. *Biomed. Opt. Express* **2**, 1377–1393 (2011).
8. Speiser, A. *et al.* Deep learning enables fast and dense single-molecule localization with high accuracy. *Nat. Methods* **18**, 1082–1090 (2021).
9. Banterle, N., Bui, K. H., Lemke, E. A. & Beck, M. Fourier ring correlation as a resolution criterion for super-resolution microscopy. *J. Struct. Biol.* **183**, 363–367 (2013).
10. Koho, S. *et al.* Fourier ring correlation simplifies image restoration in fluorescence microscopy. *Nat. Commun.* **10**, (2019).
11. Smith, C. S., Joseph, N., Rieger, B. & Lidke, K. A. Fast, single-molecule localization that achieves theoretically minimum uncertainty. *Nat. Methods* **7**, 373–375 (2010).
12. Gradinaru, V., Treweek, J., Overton, K. & Deisseroth, K. Hydrogel-Tissue Chemistry : Principles and Applications. *Annu. Rev. Biophys* **47**, 355–376 (2018).

13. Nudell, V. *et al.* HYBRiD: hydrogel-reinforced DISCO for clearing mammalian bodies. *Nat. Methods* **19**, 479–485 (2022).
14. Renier, N. *et al.* IDISCO: A simple, rapid method to immunolabel large tissue samples for volume imaging. *Cell* **159**, 896–910 (2014).
15. Chung, K. *et al.* Structural and molecular interrogation of intact biological systems. *Nature* **497**, 332–337 (2013).
16. Siemons, M. E., Hanemaaijer, N. A. K., Kole, M. H. P. & Kapitein, L. C. Robust adaptive optics for localization microscopy deep in complex tissue. *Nat. Commun.* **12**, 1–9 (2021).
17. Turcotte, R. *et al.* Dynamic super-resolution structured illumination imaging in the living brain. *Proc. Natl. Acad. Sci. U. S. A.* **116**, 9586–9591 (2019).
18. Li, Z. *et al.* Fast widefield imaging of neuronal structure and function with optical sectioning in vivo. *Sci. Adv.* **6**, 1–13 (2020).
19. Ji, N. Adaptive optical fluorescence microscopy. *Nat. Methods* **14**, 374–380 (2017).
20. Hampson, K. M. *et al.* Adaptive optics for high-resolution imaging. *Nat. Rev. Methods Prim.* **1**, 68 (2021).
21. Booth, M. J., Neil, M. A. A., Juskaitis, R. & Wilson, T. Adaptive aberration correction in a confocal microscope. *Proc. Natl. Acad. Sci. U. S. A.* **99**, 5788–92 (2002).
22. Débarre, D., Booth, M. J. & Wilson, T. Image based adaptive optics through optimisation of low spatial frequencies. *J. W. Hardy Opt. EXPRESS* **15**, 5131–5139 (1991).
23. Žurauskas, M. *et al.* IsoSense: frequency enhanced sensorless adaptive optics through structured illumination. *Optica* **6**, 370 (2019).
24. Thomas, B., Wolstenholme, A., Chaudhari, S. N., Kipreos, E. T. & Kner, P. Enhanced resolution through thick tissue with structured illumination and adaptive optics. *J. Biomed. Opt.* **20**, 26006 (2015).
25. Débarre, D., Botcherby, E. J., Booth, M. J. & Wilson, T. Adaptive optics for structured illumination microscopy. *Opt. Exp.* **16**, 9290–9305 (2008).
26. Gould, T. J., Burke, D., Bewersdorf, J. & Booth, M. J. Adaptive optics enables 3D STED microscopy in aberrating specimens. *Opt. Express* **20**, 20998 (2012).

Decision Letter, first revision:

Dear Fang,

Your Brief Communication entitled "Deep Learning Driven Adaptive Optics for Single Molecule Localization Microscopy" has now been seen again by the reviewers, whose comments are attached. In the light of their advice we have decided that we cannot offer to publish your manuscript in Nature Methods.

You will see that, while they find the paper improved, they are still not convinced about the the need for your tool, it's true performance in terms of resolution, and whether it is applicable to challenging samples. We think that these criticisms are sufficiently important as to prevent publication of your work in Nature Methods. We are not rejecting based on concerns about novelty, however, when reviewers question novelty, we do try to ensure that the performance benefits are sufficient to overcome these criticisms.

If you would like, I would be willing to consult with my colleagues at Nature Communications about a possible transfer with reviews to their journal.

I am sorry that we cannot be more positive on this occasion but hope that you find the reviewers' comments helpful when preparing your paper for submission elsewhere.

Sincerely,
Rita

Rita Strack, Ph.D.
Senior Editor
Nature Methods

Reviewer Comments:

Reviewer #1 (Remarks to the Author):

For the revised manuscript titled "Deep Learning Driven Adaptive Optics for Single Molecule Localization Microscopy", the comments are given as below.

1. Novelty

In the rebuttal, the authors pointed out that the DL-AO-smNet is different from previously published articles smNet1 in the initial value setting and cascaded networks, which is not enough to support DL-AO as an innovative deep-learning network. If the innovation of DL-AO SMLM is settled on a system-based generating a large number of reliable training sets, this kind of the generation of adaptive optical training dataset has been reported Ke Si's lab2. And DL-AO SMLM is not the first work to use deep learning to drive and optimize a hardware in microscopy instrument3. Although the above references are used for scanning fluorescence microscopes with adaptive optics, the method of reliable training dataset and the idea of microscope hardware driven by deep learning are the same with the design of DL-AO. Therefore, the novelty of this DL-AO method should be strengthened. The improvements of the optical configuration and the difference between the cascade smNet and the previous smNet except initial value setting should be explained.

2. Resolution.

The 13-57 nm super-resolution that the authors stated is very doubtful. Localization precision is not the resolution, since it is hard to differ the noise point and the signal point. The best way to verify the imaging authenticity and resolution is to compare with other super-resolution microscopy such as STED with similar resolution and the same sample, directly.

3. 3D SMLM penetration depth.

The authors claimed with $>130\ \mu\text{m}$ 3D SMLM penetration depth in brain tissues. To prove this statement, the authors should provide the whole 3D images directly of the 130 μm -thick brain tissues with continuously distributed signals from all the 0-130 μm layers, not only MIP signal with only $1\sim 2\ \mu\text{m}$ thickness.

4. Applications.

All of the results are obtained under a certain tissue depth, which show no much difference with reported super-resolution images with less penetration depth. This problem can be addressed by showing a continuous organism structure throughout large tissue thickness, and the structure cannot be clearly imaged with other super-resolution microscopies. In the body part of the manuscript and rebuttal letter, the authors emphasized that "The capacity to resolve and accurately quantify the shape and size of dendritic spines throughout large tissue thickness paves the way to link spine morphology and function and will facilitate studies of learning, memory, and brain disorders". Therefore, the authors can provide the 3D images of thick brain sections with complete neuron containing dendritic spines as an example (with claimed 13-57 nm resolution and $>130\ \mu\text{m}$ penetration depth), because the statistics of the dendritic spines on a complete neuron are closely related to animal's learning, memory, and etc.4,5.

5. Significance.

The imaging with brain sections in vitro is of less significance. Can this 3D SMLM be applied in in vivo imaging in deep tissue? If not, the significance will be dramatically reduced. For the in vitro brain sections, why not just cut it to thinner sections? Let alone that the obtained 3D imaging is through 130 μm brain section, not images of the whole 130 μm brain section.

References:

1. Zhang, P. et al. Analyzing complex single-molecule emission patterns with deep learning. *Nat Methods* 15, 913-916 (2018).
2. Hu, L., Hu, S., Gong, W. & Si, K. Deep learning assisted Shack-Hartmann wavefront sensor for direct wavefront detection. *Optics letters* 45, 3741-3744 (2020).
3. Saha, D. et al. Practical sensorless aberration estimation for 3D microscopy with deep learning. *Optics Express* 28, 29044-29053 (2020).
4. Qiao, Q. et al. Motor learning-induced new dendritic spines are preferentially involved in the learned task than existing spines. *Cell Rep* 40, 111229 (2022).
5. Bhatt, D.H., Zhang, S. & Gan, W.B. Dendritic spine dynamics. *Annu Rev Physiol* 71, 261-282 (2009).

Reviewer #3 (Remarks to the Author):

In this manuscript, titled "Deep Learning Driven Adaptive Optics for Single Molecule Localization Microscopy". Based on their published smNet, the work has used experimental data to perform wavefront-based training together with stabilising control through Kalman filter, the authors Peiyi Zhang et al. developed a deep-learning network method (DL-AO) to improve the network stability. The method can achieve accurate aberration correction with a small number of mirror update, results in an 0.1 second forward propagation time, which provide a potential for dynamic aberration correction using blinking molecules. The paper performs comparisons between metric-based AO and DL-AO and demonstrates the proposed method can achieve super-resolution in 3D in thick specimens.

The latest revision has made the manuscript clearer and successfully answered many questions raised in the first review. I only have one more question about the authors' response. I'm afraid the new added Figure SS2C-D (R5) create more confusion to me. The lower precision in panel B and the shape of PSF in Panel C suggest the metric AO with beads is not successful to even perform a system flattening. Can the authors confirm that and give an explanation?

Besides, I still have a few more general comments, given below.

- 1, Although the authors have used thick specimen for the study, I find the aberration introduced by the specimens are rather low judging by the raw data. The main aberration presented in e.g. Fig.3 and Fig.4. are spherical aberrations due to refractive index mismatch. The raw data PSF are not presented in Fig.5 and 6, so it is difficult to tell.
- 2, It can be seen from Supple. Fig. 5 that the network response to mirror modes varies unevenly with the mode number increases. How are the 28 deformation mirror modes are selected and are they sufficient for complex sample induced aberrations? Although work has been done with thick specimen only moderate aberrations in both modes and magnitude have been seen in raw data. It is not clear to me

how well and flexible the method can be upgraded to include a larger number of modes for correcting complex aberrations.

3. Further on about imaging with large aberrations from thick specimen, I can accept scattering is barely mentioned in the discussion as it is a rather separated topic, but it is common to notice in experiments with thick specimens that the image quality will not only be affected by the aberrations introduced at the equivalent back-pupil plane of the objective lens, but also the aberrations from sample conjugated plane, which will lead to image distortion and field dependent aberrations. These phenomenon are hard to correct with a single mirror shape even in a reasonable size field of view. It is likely the authors select more uniform tissues for demonstrate the method's performance, but it is a bit disappointing to see this issue is completely ignored even in the discussions especially because they have used significantly thick specimens. Especially DL-AO uses averaged PSFs from the cell image

4, It is unclear the method can be adopted equally successfully in other type of 3D-SMLM than that using biplane approach, e.g. with astigmatism or more complex phase masks. It would be interesting to see this be discussed.

Author Rebuttal, first revision:

Response to Reviewers' Comments (Summary)
Manuscript: NMETH-BC49343C, "Deep Learning Driven Adaptive Optics for Single Molecule Localization Microscopy"

We would like to thank the reviewers for their time and effort in providing us these valuable comments to improve our work. We are also glad to hear positive comments from reviewers on the comprehensiveness of our study, the technical development and demonstrations included in last revision of the manuscript.

In this new round of revision, we have taken all the comments very seriously. The revised manuscript includes new data, new figures, and new videos to highlight its achieved resolution through both decorrelation analysis and Fourier shell correlation, and to demonstrate the newly-extended capability of DL-AO in compensating wavefront distortions for Astigmatism-based 3D SMLM systems, and highly-complex wavefront distortions with up to 50 aberration (mirror) modes. With these revisions, we believe that we have addressed the reviewers' comments thoroughly and the manuscript, therefore, has been significantly improved.

Here we compiled a list of major changes that we have made in the revised manuscript for quick reference:

1. We have edited the manuscript and supplementary information to improve clarity in several aspects: description of novelty of the approach, resolution measurements, comparison with the prior state-of-the-art methods, newly extended capacities of DL-AO, and current remaining challenges and our vision for imaging through large brain slices with SMLM.
2. We added new **Supplementary Fig. 18** to show the resolution quantified through both decorrelation analysis and Fourier shell correlation, for all the super resolution images acquired with DL-AO at different imaging depths.
3. We added new **Supplementary Fig. 19** to show examples of observed field-of-view dependent aberrations when imaging through brain sections.
4. We added a new section: **Supplementary Note 7** to demonstrate DL-AO's performance in Astigmatism-based system.
5. We added a new section: **Supplementary Note 8** to demonstrate our investigations controlling 50 modes simultaneously with DL-AO.
6. We added a new section: **Supplementary Note 9** to discuss challenges in experimentally comparing super resolution techniques, challenges in whole brain slice reconstruction, and our vision for future developments.
7. We added new **Supplementary Video 9** to show single molecule blinking frames acquired at 22 different imaging areas when imaging through brain sections with and without DL-AO.
8. We added new **Supplementary Videos 10-11** to demonstrate DL-AO compensating artificially induced aberrations in Astigmatism-based system.
9. We added new **Supplementary Video 12** to demonstrate DL-AO compensating refractive index mismatch induced aberrations in Astigmatism-based system.
10. We added new **Table SS1** to show the training data generation parameters in Astigmatism-based setup.
11. We added new **Table SS2** to show the training data generation parameters for controlling 50 mirror modes simultaneously.

12. We added new Fig. SS9 to show measured mirror modes used for testing DL-AO in Astigmatism-based setup.
13. We added new Fig. SS10 to show the comparison between measured PSFs and PSFs simulated from network estimations based on a single measurement of an isolated molecule in Astigmatism-based setup.
14. We added new Fig. SS11 to show the characterization of neural network, which is trained for Astigmatism-based setup, responses to mirror mode changes using PSFs measured from fluorescent beads.
15. We added new Fig. SS12 to show the characterization of neural network, which is trained for Astigmatism-based setup, responses to mirror mode changes using PSFs measured from blinking molecules.
16. We added new Fig. SS13 to show the comparison between PSFs acquired in an Astigmatism-based setup before and after being restored from artificially introduced aberrations with DL-AO.
17. We added new Fig. SS14 to show the comparison between PSFs acquired in an Astigmatism-based setup before and after being restored from refractive index mismatch induced aberrations (at $\sim 146 \mu\text{m}$ depth) with DL-AO.
18. We added new Fig. SS15 to show the comparison between PSFs acquired in an Astigmatism-based setup before and after being restored from different levels of artificially introduced aberrations with DL-AO.
19. We added new Fig. SS16 to show the robustness tests of DL-AO in Astigmatism-based setup in both simulated and experimentally measured blinking frames.
20. We added Fig. SS17 to show the 50 mirror modes measured experimentally for training DL-AO network.
21. We added new Fig. SS18 to show the Comparison between measured PSFs and PSFs simulated from 50 mirror amplitudes estimated by the network, based on a single measurement of an isolated molecule.
22. We added new Fig. SS19 to show the characterization of neural network responses to 50 mirror mode changes using PSFs measured from fluorescent beads.
23. We added new Fig. SS20 to show the characterization of neural network responses to 50 mirror mode changes using PSFs measured from blinking molecules.
24. We added new Fig. SS21 to show the comparison between PSFs before and after being restored from artificially introduced aberrations with DL-AO when controlling 50 mirror modes simultaneously.

To help visualize these changes, we provided a highlighted version of the manuscript indicating places of addition and changes in both supplement and the main text.

Please find a detailed point-by-point response to the reviewers' comments attached below. Yours sincerely,

Fang Huang, Ph.D.
Reilly Associate Professor of Biomedical Engineering, Purdue University

Alexander A. Chubykin, Ph.D.
Associate Professor of Biological Sciences, Purdue University

Gary E. Landreth, Ph.D.
Martin Professor of Anatomy, Cell Biology and Physiology, Indiana University School of
Medicine

Detailed Responses to Address Reviewers' Comments**Reviewer #1 (Remarks to the Author):**

For the revised manuscript titled "Deep Learning Driven Adaptive Optics for Single Molecule Localization Microscopy", the comments are given as below.

1. Novelty

In the rebuttal, the authors pointed out that the DL-AO-smNet is different from previously published articles smNet^① in the initial value setting and cascaded networks, which is not enough to support DL-AO as an innovative deep-learning network. If the innovation of DL-AO SMLM is settled on a system-based generating a large number of reliable training sets, this kind of the generation of adaptive optical training dataset has been reported Ke Si's lab^②. And DL-AO SMLM is not the first work to use deep learning to drive and optimize a hardware in microscopy instrument^③. Although the above references are used for scanning fluorescence microscopes with adaptive optics, the method of reliable training dataset and the idea of microscope hardware driven by deep learning are the same with the design of DL-AO. Therefore, the novelty of this DL-AO method should be strengthened. The improvements of the optical configuration and the difference between the cascade smNet and the previous smNet except initial value setting should be explained.

Response: We thank the reviewer for this comment. We would like to clarify that the above-mentioned prior publications^{①②③} contain no demonstrations of driving hardware by deep learning. The reviewer has raised several concerns on the novelty, specifically in: (1) whether DL-AO network is an innovative deep neural network architecture. (2) whether training data generation for DL-AO network is innovative. (3) whether DL-AO hardware control is innovative. Please allow us to explain these points for each mentioned prior publication one by one.

1. Difference between DL-AO and smNet^①. smNet is our previously developed single molecule network for extracting multiplexed information (location, dipole orientation, and wavefront distortions) from single molecule emission patterns, while DL-AO is a feedback control for compensating wavefront distortions during data acquisition. Moving from estimation to control is non-trivial. First, there is a mismatch between the expected wavefront and the actual deformation in the correction element¹. To alleviate this effect, DL-AO controls deformable mirror using mirror modes, instead of the theoretical Zernike polynomials as demonstrated in the smNet work. Besides, the estimation uncertainty and bias of deep neural network increases with increasing training range, and thus the compensable range. To solve this trade-off between compensation range and control stability, DL-AO dynamically switches networks trained from different ranges and stabilizes the uncertainty using Kalman filter. Moreover, the estimation accuracy depends on whether the PSF model matches the experimental data. Some aberrations, such as the index

mismatch induced aberration that has an effective NA shrinkage, cannot be accurately modeled². We demonstrated that DL-AO restores the PSF in presence of index mismatch induced aberration, indicating DL-AO drives deformable mirror towards the right direction even if the testing scenario is outside the training range. In comparison, smNet showed no demonstrations of estimating index mismatch induced aberration. As for the difference between the deep neural networks used for DL-AO and smNet, there are three network variants in our previously published smNet work, and the current DL-AO work is developed from one of the variations with an increased number of neurons in input and output layers. The success of DL-AO reinforced the effectiveness of our developments in smNet. We hope that smNet can become a useful architecture for researchers in single molecule field to start their applications in the future.

2. Difference between DL-AO and the method reported in *Hu et al*². First, we would like to note, DL-AO is a wavefront **control** method and reference ② reports a wavefront **estimation** method. Besides, the method in reference ② estimates wavefront based on images formed by a Shack-Hartmann wavefront sensor. More specifically, the method in reference ② requires splitting the collected photons into two paths and inserting Shack-Hartmann wavefront sensor in one of the paths to detect wavefront. While our DL-AO network estimates directly from the raw data, without having to insert additional wavefront detection hardware in the original emission path. To further explain the differences, we would like to explain the wavefront estimation process using Shack-Hartmann wavefront sensor. Each segment of the wavefront is focused by one micro-lens array in the lens-array of the Shack-Hartmann wavefront sensor. The slope of each wavefront segment is estimated based on the shift in the focal point of each microlens. A final wavefront is reconstructed based on the slopes. The method reported in reference ② is a method of estimating wavefront shape from the image formed by micro-lens array, bypassing the slope-based reconstruction process. Based on the differences in the image formation process, the data generation methods between reference ② and DL-AO are fundamentally different. Besides, emitted light from blinking molecules is not bright and stable enough for Shack-Hartmann wavefront sensor to function³, and thus, the method in reference ② cannot be used in single molecule localization microscopy. As we can see, the training dataset reliability in reference ② depends on the efficacy of the Shack-Hartmann wavefront sensor—a hardware for estimating wavefront—which cannot function with signals from blinking molecules in SMLM.

3. Difference between DL-AO and the method reported in *Saha et al*³. We would like to note that DL-AO is a wavefront **control** method and reference ③ reports a wavefront **estimation** method. The method reported in reference ③ requires scanning a fluorescent bead at multiple pre-defined axial positions when used for widefield microscopes. Therefore, the method in reference ③ cannot be adapted to single molecule localization microscopy, as the strong fluorescence signals from beads will overshadow the signals of single molecules making single molecules undetectable. More importantly, embedding a fluorescent bead at the exact imaging area ($\sim 30 \mu\text{m} \times 30 \mu\text{m} \times 2 \mu\text{m}$) is impractical, especially in tissue specimens. While our DL-AO controls

deformable mirror using emission patterns from blinking molecules directly, without having to introduce a fluorescent bead.

2. Resolution.

The 13-57 nm super-resolution that the authors stated is very doubtful. Localization precision is not the resolution, since it is hard to differ the noise point and the signal point. The best way to verify the imaging authenticity and resolution is to compare with other super-resolution microscopy such as STED with similar resolution and the same sample, directly

Response: We thank the reviewer for this comment. We fully agree that the resolution of SMLM relies not only on the localization precision of individual emitters but also on labeling density, localization density, as well as localization biases. Please allow us to clarify that we stated 13-57 nm precision, instead of resolution, throughout the text. To address the reviewer's concern on resolution quantification, we added **new Supplementary Fig. 18 (Fig. R1)** to quantify resolution as a function of depth, using both decorrelation analysis⁴ and Fourier shell correlation⁵. A recent report⁶ demonstrated a 2D resolution from 37-41 nm at 5-20 μm depth in tissue specimen, which is quantified with decorrelation analysis. Using the same quantification approach, we found our resolution ranges from 14-31 nm for imaging depths from 35-134 μm shown in **Fig. R1**.

The direct comparison of DL-AO-SMLM with 3D-STED in the same sample can be challenging. This is because optimizing the labeling for the same epitopes in tissues in both SMLM and STED can be difficult. Among many other requirements, STED functions with photo-stable probes, while SMLM requires photo-switchable (blinking) probes. An attempted comparison between these modalities on the same specimen would be more of a measurement of sample or probe compatibility, rather than the technically achieved resolution between imaging modalities. We have now added **Supplementary Note 9.1** to discuss these challenges of comparing with other modalities.

Fig. R1: Resolution measurements of super resolution images acquired with DL-AO. (A) Lateral resolution measured with decorrelation analysis⁴. The resolutions measured by decorrelation are: 16.87 nm for Supplementary Fig. 14B, 17.15 nm for Fig. 5D, 14.87 nm for Fig. 5E, 27.77 nm for Fig. 6C, 21.85 nm for Fig. 6D, 29.16 nm for Supplementary Fig. 16D and Fig. 5A, 17.00 nm for Supplementary Fig. 13B, 30.27 nm for Supplementary Fig. 16E, 13.88 nm for Fig. 4B, 29.16 nm for Supplementary Fig. 16F, 17.00 nm for Fig. 3B, 19.93 nm for Fig. 3E, 30.27 nm for Fig. 6B. **(B)** 3D resolution measured with Fourier Shell Correlation (FSC) analysis⁵. The images were segmented into 49 regions and FSC was calculated for each region separately due to the limited computation resources. The mean and standard deviation of the resolution measured from 49 regions are: 41.10 ± 5.17 nm for Supplementary Fig. 14B, 55.52 ± 6.40 nm for Fig. 5D, 61.25 ± 7.48 nm for Fig. 5E, 61.22 ± 8.73 nm for Fig. 6C, 59.05 ± 5.20 nm for Fig. 6D, 78.50 ± 11.65 nm for Supplementary Fig. 16D, 81.60 ± 7.51 nm for Fig. 5A, 61.10 ± 6.69 nm for Supplementary Fig. 13B, 76.37 ± 13.10 nm for Supplementary Fig. 16E, 55.50 ± 8.19 nm for Fig. 4B, 81.04 ± 9.70 nm for Supplementary Fig. 16F, 53.50 ± 5.46 nm for Fig. 3B, 52.79 ± 6.19 nm for Fig. 3E, 80.22 ± 11.58 nm for Fig. 6B. Image reconstruction without AO is not included as a comparison here, because the image artifacts cannot be reflected by resolution measurements. This figure comes from new Supplementary Fig. 18.

3. 3D SMLM penetration depth.

The authors claimed with >130 μm 3D SMLM penetration depth in brain tissues.

To prove this statement, the authors should provide the whole 3D images directly of the 130 μm-thick brain tissues with continuously distributed signals from all the 0-130 μm layers, not only MIP signal with only 1~2 μm thickness.

Response: We thank the reviewer for this comment. The reviewer pointed out one of the major limitations in antibody-stained tissue imaging which we are also facing in this project: regardless of the fluorescent background and the acquisition time for scanning through the entire brain section, we were not able to observe continuous signals throughout the brain section. We expect that this is mainly due to the staining methods we have used. Recently developed tissue clearing technologies would be extremely important here in allowing homogenous labeling throughout the tissue depths. Combining the developed DL-AO approach with optically cleared tissues would be an exciting step forward allowing ultra-structures of tissue constituents to be imaged at nanometer scale. In addition, SMLM images are constructed by assembling hundreds of thousands of localized 3D centers of fluorescent molecules distribution in $\sim 30 \mu\text{m} \times 30 \mu\text{m} \times 2 \mu\text{m}$ volume. Thus, these reconstructed 3D SMLM images are different from MIP images (maximum intensity projection), which are commonly used in two-photon imaging. We have now added **Supplementary Note 9.2** to discuss challenges in whole slice reconstruction and our vision for future development.

4. Applications.

All of the results are obtained under a certain tissue depth, which show no much difference with reported super-resolution images with less penetration depth. This problem can be addressed by showing a continuous organism structure throughout large tissue thickness, and the structure cannot be clearly imaged with other super-resolution microscopies. In the body part of the manuscript and rebuttal letter, the authors emphasized that "The capacity to resolve and accurately quantify the shape and size of dendritic spines throughout large tissue thickness paves the way to link spine morphology and function and will facilitate studies of learning, memory, and brain disorders". Therefore, the authors can provide the 3D images of thick brain sections with complete neuron containing dendritic spines as an example (with claimed 13-57 nm resolution and >130 μm penetration depth), because the statistics of the dendritic spines on a complete neuron are closely related to animal's learning, memory, and etc⁴ ⁵.

Response: We thank the reviewer for pointing out the inconsistent wording in our manuscript. We have revised the above-mentioned sentences (**Lines 324-325**) and added **Supplementary Note 9.2** to reflect the current challenges in antibody-stained tissue specimens.

5. Significance.

The imaging with brain sections in vitro is of less significance. Can this 3D SMLM be applied in vivo imaging in deep tissue? If not, the significance will be dramatically reduced. For the in vitro brain sections, why not just cut it to thinner sections? Let alone that the obtained 3D imaging is through 130 um brain section, not images of the whole 130 um brain section

Response: We respectfully disagree with the reviewer that imaging *in vitro* is of less significance. We can understand the reviewer's point-of-view from the perspective of imaging neuron activities. However, fluorescent imaging in tissues reaches much broader scientific significance other than neuron activity measurements. It is often that with fixed or vitrified specimens, ultra-high resolution can be achieved for dissecting nanoscale features within the complex tissue environment. A prime example would be the demonstrated dataset on amyloid beta fibrils (submitted manuscript), which are the key factor in disease progression of AD (Alzheimer's disease). DL-AO aims at restoring the PSFs in these complex tissue environments to prevent resolution loss. We have demonstrated that DL-AO resolved these large aggregates of A β fibrils with high precision and accuracy in 3D, allowing us to trace individual fibrils through the A β aggregates. On the other hand, for investigating brain diseases, understanding the pathology primarily relies on analyzing fixed post-mortem brains or resected human tissues. This establishes another impactful direction for SMLM to decipher human diseases.

Cells and tissues are fixed in order to provide a 'snapshot' of the nature and distribution of molecules while minimizing changes from cell movement, sample degradation, etc. Modern microscopy technologies such as expansion microscopy, tissue clearing, DNA or Exchange-PAINT, and electron tomography all focus on imaging fixed/vitrified tissues/cells demonstrating the importance of imaging thick and fixed tissues. In addition, when sectioning a thick tissue into thin slices, important structures and rare phenotypes could be lost during the cutting process. Furthermore, the ability to image through thicker sections will also allow us to correlate super resolution structural imaging with functional imaging of the brain. We are very excited in the direction to move SMLM toward scanning the whole brain sections for the reasons stated above. We believe DL-AO is an indispensable tool in this next chapter of SMLM. We have now added **Supplementary Note 9.2** to discuss the challenges in imaging whole brain section.

Towards *in vivo* corrections using DL-AO, in our previous revision, we have provided proof-of-principle demonstrations of DL-AO's potential in correcting random and dynamic aberrations. We demonstrated DL-AO's capacity in correcting random and sudden wavefront changes autonomously (**Fig. 2H**, **Supplementary Fig. 17**, and **Supplementary Videos 6-7**) and its performance in aberration compensation during sample movement (**Supplementary Video 8**).

References:

- ① Zhang, P. et al. Analyzing complex single-molecule emission patterns with deep learning. *Nat Methods* 15, 913-916 (2018).
- ② Hu, L., Hu, S., Gong, W. & Si, K. Deep learning assisted Shack-Hartmann wavefront sensor for direct wavefront detection. *Optics letters* 45, 3741-3744 (2020).
- ③ Saha, D. et al. Practical sensorless aberration estimation for 3D microscopy with deep learning. *Optics Express* 28, 29044-29053 (2020).
- ④ Qiao, Q. et al. Motor learning-induced new dendritic spines are preferentially involved in the learned task than existing spines. *Cell Rep* 40, 111229 (2022).
- ⑤ Bhatt, D.H., Zhang, S. & Gan, W.B. Dendritic spine dynamics. *Annu Rev Physiol* 71, 261-282 (2009).

Reviewer #3 (Remarks to the Author):

In this manuscript, titled "Deep Learning Driven Adaptive Optics for Single Molecule Localization Microscopy". Based on their published smNet, the work has used experimental data to perform wavefront-based training together with stabilising control through Kalman filter, the authors Peiyi Zhang et al. developed a deep-learning network method (DL-AO) to improve the network stability. The method can achieve accurate aberration correction with a small number of mirror update, results in an 0.1 second forward propagation time, which provide a potential for dynamic aberration correction using blinking molecules. The paper performs comparisons between metric-based AO and DL-AO and demonstrates the proposed method can achieve super-resolution in 3D in thick specimens.

Response: We thank the reviewer for the positive evaluation of our work.

The latest revision has made the manuscript clearer and successfully answered many questions raised in the first review. I only have one more question about the authors' response. I'm afraid the new added Figure SS2C-D (R5) create more confusion to me. The lower precision in panel B and the shape of PSF in Panel C suggest the metric AO with beads is not successful to even perform a system flattening. Can the authors confirm that and give an explanation?

Response: We thank the reviewer for recognizing our effort in the previous revision. We agree with the reviewer that metric-based AO can perform system flattening, when the PSF is in-focus. We have demonstrated the success of metric-based AO in **Supplementary Video 4 (Fig. R2)** and

in obtaining our instrument optimum pupil function (Methods). But we would like to confirm that metric-based AO is not successful in performing system flattening robustly when the initial PSF is out-of-focus. We have visualized this performance in **Supplementary Video 5 (Fig. R3)**, which corresponds to the result in **Fig. SS2C-D**. Out-of-focus PSFs are unavoidable when imaging non-planar structures with single molecule localization microscopy, and these PSFs encode information about the axial molecular positions that are important for 3D super resolution reconstruction. To clarify the confusion in **Fig. SS2**, we modified this figure by labeling “Compensating aberration when PSF is out-of-focus” on top of it. To highlight that metric-based AO works well for in-focus PSF, we have revised our main text and included the following sentence: “We note that when the PSF is in-focus, metric-based AO works robustly to compensate aberrations, and thus metric-based AO was used in this work to perform system flattening in obtaining an instrument optimum pupil function for training DL-AO networks.” (Lines 224-227).

Fig. R2: Comparison between metric-based AO and DL-AO when compensating with a PSF in-focus. Compensations are performed with in-focus PSFs from 100-nm-fluorescent beads. The left and right panel shows the raw data during the metric-based AO compensation and DL-AO compensation, respectively. The timestamp of each camera frame during imaging with AO is displayed on the top right corner of each panel. The bottom left panel shows the deformable mirror voltage map w.r.t current detection. The deformable mirror was updated every 1 camera frame. The grey levels indicate the photon counts per pixel. This figure comes from **Supplementary Video 4**.

Fig. R3: Comparison between metric-based AO and DL-AO when compensating with a PSF out-of-focus. Compensations are performed with 100-nm-fluorescent beads that are slightly out-of-focus. The left and right panel shows the raw data during the metric-based AO compensation and DL-AO compensation, respectively. The timestamp of each camera frame during imaging with AO is displayed on the top right corner of each panel. The bottom left panel shows the deformable mirror voltage map w.r.t current detection. The deformable mirror was updated every 1 camera frame. The grey levels indicate the photon counts per pixel. This figure comes from **Supplementary Video 5**.

Besides, I still have a few more general comments, given below.

1, Although the authors have used thick specimen for the study, I find the aberration introduced by the specimens are rather low judging by the raw data. The main aberration presented in e.g. Fig.3 and Fig.4. are spherical aberrations due to refractive index mismatch. The raw data PSF are not presented in Fig.5 and 6, so it is difficult to tell.

Response: We thank the reviewer for this comment. We agree with the reviewer that the main aberration presented in **Fig. 3** is spherical aberration introduced by the refractive index mismatch. This is because **Fig. 3** is a demonstration of DL-AO's capacity in compensating significant index mismatch induced aberrations using constructed specimens. This capacity has never been demonstrated in existing AO methods for SMLM at this depth. As for **Fig. 4**, according to our *in situ* retrieved PSF model⁷ shown in **Fig. 4F**, we observed other aberration types than spherical aberrations. We also observed that these aberrations cause significant resolution loss and reconstruction artifacts. We have now added **Supplementary Video 9** (with selected screenshots

shown in Fig. R4) to visualize the raw data of Fig. 5A and Fig. 6A, as well as raw data of additional 20 areas imaged through tissue sections. Besides, we have quantified the working range of DL-AO by introducing distortions from 0.25 to 2.75 radians with deformable mirror (Fig. 2E-F). Examples of raw data before and after compensating distortions induced at different levels are shown in Supplementary Fig. 6 (Fig. R5). To have a clearer view of the PSF shapes in the blinking frames in Fig. R5, we measured PSFs from fluorescent beads nearby the blinking area, and the measured PSFs before and after DL-AO are shown in Supplementary Fig. 7 (Fig. R6).

Fig. R4: Single molecule blinking frames acquired with and without DL-AO when imaging through brain sections. The two panels are the raw blinking frames (after converting the analog-to-digital unit readings in camera frames to the effective photoelectrons, referred to as photon #) captured without and with DL-AO, respectively. The grey levels indicate the photon # per pixel. This figure comes from **new Supplementary Video 9**.

Fig. R5: SMLM frames before and after DL-AO compensating various amounts of induced aberrations. DL-AO was compensating aberrations at different levels (in W_{rms} , **Methods**) based on experimental blinking frames from immune-fluorescence-labeled Tom20 specimens. This figure comes from **Supplementary Fig. 6**.

Fig. R6: PSFs before and after DL-AO at various levels of induced aberrations. (A) Examples of PSFs before and after DL-AO, when compensating artificially induced aberrations. Compensations are performed in real time during SMLM experiments shown in Fig. R5 (Supplementary Fig. 6). PSFs are measured from 100-nm-diameter crimson beads nearby the compensation area post SMLM acquisition. Scale bar: 5 μm . (B) PSFs are measured under instrument optimum (Methods) from 100-nm-diameter crimson beads. Scale bar: 5 μm . (C) Quantitative comparisons between PSFs measured under instrument optimum and those measured before and after DL-AO using 3D normalized cross correlation (NCC). This figure comes from Supplementary Fig. 7.

2, It can be seen from Supple. Fig. 5 that the network response to mirror modes varies unevenly with the mode number increases. How are the 28 deformation mirror modes are selected and are they sufficient for complex sample induced aberrations? Although work has been done with thick specimen only moderate aberrations in both modes and magnitude have been seen in raw data. It is not clear to me how well and flexible the method can be upgraded to include a larger number of modes for correcting complex aberrations.

Response: We thank the reviewer for this comment. We have now added **Supplementary Note 8, Lines 174-177**, and **Lines 363-367** to discuss our investigations on controlling 50 mirror modes simultaneously with DL-AO. We observed that DL-AO network responded towards individual mirror deformations mostly in a one-to-one manner, when testing with both bead samples and blinking molecules cell specimens (Figs. R7 and R8). To test DL-AO's performance in restoring PSFs, when controlling 50 mirror modes, we introduced random wavefront distortions using the deformable mirror and compensated these distortions with DL-AO during SMLM experiments with immune-fluorescence-labeled Tom20 in COS-7 cells. We verified the PSF shape post correction by axially scanning fluorescent beads nearby the compensation areas. We observed similarities of 0.96 ± 0.02 (mean \pm s.t.d, N=22) between the PSFs post DL-AO and the instrument optimum PSF (Fig. R9). As we have included in our discussion, DL-AO requires at least two isolated and detectable PSFs to start compensation, and this requirement might be challenging to meet when single molecule emissions are no longer identifiable, a situation that is more probable to happen in aberrations composed of more complex mirror modes. In those cases, an initial compensation with the conventional metric-based AO method⁸ would serve as a good start while DL-AO provides subsequent and continuous fine aberration corrections for high resolution single molecule reconstruction (Lines 347-352).

Fig. R7: Characterizing neural network responses to 50 mirror mode changes using PSFs measured from fluorescent beads. (A) Network response to individual mirror mode changes. Each row of the response matrix shows the network responded mirror coefficients under a unit

change of each mirror deformation mode. After linear combining measured mirror modes (images below the title) with network responded coefficients, we obtained network estimated wavefront shape w.r.t. individual mirror mode changes (the 2nd column). The 1st column shows phase retrieved wavefronts from beads imaged individual mirror mode changes. The PSFs were measured with 100-nm-diameter crimson beads. PSFs from -1.5 μm to 1.5 μm around the focus, with 0.1 μm step size, were collected for characterizing network responses. (B) Difference between network estimated wavefront and phase retrieved wavefront (the first two columns in A). The top row shows the pixel-wise differences between wavefronts obtained from network estimation and that obtained from phase retrieval. The plot below shows the root mean square wavefront error⁹ (W_{rms} , **Methods**) of each wavefront difference. (C) Similarity between network estimated wavefront and phase retrieved wavefront. The similarity is quantified with 2D normalized cross correlation (NCC). This figure comes from new Fig. SS19.

Fig. R8: Characterizing neural network responses to 50 mirror mode changes using PSFs measured from blinking molecules. (A) Network response to individual mirror mode changes. Each row of the response matrix shows the network responded mirror coefficients under a unit

change of each mirror deformation mode. After linear combining measured mirror modes (images below the title) with network responded coefficients, we obtained network estimated wavefront shape w.r.t. individual mirror mode changes. The PSFs were experimental blinking frames measured from immune-fluorescence-labeled Tom20 specimens. A background map estimated by the temporal median filter was subtracted from each camera frame before segmentation. The intensity of each sub-region was estimated by summing up the photon counts in each pixel, after subtracting the median map. An intensity threshold of 2500 photons was applied to the segmented subregions to filter out PSFs with low photon counts. (B) Difference between network estimated wavefront (left column in A) and phase retrieved from beads (left column in Fig. R7). The top row shows the pixel-wise differences between wavefronts obtained from network estimation and that obtained from phase retrieval. The plot below shows the root mean square wavefront error⁹ (W_{rms} , **Methods**) of each wavefront difference. (C) Similarity between network estimated wavefront and phase retrieved wavefront. The similarity is quantified with 2D normalized cross correlation (NCC). This figure comes from new Fig. SS20.

Fig. R9: DL-AO simultaneously controls 50 mirror modes to restore PSFs from artificially induced aberrations. Compensations are performed in real time during SMLM experiment based on experimental blinking frames from immune-fluorescence-labeled Tom20 specimens. A background map estimated by the temporal median filter was subtracted from each camera frame before segmentation. The intensity of each sub-region was estimated by summing up the photon counts in each pixel, after subtracting the median map. An intensity threshold of 2500

photons was applied to the segmented subregions to filter out PSFs with low photon counts. Five examples of PSFs, pupil phases and mirror mode coefficients before and after DL-AO are shown in this figure. The artificial aberrations are induced at 0.5 radian level. PSFs are measured from 100-nm-diameter crimson beads nearby the compensation area post SMLM acquisition. This figure comes from new Fig. SS21.

3. Further on about imaging with large aberrations from thick specimen, I can accept scattering is barely mentioned in the discussion as it is a rather separated topic, but it is common to notice in experiments with thick specimens that the image quality will not only be affected by the aberrations introduced at the equivalent back-pupil plane of the objective lens, but also the aberrations from sample conjugated plane, which will lead to image distortion and field dependent aberrations. These phenomenon are hard to correct with a single mirror shape even in a reasonable size field of view. It is likely the authors select more uniform tissues for demonstrate the method's performance, but it is a bit disappointing to see this issue is completely ignored even in the discussions especially because they have used significantly thick specimens. Especially DL-AO uses averaged PSFs from the cell image

Response: We thank the reviewer for this comment. We agree with the reviewer that one correction element placed at the pupil plane cannot compensate for field-of-view dependent aberration, and thus DL-AO only corrects aberration shared within the field-of-view. We have now added a new Supplementary Fig. 19 (Fig. R10) to show the wavefront variations at four different segments in Fig. 5A and Fig. 6A. For the residual wavefront differences, analytical method, such as INSPR⁷, can be potentially applied to retrieve region-specific PSF models to localize molecules at different segments of the field-of-view. For more severe field-of-view dependent aberrations, one can apply DL-AO at different regions sequentially. Future developments that combine with the multi-pupil adaptive optics approach¹⁰ would allow simultaneous and independent correction for a large field of view. We thank the reviewer for this reminder, and we have now added discussions about field-of-view dependent aberrations in Lines 373-378.

Fig. R10: Examples of observed wavefront variations at four different segments of the imaging area. The region-specific wavefronts were calculated at each segment of the field-of-view with INSPR⁷. The left column shows wavefronts retrieved in dataset acquired without DL-AO. The right column shows wavefronts retrieved in dataset acquired with DL-AO. The two examples correspond to reconstructions in Fig. 5A and Fig. 6A respectively. This figure comes from new Supplementary Fig. 19.

4, It is unclear the method can be adopted equally successfully in other type of 3D-SMLM than that using biplane approach, e.g. with astigmatism or more complex phase masks. It would be interesting to see this be discussed.

Response: We thank the reviewer for this comment. We now extended DL-AO's demonstration to Astigmatism-based SMLM system as well (Supplementary Note 7). In addition, new Supplementary Videos 10-12 to show the DL-AO compensation process in Astigmatism-based setup. We found the PSFs became less distorted and similar to Astigmatism shape even after a

single compensation (**Fig. R11**). To characterize DL-AO's capacity in restoring Astigmatism PSFs, we introduced random wavefront distortions using the deformable mirror and compensated these distortions with DL-AO during SMLM experiments with immune-fluorescence-labeled Tom20 in COS-7 cells. We further verified the PSF shape post correction by axially scanning fluorescent beads nearby the compensation areas. Through phase retrieval, we found that the wavefront after DL-AO approaches the instrument optimum Astigmatism shape, with a residual of 0.35 ± 0.01 rad in W_{rms} (mean \pm s.t.d, $N=7$, **Fig. R12**). Besides, we observed that DL-AO is capable of restoring Astigmatism PSFs at ~ 146 μm in thickness with water-based imaging media (**Fig. R13**). We further evaluated the robustness of DL-AO on compensating different levels of wavefront distortion, from 0.25 to 2 radians in W_{rms} . We observed that PSFs were restored to approach the instrument optimum Astigmatism shape, with similarities of 0.96 ± 0.01 (mean \pm s.t.d, $N=15$ in NCC, examples shown in **Fig. R14**). By assessing the residual wavefront error post correction using both simulated data and experimental single molecule blinking data, we observed a residual level at 0.37 ± 0.1 radians (mean \pm s.t.d, $N=135$) for simulated data (**Fig. R15A**), and at 0.48 ± 0.26 radians (mean \pm s.t.d, $N=120$) of the induced level was compensated for experimental data (**Fig. R15B**). For other types of 3D-SMLM that use more complex phase masks, except for adapting DL-AO individually for each specific imaging modality, a potential solution is to electronically/mechanically remove the phase mask during compensation, and restore the phase mask after DL-AO. We have added the above discussions in **Supplementary Note 7**.

Fig. R11: Single molecule blinking frames acquired before and after one mirror update when using DL-AO in Astigmatism-based setup. Compensations are performed in real time during SMLM experiments when imaging immune-fluorescence-labeled Tom20 specimens. The left panel is the raw blinking frames (after converting the analog-to-digital unit readings in camera frames to the effective photoelectrons, referred to as photon #). The displayed frame rate is set to be the same as the acquisition frame rate, which is 50 Hz. The top right panel shows the deformable mirror voltage map w.r.t current detection. The deformable mirror was updated every 100 camera frames. This figure comes from **new Supplementary Video 10**.

Fig. R12: DL-AO restores Astigmatism PSF from artificially induced aberrations. Compensations are performed in real time during SMLM experiment based on experimental blinking frames from immune-fluorescence-labeled Tom20 specimens. A background map estimated by the temporal median filter was subtracted from each camera frame before segmentation. The intensity of each sub-region was estimated by summing up the photon counts in each pixel, after subtracting the median map. An intensity threshold of 1500 photons was applied to the segmented subregions to filter out PSFs with low photon counts. Five examples of PSFs, pupil phases and mirror mode coefficients before and after DL-AO are shown in this figure. The artificial aberrations are induced at 0.5 radian level. PSFs are measured from 100-nm-diameter crimson beads nearby the compensation area post SMLM acquisition. This figure comes from **new Fig. SS13**.

Fig. R13: DL-AO restores Astigmatism PSF from index mismatch induced aberrations. The SMLM blinking frames for compensation were acquired from immune-fluorescence-labeled Tom20 specimen at $\sim 146 \mu\text{m}$ from bottom coverslip surface in water-based media ($n = 1.35$). A background map estimated by the temporal median filter was subtracted from each camera frame before segmentation. The intensity of each sub-region was estimated by summing up the photon counts in each pixel, after subtracting the median map. An intensity threshold of 1500 photons was applied to the segmented subregions to filter out PSFs with low photon counts. PSFs were measured from 100-nm-diameter crimson beads nearby the compensation area post SMLM acquisition. Imaging depths were measured by the differences of PIFOC readings between the apparent focus of the region-of-interest and the bottom coverslip surface. Scale bar on PSFs from fluorescent beads: $2 \mu\text{m}$. See **new Supplementary Video 12** for the compensation process. This figure comes from **new Fig. SS14**.

Fig. R14: Examples of Astigmatism PSFs before and after DL-AO at various amounts of induced aberrations. (A) SMLM frames of Astigmatism setup before and after DL-AO compensating various amounts of induced aberrations. DL-AO was compensating aberrations in different levels (in W_{rms} , **Methods**) based on experimental blinking frames from immune-fluorescence-labeled Tom20 specimens. 100 camera frames were used for DL-AO estimation before each mirror update. A background map estimated by the temporal median filter was subtracted from each camera frame before segmentation. The intensity of each sub-region was estimated by summing up the photon counts in each pixel, after subtracting the median map. An intensity threshold of 1500 photons was applied to the segmented subregions to filter out PSFs

with low photon counts. The blinking data after DL-AO were compensation results after 19 mirror updates. **(B)** Examples of Astigmatism PSFs before and after DL-AO, when compensating artificially induced aberrations. PSFs are measured from 100-nm-diameter crimson beads nearby the compensation area post SMLM acquisition. Scale bar: 5 μm . **(C)** Quantitative comparisons between PSFs measured under instrument optimum and those measured before and after DL-AO using 3D normalized cross correlation (NCC). 31 PSFs from -1.5 μm to 1.5 μm axial positions were used to calculate the normalized cross correlation. **(D)** PSFs are simulated using pupil measured under instrument optimum (**Methods**) with an +1.5 (A.U.) Mirror Mode 3 (**new Fig. SS9**) offset to mimic Astigmatism-based setup. Scale bar: 5 μm . This figure comes from **new Fig. SS15**.

Fig. R15: Repeated tests of DL-AO in Astigmatism-based setup. (A) Summary of repeated tests of DL-AO for compensating aberrations of different levels (in W_{rms}) based on simulated SMLM blinking data. Each simulated SMLM frames contain 128×128 pixels, with a pixel size of 119 nm. The number of PSFs per frame was generated from the Poisson distribution with a mean of 13. Axial positions of molecules were generated from a uniform distribution from -1 to 1 μm range. The number of photon counts in each PSF was generated from an exponential distribution with a mean equal to 2500. The number of background photon counts in each frame was set to 10. (B) Summary of repeated tests of DL-AO for compensating aberrations in different levels (in W_{rms}) based on experimental blinking frames from immune-fluorescence-labeled Tom20 specimens. A background map estimated by the temporal median filter was subtracted from each camera frame before segmentation. The intensity of each sub-region was estimated by summing up the photon counts in each pixel, after subtracting the median map. An intensity threshold of 1500 photons was applied to the segmented subregions to filter out PSFs with low photon counts. This figure comes from new Fig. SS16.

References:

1. Haber, A. & Bifano, T. General approach to precise deformable mirror control. *Opt. Express* **29**, 33741–33759 (2021).
2. Petrov, P. N. & Moerner, W. E. Addressing systematic errors in axial distance measurements in single-emitter localization microscopy. *Opt. Express* **28**, 18616–18632 (2020).
3. Liu, S., Huh, H., Lee, S.-H. & Huang, F. Three-Dimensional Single-Molecule Localization Microscopy in Whole-Cell and Tissue Specimens. *Annu. Rev. Biomed. Eng.* **22**, 155–184 (2020).
4. Descloux, A., Größmayer, K. S. & Radenovic, A. Parameter-free image resolution estimation based on decorrelation analysis. *Nat. Methods* **16**, 918–924 (2019).
5. Diederich, B., Then, P., Jugler, A., Forster, R. & Heintzmann, R. cellSTORM—Cost-effective super-resolution on a cell phone using dSTORM. *PLoS One* **14**, e0209827 (2019).
6. Hao, X. *et al.* Three-dimensional adaptive optical nanoscopy for thick specimen imaging at sub-50-nm resolution. *Nat. Methods* **18**, 688–693 (2021).
7. Xu, F. *et al.* Three-dimensional nanoscopy of whole cells and tissues with in situ point spread function retrieval. *Nat. Methods* **17**, 531–540 (2020).
8. Siemons, M. E., Hanemaaijer, N. A. K., Kole, M. H. P. & Kapitein, L. C. Robust adaptive optics for localization microscopy deep in complex tissue. *Nat. Commun.* **12**, 1–9 (2021).
9. Wyant, J. C. & Creath, K. Basic Wavefront Aberration Theory for Optical Metrology. in *Applied Optics and Optical Engineering* **11**, 2 (Academic Press, 1992).
10. Park, J.-H., Kong, L., Zhou, Y. & Cui, M. Large-field-of-view imaging by multi-pupil adaptive optics. *Nat. Methods* **14**, 581–583 (2017).

Decision Letter, second revision:

Dear Fang,

Thank you for submitting your revised manuscript "Deep Learning Driven Adaptive Optics for Single Molecule Localization Microscopy" (NMETH-BC49343C). It has now been seen by one of the original referees and their comments are below. The reviewers find that the paper has improved in revision, and therefore we'll be happy in principle to publish it in Nature Methods, pending minor revisions to satisfy the referees' final requests and to comply with our editorial and formatting guidelines.

In response to the remaining referee concerns, we ask that you update the manuscript to discuss these concerns, especially the possibility of live/functional imaging with your method. We are not concerned about comments regarding novelty.

TRANSPARENT PEER REVIEW

Nature Methods offers a transparent peer review option for new original research manuscripts submitted from 17th February 2021. We encourage increased transparency in peer review by publishing the reviewer comments, author rebuttal letters and editorial decision letters if the authors agree. Such peer review material is made available as a supplementary peer review file. Please state in the cover letter 'I wish to participate in transparent peer review' if you want to opt in, or 'I do not wish to participate in transparent peer review' if you don't. Failure to state your preference will result in delays in accepting your manuscript for publication.

ORCID

IMPORTANT: Non-corresponding authors do not have to link their ORCIDs but are encouraged to do so. Please note that it will not be possible to add/modify ORCIDs at proof. Thus, please let your co-authors

know that if they wish to have their ORCID added to the paper they must follow the procedure described in the following link prior to acceptance:

Sincerely,
Rita

Rita Strack, Ph.D.
Senior Editor
Nature Methods

Reviewer #1 (Remarks to the Author):

Thanks for the response. For the revised manuscript titled "Deep Learning Driven Adaptive Optics for Single Molecule Localization Microscopy", there are still some issues that have not been positively addressed. Details are given as below.

1. Novelty

The methodological novelty of DLAO is incremental. DLAO is a kind of three network variants in previously published smNet work, which moves the aberration from estimation to hardware control. The achievement of network-driven aberration compensation has three major innovations. Firstly, the training dataset is generated from the orthogonal mirror modes that have been better adapted to the SMLM system. Secondly, the Kalman filter is used to enhance the stability of the network. Thirdly, the network obtained from a large training dataset is able to correct for index mismatch induced aberration (reply to Comment #1).

However, the three points cannot support DLAO as a new network, because the network architectures of DLAO and previous smNet are basically the same, both containing 3-5 convolutional layers, 7-11 residual blocks, and 0-2 fully connected layers, inter-layer batch normalization, an activation function of PreLU, and using 1×1 convolutional layers instead of fully connected layers for the final output (line 114-118 and Supplementary Table 2 in the manuscript).

The new network architecture has not been developed, and the advantage of DLAO SMLM over smNet for application to the system is that the cascaded network uses a more adapted training dataset for the system, which is not sufficient to support the innovation of DLAO.

The conclusion about a large number of aberration types enabling DL-AO to correct the aberration outside the training range like refractive index mismatch (Paragraph 2 in reply for Comment #1, and line 109-112 in the revised manuscript) is totally unfounded. Firstly, the increasing the number of mirror modes is not necessarily beneficial, which can be found from the experiments based on 50 mirror modes. The second is that the type of aberration introduced by the refractive index mismatch is spherical aberration^{1, 2}, which is already included in the training range (mirror modes 4&14 shown in Fig. S4-5).

2. Validating of the result authenticity of DLAO SMLM

Verifying the authenticity of the results is an important step in all deep learning methods. The authenticity of DLAO SMLM results can be demonstrated by imaging the same sample using a microscope with comparable localization accuracy and resolution, or by imaging a continuous sample in 3D, but the authors only explain the experimental difficulties of these two methods (Supplementary Note 9.1 and 9.2), and do not provide experiments to verify the imaging authenticity of DLAO SMLM.

Although artificial aberrations loaded on the deformable mirror to verify the authenticity of DLAO have been provided in the revised manuscript (lines 115-227, Fig. 1 and Supplementary Fig. 6), these artificial aberrations are actually included in the training set and are still significantly different from the actual aberrations of the biological samples.

3. Significance

There are contradictions regarding the experimental results shown in the revised manuscript and limited applications of DLAO SMLM:

1) The contradiction between real-time compensation (line 203) and the only imaging results of fixed or vitrified specimens. A pain point addressed by the development of a super-resolution imaging technique with real-time adaptive optics is the implementation of aberration calibration for dynamic biological samples. For example, the 188 Hz imaging rate of Hessian-SIM solves the long-standing problem of structural dynamics of mitochondrial cristae and structures by fluorescence imaging. There is little necessity to develop a fast adaptive optics technique for static vitrification samples.

2) The contradiction between static samples and functional imaging. The authors additionally state that “the ability to image through thicker sections will correlate super resolution structural imaging with functional imaging of the brain” and quote Ref. 61 to support this opinion (line 310). However, Ref. 61 is

a research based on living mice, and functional imaging of the brain presupposes living tissue samples. However, the results of living sample imaging in thick tissue are NOT shown in the revised manuscript, and the dynamic correction of artificial aberrations (Fig. 2H and lines 206-209) is not comparable to the variation in aberrations in living biological tissue.

3) The contradiction between thick-tissue aberration compensation and the inability to image continuous whole 3D samples. The authors state that the reason for penetrating thick tissue of SMLM imaging is that “important structures and rare phenotypes could be lost during the cutting process”. However, the DLAO SMLM cannot locate the axial position of important structures in continuous whole 3D samples (Supplementary Note 9.2), and the specific requirements of the DLAO SMLM for fluorescent dyes do not allow it to be used in conjunction with other microscopes known to be capable of continuous 3D imaging (Supplementary Note 9.1). The so-called axial location of important structures is unknown, so the experimental results of the DLAO SMLM in the revised manuscript have NOT shown the potential to address this issue.

4. Minor

The resolution reference used as a comparison (Ref. 6 in the reply) is missed.

Reference:

1. Marcinkevičius, A., Mizeikis, V., Juodkazis, S., Matsuo, S. & Misawa, H. Effect of refractive index-mismatch on laser microfabrication in silica glass. *Applied Physics A: Materials Science & Processing* 76, 257-260 (2003).
2. Diel, E.E., Lichtman, J.W. & Richardson, D.S. Tutorial: avoiding and correcting sample-induced spherical aberration artifacts in 3D fluorescence microscopy. *Nat Protoc* 15, 2773-2784 (2020).

Author Rebuttal, second revision:

Detailed Responses to Address Reviewers' Comments
Manuscript: NMETH-BC49343C, "Deep Learning-Driven Adaptive Optics
for Single Molecule Localization Microscopy"

We would like to thank the reviewers for their time and effort in providing us with comments and suggestions on how to improve our work. Please find below a point-by-point response to our reviewer's comments.

Reviewer #1 (Remarks to the Author):

Thanks for the response. For the revised manuscript titled "Deep Learning-Driven Adaptive Optics for Single Molecule Localization Microscopy", there are still some issues that have not been positively addressed. Details are given below.

1. Novelty

The methodological novelty of DLAO is incremental. DLAO is a kind of three network variants in previously published smNet work, which moves the aberration from estimation to hardware control. The achievement of network-driven aberration compensation has three major innovations. Firstly, the training dataset is generated from the orthogonal mirror modes that have been better adapted to the SMLM system. Secondly, the Kalman filter is used to enhance the stability of the network. Thirdly, the network obtained from a large training dataset is able to correct for index mismatch induced aberration (reply to Comment #1).

However, the three points cannot support DLAO as a new network, because the network architectures of DLAO and previous smNet are basically the same, both containing 3-5 convolutional layers, 7-11 residual blocks, and 0-2 fully connected layers, inter-layer batch normalization, an activation function of PreLU, and using 1×1 convolutional layers instead of fully connected layers for the final output (line 114-118 and Supplementary Table 2 in the manuscript).

The new network architecture has not been developed, and the advantage of DLAO SMLM over smNet for application to the system is that the cascaded network uses a more adapted training dataset for the system, which is not sufficient to support the innovation of DLAO.

Response: We thank the reviewer for this comment. However, we have different opinions on this novelty concern. New architectures of networks do not necessarily constitute innovation as similar performances can be achieved using completely different neural network architectures^{1,2}. In our design, we avoided unnecessary architecture modifications if an existing architecture already meets our target, in this case 'smNet'. Commonly used DL architectures such as 'ResNet-18' and 'U-Net' are adopted massively by the community and are generating impactful innovations across disciplines. As we have stated in our previous response letter, the success of DL-AO reinforced

the effectiveness of our prior developments in smNet³. We hope that smNet can become a useful architecture for researchers in the single molecule field to initiate their application or development in the future.

Moving from inference to hardware control during imaging, our focus is no longer on developing a new network architecture, but rather on improving network estimation accuracy on experimental PSFs, scaling the compensation capacity, as well as stabilizing the feedback control loop under different signal conditions. These were achieved through our developments in experimental wavefront-based training, stacked estimation networks, and stabilized feedback controls through Kalman filter, which allows, for the first time, robust control of an adaptive element correcting 28 aberration modes in near real-time during SMLM imaging.

The conclusion about a large number of aberration types enabling DL-AO to correct the aberration outside the training range like refractive index mismatch (Paragraph 2 in reply for Comment #1, and line 109-112 in the revised manuscript) is totally unfounded. Firstly, the increasing the number of mirror modes is not necessarily beneficial, which can be found from the experiments based on 50 mirror modes. The second is that the type of aberration introduced by the refractive index mismatch is spherical aberration^① ^②, which is already included in the training range (mirror modes 4&14 shown in Fig. SS4-5).

Response: We apologize for the reviewer's confusion on our demonstration against complex aberration modes and IMM cases. In response to reviewer 3's comment in the previous revision "It is not clear to me how well and flexible the method can be upgraded to include a larger number of modes for correcting complex aberrations.", we added demonstration on simultaneously compensating using 50 modes. Our experiments on compensating artificially induced distortions serve as an initial investigation on controlling 50 mirror modes simultaneously with DL-AO. We expect that future development in designing training data and neural network architecture will improve the inference accuracy of DL-AO through a large compensation range, ultimately enabling single-shot compensation during SMLM imaging in tissue specimens. Our measured pupil using phase retrieval algorithm⁴ showed significant wrapping in magnitude (Figs. 3H, 4F), while we kept the magnitude to be the same when generating training data (Fig. SS8). Therefore, we believe that refractive index mismatch-induced aberration is outside our training range. The capacity of DL-AO in compensating refractive index mismatch-induced aberration is our observation (Figs. 2D, 3, 4; Extended Data Figs. 8, 9; Supplementary Fig. 4, Fig. SS14).

To avoid confusion from the readers, we have now reworded the related sentences into "Here, we describe our developments in experimental-wavefront based training, stacked estimation networks, and stabilized feedback controls through Kalman filter (Fig. 1) built to allow a robust control an adaptive element correcting 28 aberration modes in near real-time during SMLM imaging, in presence of complex wavefront distortions, including the distortion induced by

refractive index mismatch. Simultaneously compensating a large number of aberration types also enables DL-AO's capacity in the autonomous control of the deformable mirror in response to random and dynamic aberration changes." (Line 105-112).

2. Validating of the result authenticity of DLAO SMLM

Verifying the authenticity of the results is an important step in all deep learning methods. The authenticity of DLAO SMLM results can be demonstrated by imaging the same sample using a microscope with comparable localization accuracy and resolution, or by imaging a continuous sample in 3D, but the authors only explain the experimental difficulties of these two methods (Supplementary Note 9.1 and 9.2), and do not provide experiments to verify the imaging authenticity of DLAO SMLM.

Although artificial aberrations loaded on the deformable mirror to verify the authenticity of DLAO have been provided in the revised manuscript (lines 115-227, Fig. 1 and Supplementary Fig. 6), these artificial aberrations are actually included in the training set and are still significantly different from the actual aberrations of the biological samples.

Response: We thank the reviewer for this comment. To demonstrate the resolving power of DL-AO SMLM, we presented images of previously characterized structures in cells and tissues (Figs. 3-6, such as mitochondria membrane, amyloid- β (A β) fibrils and membrane of the dendritic spines). These characterized three-dimensional cell/tissue structures enable us to assess aberration compensation effectiveness and reconstruction quality, help us in identifying the potential artifacts and provide visual assessments of the achievable resolution through the complex tissue environments. When imaging through thick tissues, the ground truth of the structure-of-interest is indeed challenging to obtain. It is thus important to establish well-characterized nanoscale endogenous tissue targets similar to the ones established by Thevathasan *et al.*⁵, for nuclear pore complex in cultured cells. Having such a target would streamline much of the validation and quantitative analysis of nanoscopy for tissue imaging.

3. Significance

There are contradictions regarding the experimental results shown in the revised manuscript and limited applications of DLAO SMLM:

1) The contradiction between real-time compensation (line 203) and the only imaging results of fixed or vitrified specimens. A pain point addressed by the development of a super-resolution imaging technique with real-time adaptive optics is the implementation of aberration calibration for dynamic biological samples. For example, the 188 Hz imaging rate of HessianSIM solves the long-standing problem of structural dynamics of mitochondrial cristae and structures by

fluorescence imaging. There is little necessity to develop a fast adaptive optics technique for static vitrification samples.

Response: We understand the reviewer's concern. The resolution of SMLM relies on the localization precision of individual emitters, labeling density, localization density, as well as localization biases⁶. As such, when imaging epitopes/protein targets that are abundant in terms of their copy numbers, prolonged AO compensation will not affect SMLM resolving power. However, many cellular/tissue targets that are key in their functions are low in abundance (32 copies for Nup96 per NPC), unnecessary photobleaching during AO will render incomplete reconstructions of these nanoscale features and thus limit the general applicability of SMLM. Prolonged compensation will significantly reduce the throughput of SMLM systems⁷. We are very excited in the direction to move SMLM toward scanning the whole brain section where throughput and automation are the key. We believe DL-AO is an indispensable tool in this next chapter of SMLM.

2) The contradiction between static samples and functional imaging. The authors additionally state that "the ability to image through thicker sections will correlate super resolution structural imaging with functional imaging of the brain" and quote Ref. 61 to support this opinion (line 310). However, Ref. 61 is a research based on living mice, and functional imaging of the brain presupposes living tissue samples. However, the results of living sample imaging in thick tissue are NOT shown in the revised manuscript, and the dynamic correction of artificial aberrations (Fig. 2H and lines 206-209) is not comparable to the variation in aberrations in living biological tissue.

Response: We thank the reviewer for bringing up this topic. We would like to note a very exciting demonstration in the previous AO-SMLM work by Siemons et al., ('REALM')⁸ demonstrates the possibility of correlating AO-SMLM and functional imaging in brain sections. Although experiments were performed in fixed tissue, the possibility of accessing tissue nanoscale features in the context of its function illustrated an impactful direction of SMLM in neuroscience. We focused on technology demonstration and extensive tissue-based validation in our work. Correlated functional imaging with live tissue nanoscopy is an exciting direction in the field. We expect that the demonstrated capacity of DL-AO makes it a central player in connecting our understanding of the brain's ultrastructure and function. SMLM in live tissue is non-trivial. Tissue-induced aberration and scattering, the limited temporal resolution, live-tissue compatible probes, and its labeling strategy represent challenges in revealing the ultrastructural dynamics in living tissues and animals. DL-AO allows robust compensation of complex wavefront through tissues in near real-time. We believe it represents one solid step towards this grand challenge of live tissue nanoscopy. We have highlighted the above considerations in the discussion section of the manuscript.

3) The contradiction between thick-tissue aberration compensation and the

inability to image continuous whole 3D samples. The authors state that the reason for penetrating thick tissue of SMLM imaging is that “important structures and rare phenotypes could be lost during the cutting process”. However, the DLAO SMLM cannot locate the axial position of important structures in continuous whole 3D samples (Supplementary Note 9.2), and the specific requirements of the DLAO SMLM for fluorescent dyes do not allow it to be used in conjunction with other microscopes known to be capable of continuous 3D imaging (Supplementary Note 9.1). The so-called axial location of important structures is unknown, so the experimental results of the DLAO SMLM in the revised manuscript have NOT shown the potential to address this issue.

Response: We understand the reviewer’s concern. However, we believe that we addressed this issue in our previous revision. Recently developed tissue clearing technologies⁹ would be extremely important here in allowing homogenous labeling throughout the tissue depths. Combining the developed DL-AO approach with optically cleared tissues would be an exciting step forward allowing ultra-structures of tissue constituents to be imaged at the nanometer scale. The SMLM field is moving towards scanning through a whole brain section, with advancements in adaptive optics, high throughput imaging⁷, light-sheet illumination^{10,11}, and tissue clearing⁹. We hope DL-AO can be an indispensable tool in this promising future of SMLM. We apologize for the reviewer’s confusion on **Supplementary Note 9.1** which explained the differences between SMLM and STED labeling method. This is the reason why we could not perform both SMLM and STED on the same specimen as requested in the previous review cycle. The developed DL-AO-SMLM approach is compatible with expansion microscopy¹² and cleared tissue, which addresses the labeling penetration as well as the possibility of achieving molecular resolution in tissue specimens.

4. Minor

The resolution reference used as a comparison (Ref. 6 in the reply) is missed.

References:

① *Marcinkevičius, A., Mizeikis, V., Juodkazis, S., Matsuo, S. & Misawa, H. Effect of refractive index mismatch on laser microfabrication in silica glass. Applied Physics A: Materials Science & Processing 76, 257-260 (2003).*

② *Diel, E.E., Lichtman, J.W. & Richardson, D.S. Tutorial: avoiding and correcting sample-induced spherical aberration artifacts in 3D fluorescence microscopy. Nat Protoc 15, 2773-2784 (2020).*

Response: We thank the reviewer for this reminder. In our previous revision, we quantified our resolution through decorrelation analysis¹³ and Fourier Shell Correlation¹⁴ in both 2D and 3D (**Supplementary Fig. 8**). These measurements provide initial estimates on the quality of the DL-

AO-SMLM images in tissue at various depths, considering localization precision of individual emitters, labeling density, localization density. However, we caution on using these metrics as benchmarks when comparing different imaging modalities as they do not account for system-and-algorithm generated biases and artifacts. AO-isoSTED¹⁵ and DL-AO-SMLM are novel but distinct imaging modalities that leverage different advanced hardware and computing innovations. In our opinion, the resolution is best demonstrated by the resolvability of target nanoscale structures, either intracellular or extracellular in tissues. For this reason, we focused on demonstrating DL-AO's performance in 3D reconstructions of the spine and amyloid-beta fibrils in the brain (Figs. 5, 6, Extended Data Fig. 10).

References

1. Bressemer, K. K. et al. Comparing different deep learning architectures for classification of chest radiographs. *Sci Rep* 10, (2020).
2. Ye, Z. et al. Comparison of Neural Network Architectures for Spectrum Sensing. in 2019 IEEE Globecom Workshops (GC Wkshps) 1–6 (2019). doi:10.1109/GCWkshps45667.2019.9024482.
3. Zhang, P. et al. Analyzing complex single-molecule emission patterns with deep learning. *Nat Methods* 15, 913–916 (2018).
4. Liu, S., Kromann, E. B., Krueger, W. D., Bewersdorf, J. & Lidke, K. A. Three dimensional single molecule localization using a phase retrieved pupil function. *Opt Express* 21, 29462–29487 (2013).
5. Thevathasan, J. V. et al. Nuclear pores as versatile reference standards for quantitative superresolution microscopy. *Nat Methods* 16, 1045–1053 (2019).
6. Möckl, L. & Moerner, W. E. Super-resolution Microscopy with Single Molecules in Biology and Beyond-Essentials, Current Trends, and Future Challenges. *J Am Chem Soc* 142, 17828–17844 (2020).
7. Barentine, A. E. S. et al. An integrated platform for high-throughput nanoscopy. *Nat Biotechnol* (2023) doi:10.1038/s41587-023-01702-1.
8. Siemons, M. E., Hanemaaijer, N. A. K., Kole, M. H. P. & Kapitein, L. C. Robust adaptive optics for localization microscopy deep in complex tissue. *Nat Commun* 12, 1–9 (2021).
9. Gradinaru, V., Treweek, J., Overton, K. & Deisseroth, K. Hydrogel-Tissue Chemistry: Principles and Applications. *Annu. Rev. Biophys* 47, 355–376 (2018).
10. Legant, W. R. et al. High-density three-dimensional localization microscopy across large volumes. *Nat Methods* 13, 359–365 (2016).
11. Power, R. M. & Huisken, J. A guide to light-sheet fluorescence microscopy for multiscale imaging. *Nat Methods* 14, 360–373 (2017).

12. Chen, F., Tillberg, P. W. & Boyden, E. S. Expansion microscopy. *Science* (1979) 347, 543–548 (2015).
13. Descloux, A., Grubmayer, K. S. & Radenovic, A. Parameter-free image resolution estimation based on decorrelation analysis. *Nat Methods* 16, 918–924 (2019).
14. Diederich, B., Then, P., Jugler, A., Forster, R. & Heintzmann, R. cellSTORM—Cost-effective super-resolution on a cellphone using dSTORM. *PLoS One* 14, e0209827 (2019).
15. Hao, X. et al. Three-dimensional adaptive optical nanoscopy for thick specimen imaging at sub-50-nm resolution. *Nat Methods* 18, 688–693 (2021).

Final Decision Letter:

Dear Fang,

I am pleased to inform you that your Article, "Deep Learning Driven Adaptive Optics for Single Molecule Localization Microscopy", has now been accepted for publication in Nature Methods. Your paper is tentatively scheduled for publication in our November print issue, and will be published online prior to that. The received and accepted dates will be May 24, 2022 and August 23, 2023. This note is intended to let you know what to expect from us over the next month or so, and to let you know where to address any further questions.

Over the next few weeks, your paper will be copyedited to ensure that it conforms to Nature Methods style. Once your paper is typeset, you will receive an email with a link to choose the appropriate publishing options for your paper and our Author Services team will be in touch regarding any additional information that may be required.

You will receive a link to your electronic proof via email with a request to make any corrections within 48 hours. If, when you receive your proof, you cannot meet this deadline, please inform us at rjsproduction@springernature.com immediately.

Please note that *Nature Methods* is a Transformative Journal (TJ). Authors may publish their research with us through the traditional subscription access route or make their paper immediately open access through payment of an article-processing charge (APC). Authors will not be required to make a final decision about access to their article until it has been accepted. [Find out more about Transformative Journals](https://www.springernature.com/gp/open-research/transformative-journals)

Authors may need to take specific actions to achieve [compliance](https://www.springernature.com/gp/open-research/funding/policy-compliance-faqs) with funder and institutional open access mandates. If your research is supported by a funder that requires immediate open access (e.g. according to [Plan S principles](https://www.springernature.com/gp/open-research/plan-s-compliance)) then you should select the gold OA route, and we will direct you to the compliant route where possible.

For authors selecting the subscription publication route, the journal's standard licensing terms will need to be accepted, including [self-archiving policies](https://www.springernature.com/gp/open-research/policies/journal-policies). Those licensing terms will supersede any other terms that the author or any third party may assert apply to any version of the manuscript.

Your paper will now be copyedited to ensure that it conforms to Nature Methods style. Once proofs are generated, they will be sent to you electronically and you will be asked to send a corrected version within 24 hours. It is extremely important that you let us know now whether you will be difficult to contact over the next month. If this is the case, we ask that you send us the contact information (email, phone and fax) of someone who will be able to check the proofs and deal with any last-minute problems.

If, when you receive your proof, you cannot meet the deadline, please inform us at rjsproduction@springernature.com immediately.

Once your manuscript is typeset and you have completed the appropriate grant of rights, you will receive a link to your electronic proof via email with a request to make any corrections within 48 hours. If, when you receive your proof, you cannot meet this deadline, please inform us at rjsproduction@springernature.com immediately.

Once your paper has been scheduled for online publication, the Nature press office will be in touch to confirm the details.

Once your paper has been scheduled for online publication, the Nature press office will be in touch to confirm the details.

Content is published online weekly on Mondays and Thursdays, and the embargo is set at 16:00 London time (GMT)/11:00 am US Eastern time (EST) on the day of publication. If you need to know the exact publication date or when the news embargo will be lifted, please contact our press office after you have submitted your proof corrections. Now is the time to inform your Public Relations or Press Office about your paper, as they might be interested in promoting its publication. This will allow them time to

prepare an accurate and satisfactory press release. Include your manuscript tracking number NMETH-A49343D and the name of the journal, which they will need when they contact our office.

About one week before your paper is published online, we shall be distributing a press release to news organizations worldwide, which may include details of your work. We are happy for your institution or funding agency to prepare its own press release, but it must mention the embargo date and Nature Methods. Our Press Office will contact you closer to the time of publication, but if you or your Press Office have any inquiries in the meantime, please contact press@nature.com.

Nature Portfolio journals [encourage authors to share their step-by-step experimental protocols](https://www.nature.com/nature-research/editorial-policies/reporting-standards#protocols) on a protocol sharing platform of their choice. Nature Portfolio 's Protocol Exchange is a free-to-use and open resource for protocols; protocols deposited in Protocol Exchange are citable and can be linked from the published article. More details can found at www.nature.com/protocolexchange/about.

Best regards,
Rita Strack